# FATE-MAP predicts teratogenicity and human gastrulation failure modes by integrating deep learning and mechanistic modeling

Joseph Rufo[1,2,3], Chongxu Qiu[1], Dasol Han[1,3], Naomi Baxter[1], Gabrielle Daley[1], Jasmine Dhillon[1], Felix Wong[4], James J. Collins [5,6,7] & Maxwell Z. Wilson [1,2,3] ✉

Gastrulation, a critical developmental stage involving germ layer specification and axes formation, is a major point of failure in human development, contributing to pregnancy loss and congenital malformations. However, due to ethical constraints and anatomical differences in animal models, the failure modes underlying human gastrulation remain poorly understood. To elucidate these failure modes, we introduce FATE-MAP (Failure Analysis and Trajectory Evaluation via Mechanistic-AI Prediction), an integrated platform that combines high-throughput perturbations of human 2D gastruloids with quantitative phenotypic mapping, predictive deep learning, and mechanistic morphogen modeling. Analyzing over 2000 drug-treated human 2D gastruloids, we mapped a phenotypic morphospace that separates canonical patterning, in which primitive-streak fates are correctly specified and radially organized, from failure modes, defined as departures from this organization and marked by a loss of a required fate and/or radial symmetry. To predict and interpret patterning outcomes, FATE-MAP combines a transformer linking chemical structure to phenotype with PDE simulations of morphogen transport and cell fate specification, and projects both outputs onto the experimentally defined morphospace. Applying this framework, we flagged two clinical molecules as potential teratogens and identified two parameters, cell density and SOX2 stability, that form orthogonal morphospace axes along which canonically patterned gastruloids systematically vary. FATE-MAP thus provides a roadmap for decoding human developmental trajectories and accelerating safe therapeutic discovery.

Human gastrulation, which involves two symmetry-breaking events and the specification of the germ layers, has been hypothesized to act as a go, no-go decision for the embryo, serving as an early test of the fidelity of developmental processes that are essential for proper formation later[1,2]. As many as 30% of human pregnancies are predicted to fail at this critical stage[3,4], yet because of anatomical differences between human and other vertebrate models of gastrulation[5], as well as ethical constraints on experimentation with human embryos[6], we lack a comprehensive view of the key factors that govern human gastrulation and how perturbations to these factors lead to failure.

Inborn errors of gastrulation can lead to either spontaneous abortion or congenital malformations[7,8]. Notably, congenital malformations cause more deaths than all pediatric cancers[9] and have a higher mortality rate than that of stroke, Alzheimer's disease, and diabetes combined[10,11]. Despite this societal burden, little progress has been made in prevention or prediction, with 50–80% of congenital malformations having no known cause[12,13]. Teratogenicity testing has traditionally relied on exposing model organisms, such as rats, mice, and zebrafish, to potential teratogens and assessing gross morphological changes. Critically, such models do not capture many human-specific teratogens as mice and rats are notoriously resistant to thalidomide-induced birth defects[14,15], highlighting the need for quantitative methods to predict even minor harm to the developing embryo.

To address these challenges, we developed FATE-MAP (Failure Analysis and Trajectory Evaluation via Mechanistic-AI Prediction), an integrated experimental and computational pipeline that combines high-throughput drug screening with phenotypic mapping, mechanistic modeling, and deep learning models to identify the failure modes of human gastrulation, uncover their mechanistic origins, and improve our ability to forecast human-specific developmental risks. Central to this approach is the construction of a morphospace[16,17], a low-dimensional, quantitative map of morphological phenotypes, that captures both canonical and disrupted trajectories of human gastrulation. Here, we use morphospace specifically to describe the diversity of 2D patterning outcomes in micropatterned stem cell colonies, not to imply full embryonic morphology or organization. Generating such a map requires an experimental model system that faithfully recapitulates early human development while enabling high-throughput perturbation and analysis. Recent advances in stem cell–based embryo models provide this opportunity[18], offering experimentally tractable systems to probe human developmental trajectories under defined conditions.

While 3D models such as blastoids and synthetic embryo models (SEMs) can recapitulate aspects of early development, they often suffer from variability and low formation efficiencies[19–25]. On the other hand, 2D or 2.5D models that leverage lithographic micropatterned surfaces achieve near uniform formation of $10^3$-$10^4$ constructs per experiment[26–28], offering highly scalable and reproducible systems. These models accurately reproduce the disc-like geometry of the epiblast, albeit at the cost of altering the topology of surrounding extraembryonic tissues such as the amnion. Among these, the 2D gastruloid, a micropatterned disc of embryonic stem cells, serves as a reliable model of the human primitive streak, differentiating into radially patterned germ layers and recapitulating all major cell types of the gastrulating human embryo[29]. Leveraging this platform, we generated an experimental morphospace from ~2000 gastruloids exposed to a diverse library of small molecules targeting developmental pathways. Using nonlinear dimensionality reduction, we embedded the resulting phenotypic profiles into a low-dimensional map, where both canonical and disrupted trajectories formed structured regions of morphospace, revealing discrete failure modes marked by symmetry loss or germ layer absence.

Although nonlinear dimensionality reduction effectively organizes phenotypic diversity and reveals failure modes, it lacks interpretability: the approach offers no direct path for predicting outcomes from molecular inputs, and the developmental parameters that determine a colony's position in morphospace remain obscured. FATE-MAP addresses this limitation through a two-pronged modeling strategy. First, we developed a transformer-based neural network that predicts morphological phenotypes directly from chemical structure. This model enables in silico forecasting of developmental risk for untested compounds and provides a scalable, prospective approach to identify human-relevant teratogens based on molecular features alone. However, this chemical structure-to-phenotype model operates as a black box; it maps inputs to outputs without explicitly modeling the developmental processes that generate phenotypes. To provide mechanistic insight, we developed complementary partial differential equation (PDE)-based models of morphogen diffusion and cell fate transitions that simulate patterning outcomes from first principles.

To unify these predictive and mechanistic approaches, we trained a neural network to embed simulated phenotypes from our mechanistic model and deep learning model into a shared, experimentally defined morphospace. This integrated framework allows direct comparison between model predictions and observed outcomes, enabling both validation of predictions and mechanistic interpretation of developmental failure modes. Together, these complementary strategies demonstrate the power of combining chemical structure-based modeling with biologically grounded simulations to enhance predictive accuracy and reveal the underlying rules of developmental fate patterning.

## Results

### A high-throughput platform for 2D gastruloid phenotyping

We sought to leverage the 2D gastruloid model in a high-throughput screen to map the morphospace of human gastrulation based on prior demonstrations that the model mimics the cell fate decisions of human gastrulation and produces well-defined spatial regions of differentiation that correspond to cell fate specification of the primitive streak (Fig. 1a, b). We chose to perturb BMP4-initiated gastruloids with compounds from a library of 210 drugs annotated for their activity against stem cell signaling pathways (Fig. 1c). While it is possible that a given drug may have multiple protein targets, our approach is target-agnostic and only requires a large diversity of perturbations to capture the variation in outcomes. To quantify phenotypes, we chose to use immunofluorescence staining of germ layer markers: GATA3 for amniotic ectoderm, Brachyury (BRA) for mesoderm, and SOX2 for the undifferentiated epiblast. We imaged ~10 colonies per drug condition and used a custom image segmentation algorithm to identify the levels of each cell fate marker in every nucleus of each colony (Supplementary Fig. 1a), collecting both cell fate and location for ~2 million cells across 2025 colonies.

We leveraged the fact that colonies are generally radially-symmetric to build a data structure for downstream pattern analysis. Inspired by approaches that vectorize animal behavior[30], we reasoned that we could meaningfully compress colony morphology through averaging cell fates over a set of 50 concentric annular bins, each ~5 μm in width, ranked by their position, from the edge to the center of each colony (Fig. 1d). This yielded a 150-dimensional vector for each colony containing the average azimuthal signal for GATA3, BRA, and SOX2, as a function of colony position (Fig. 1e). Although this embedding does not explicitly preserve geometric distances between adjacent bins, it preserves the exact radial ordering, enabling spatially coherent comparison of radial expression patterns across colonies (Supplementary Fig. 1b-c). As controls, we included several BMP4-only and untreated (no-BMP4) colonies from multiple independent wells, as well as a Wnt-activating (CHIR-98014) positive control which, as expected, differentiated the entire colony into mesoderm (Supplementary Fig. 1d). These controls demonstrated that patterning phenotypes were reproducible across multiple wells and days of running the assay (Supplementary Fig. 1e). Across the broader compound screen, plate-to-plate variability was minimal compared to drug-induced effects (Supplementary Fig. 1f), confirming that the 2D gastruloid model analyzed in this way meets the criteria for constructing a morphospace map.

### Human gastruloid morphospace

Upon visual inspection of the drug-treated gastruloids, we noticed a variety of deviations from the canonical patterning observed in the controls. We sought to develop an unsupervised algorithm that would (1) project each colony down to two dimensions, and (2) identify

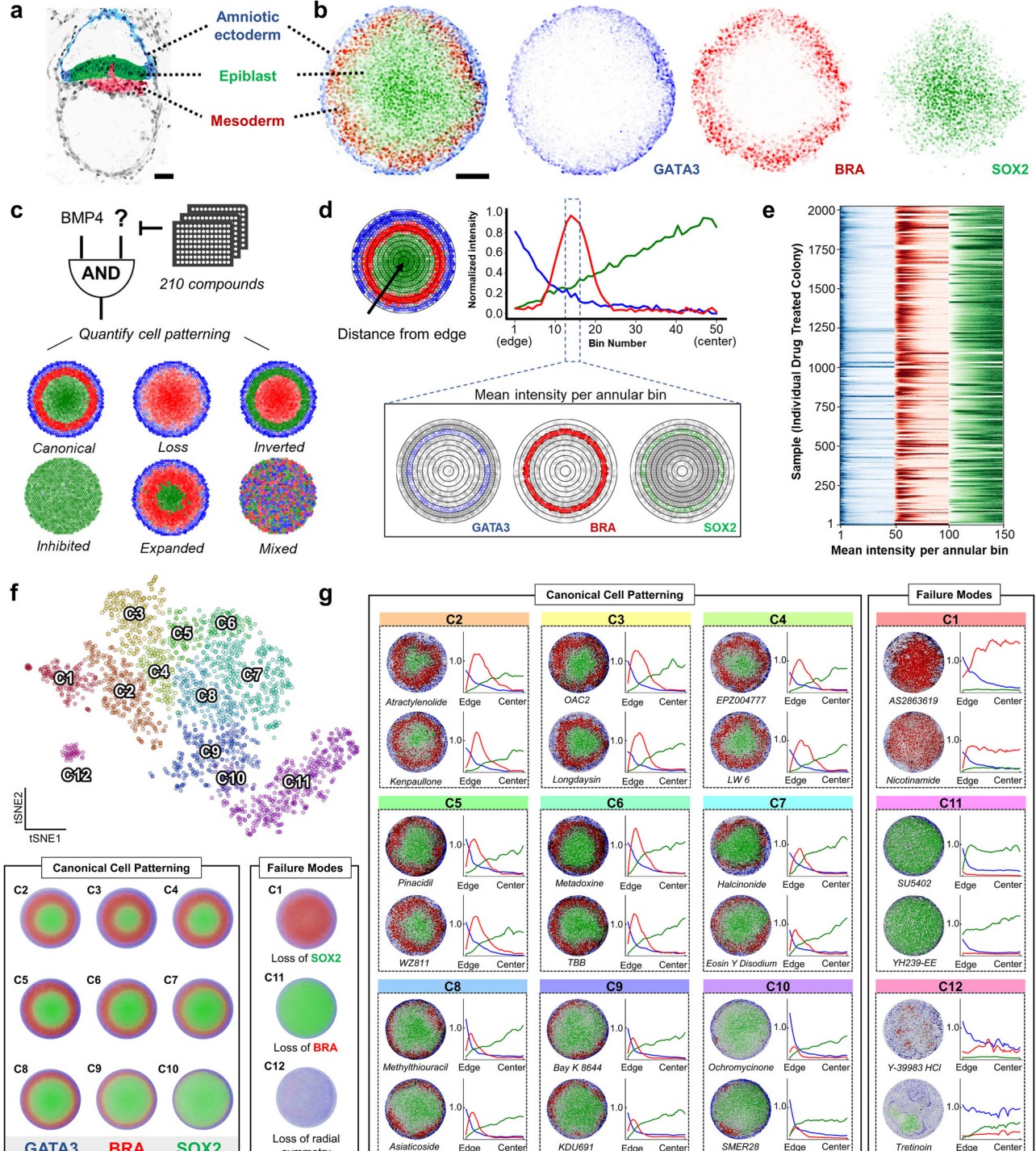

**Fig. 1 | FATE-MAP framework for quantitative characterization and perturbation of morphological phenotypes in 2D gastruloids. a** Schematic of human embryo showing cell fates observed during gastrulation: amniotic ectoderm (blue), undifferentiated epiblast (green), and mesoderm (red). The schematic is based on Carnegie Embryo #7801 from the Virtual Human Embryo (VHE) project (https://www.ehd.org/virtual-human-embryo/). **b** Representative BMP4-treated 2D gastruloid with immunofluorescence staining for GATA3 (amniotic ectoderm, blue), BRA (mesoderm, red), and SOX2 (embryonic disk, green). **c** Schematic showing the combinatorial drug screening approach used to perturb gastruloid patterning. While not all phenotypes are observed in our dataset, the schematic outlines potential perturbation modes, including canonical, loss, inverted, inhibited,

expanded, and mixed patterns. **d** Approach for quantifying cell patterning in 2D gastruloids by segmenting colonies into 50 concentric annular bins from edge to center; intensity per bin reflects average nuclear signal. Line plot shows mean GATA3, BRA, and SOX2 profiles. **e** Heatmap representing the 150-dimensional vectorized morphological features of colonies. **f** t-SNE embedding of ~2000 gastruloids reveals 12 distinct morphological clusters, including canonical patterns and failure modes characterized by loss of SOX2, BRA, or radial symmetry. Mean cluster images, shown below, highlight shared features of each cluster. **g** Representative drug-treated colonies from each cluster, shown with corresponding normalized radial fate marker distribution intensities reported in arbitrary units. Scale bars: (**a**) 50 μm, (**b**) 100 μm.

groups of perturbations that result in similar phenotypes so that they could be evaluated as potential failure modes. Inspired by previous methods[31], we coupled t-SNE to an unsupervised clustering procedure which applies watershed segmentation to the two-dimensional embedding after convolution with a Gaussian kernel (Supplementary Fig. 1g; Methods). This method does not impose a fixed number of clusters, as other clustering algorithms do (k-means, gaussian mixture models, spectral clustering, etc.), and thus produces clusters that reflect the local variations in phenotype without supervision (Fig. 1f, top). We identified 12 clusters (C1-C12) which we call morphological phenotypes.

To visualize the phenotype of each cluster, we generated composite images by taking the mean pixel intensity for each cell fate marker from all colonies in each cluster (Fig. 1f, bottom). This allowed us to build a coarse-grained understanding of colony patterning across the entirety of morphospace. Clusters could be divided into two major categories. The canonical patterning region (C2–C10) contains colonies that retain all three fate markers in concentric radial order (SOX2 central, BRA intermediate, GATA3 peripheral) and differ primarily in the relative sizes of these domains. BMP4-only controls map exclusively within this region (C2, C3, C4, C5, C8, C9; Supplementary Fig. 2a). In contrast, clusters C1, C11, and C12 lie outside the canonical region and exhibit either the loss of a required cell-fate marker (C1: loss of SOX2; C11: loss of BRA) and/or loss of radial symmetry (C12). Since either of the aberrations captured in these clusters would result in a failure of gastrulation, we classify these clusters as failure modes.

Because failure modes can arise from either differentiation failure or compound-induced cytotoxicity, we quantified total nuclear counts for every colony and identified a small subset of drugs that reduce viable cell number below 50% of the mTeSR-only control (Supplementary Fig. 3). This analysis separates genuine patterning failures from cytotoxicity-driven collapse within the failure-mode regions. Representative drug treatments from each cluster are shown in Fig. 1g. For more examples, see Supplementary Fig. 2b, and for the entire dataset, visit (https://max-wilson.mcdb.ucsb.edu/research/gastruloid-morphospace).

### Predicting morphological phenotype from chemical structure

Having defined a morphospace of human gastrulation and identified discrete failure modes, we next sought to develop a predictive model capable of mapping a chemical structure directly to morphological phenotype and morphospace location. To do so, we designed a two-step modeling framework in which chemical structure is first converted into a predicted phenotypic vector, and then embedded into the low-dimensional morphospace (Fig. 2a). Each step is performed by a separately trained neural network.

To build the chemical structure-to-phenotype model (Fig. 2a, left), we began by embedding each compound's SMILES representation using ChemBERTa, a pretrained transformer model trained on -10 million PubChem compounds[32]. This yielded a 768-dimensional molecular feature vector for each drug, $\vec{x}_{768}$, which we used as input to a neural network, $\phi(\vec{x})$, that was trained to predict the 150-dimensional phenotype vector of the gastruloid (Methods). Ten percent of the compounds were randomly held out as a blinded test set to assess generalization.

To map phenotype vectors into morphospace, we trained a second neural network to approximate the relationship between the 150-dimensional phenotype profile, $\vec{y}_{150}$, and its corresponding t-SNE coordinates (t-SNE$_1$, t-SNE$_2$). While t-SNE lacks a formal inverse function, neural networks can be used to learn the embedding (i.e., parametric t-SNE). We trained a separate neural network to minimize the distance between predicted and true t-SNE coordinates based on experimental data (Fig. 2a, right; Methods). The resulting network, $\psi(\vec{y})$, accurately reconstructed the original experimental

morphospace and was used for all subsequent projections (Supplementary Fig. 2c).

The combined model accurately predicts 150-dimensional phenotype vectors and corresponding morphospace positions for both the training and blinded compounds, capturing canonical and failure-mode clusters. Across 188 training compounds, the mean RMSE was 0.17, with a mean cosine similarity of 0.90. Importantly, on average 74% of features per compound fell within one standard deviation of the empirical replicate distribution, indicating that most predicted feature values lie inside the observed biological variability. On the blinded set ($n = 21$), performance remained strong (mean RMSE = 0.24; mean cosine similarity = 0.85), with 54% of features per compound within one standard deviation of replicate variation. Aggregating all 209 compounds, the overall means were RMSE = 0.17, cosine similarity = 0.90, and 72% within one standard deviation. Representative predictions for all 12 clusters are shown in Fig. 2b, including both phenotype profiles and projected morphospace coordinates.

Model performance was maintained despite considerable chemical dissimilarity between the blind and training compounds (Supplementary Fig. 4a–b), indicating robust generalization to structurally diverse molecules. To explore this further, we clustered all experimentally screened compounds based on structural similarity and compared these groupings to morphology-defined phenotypic clusters (Supplementary Fig. 5). Despite the presence of chemically coherent scaffold-based clusters, we observed minimal spatial overlap between structure- and phenotype-defined groupings. This suggests that even compounds with similar scaffolds can produce distinct developmental effects, reinforcing the need for neural network–based approaches to uncover structure–function relationships that scaffold similarity alone may obscure.

Because the most consequential classification task is distinguishing failure-mode phenotypes from canonical patterning outcomes, we evaluated performance at two levels: (i) strict 12-cluster classification, which requires assigning each compound to the correct morphospace cluster, and (ii) binary failure-mode discrimination, treating the task as failure-mode vs. canonical patterning classification. Across the 50 compounds whose experimental centroids fall in failure-mode clusters, the model predicted all 50 to reside within the failure-mode region, yielding a binary failure-mode sensitivity of 1.00. Within this region, strict cluster identity was correct for 39 of 50 compounds (78%), with most misassignments reflecting shifts toward C11, the densest failure-mode attractor in morphospace. Overall binary classification accuracy was 0.71 (0.72 in the training set and 0.67 in the blind set). These results indicate that the model is highly reliable for detecting whether a compound induces a failure-mode phenotype, even if the precise sub-cluster label within the failure-mode region is occasionally uncertain. Of note, drugs were assigned to clusters not by visual inspection of their position in morphospace, which can be ambiguous, but using the same watershed-derived morphospace boundaries defined during the original clustering, ensuring consistent, data-driven classification across experimental and predicted phenotypes (see Supplementary Note 1 for details on the morphospace classification boundary and thresholding approach).

### FATE-MAP predicts teratogenic risk from chemical structure

To evaluate the translational potential of FATE-MAP for predicting teratogenicity, we first examined drugs from our initial screen with known pregnancy-related risks. Several established teratogens[33–35], including isotretinoin, TTNPB, and bexarotene, were experimentally determined to fall within cluster 11 and shared a common morphological phenotype characterized by loss of mesodermal identity (BRA expression) (Fig. 3a). These colonies frequently displayed lighter central regions that reflect vertical protrusions caused by increased cell density under micropattern confinement (Supplementary Fig. 6). The observation that known teratogens induce failure-mode

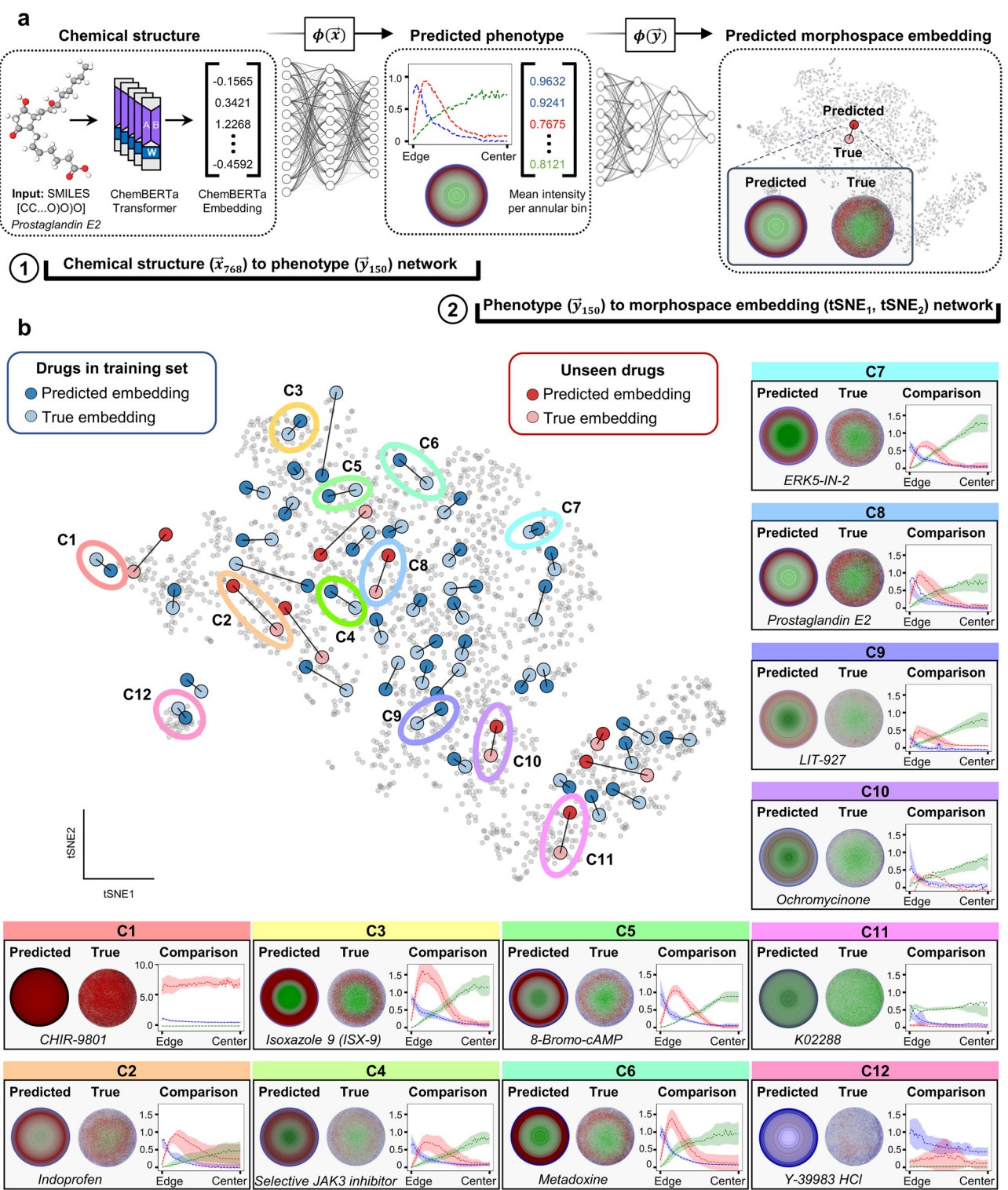

**Fig. 2 | Deep learning framework for predicting gastruloid phenotype and morphospace location from chemical structure. a** Schematic of the FATE-MAP pipeline. SMILES strings are embedded using a pretrained ChemBERTa transformer and passed to a neural network trained to predict spatial phenotypes (150-dimensional vector of fate marker intensities across radial bins). Predicted phenotypes are then embedded into morphospace using a second neural network trained to approximate the original t-SNE projection. **b** Predicted versus true morphospace embeddings for all compounds, colored by training (blue) or held-out (red) sets. Lines connect predicted and true positions, illustrating model accuracy. Representative predictions from each cluster are highlighted, including training and blind compounds. Each set shows the predicted and true phenotypes as radial intensity plots reported in arbitrary units and corresponding colony images. Predicted images are colorized schematics representing the spatial distribution of GATA3 (blue), BRA (red), and SOX2 (green) derived from the model-predicted phenotype vector. Radial intensity plots show the model-predicted profiles (dashed lines) overlaid on shaded bands representing the mean ± standard deviation (SD) of experimentally measured marker intensities across biological replicates at each radial bin.

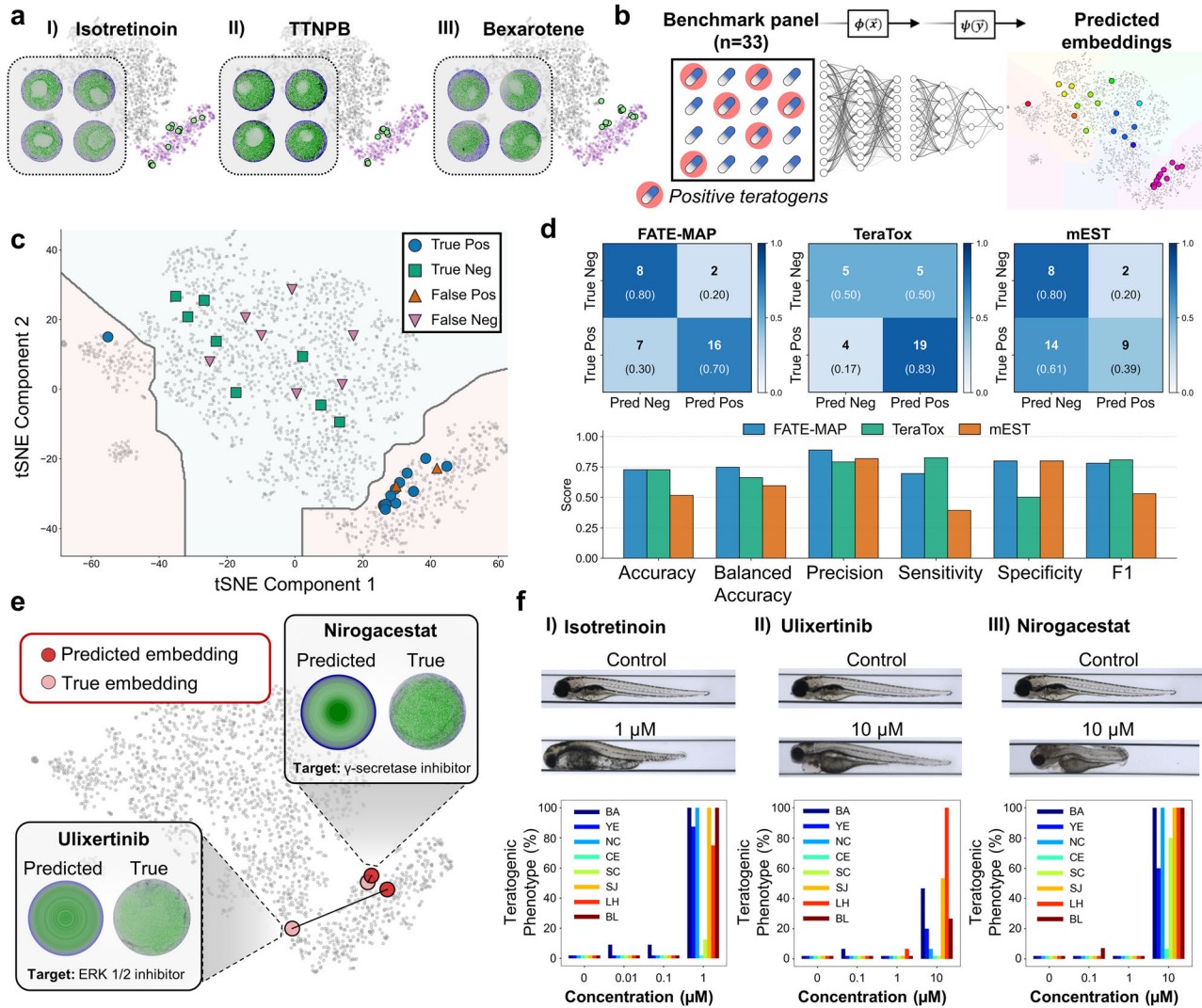

**Fig. 3 | FATE-MAP predicts teratogenic risk from chemical structure and identifies novel human-relevant teratogens. a** Representative 2D gastruloids treated with known teratogens isotretinoin, TTNPB, and bexarotene, all converging on cluster 11 and exhibiting a shared failure mode marked by BRA loss. **b** A benchmark panel of 33 compounds with known teratogenicity outcomes, previously evaluated in stem cell-based assays of developmental toxicity (TeraTox and mEST), was used to assess predictive performance. Predicted phenotypes were embedded into morphospace and classified using the fixed cluster boundaries derived from the experimental dataset. **c** Morphospace-based classification of the benchmark set: embeddings falling within failure-mode clusters (pink region; C1, C11, C12) are assigned positive teratogen status, whereas embeddings within the canonical patterning region (light blue region; C2–C10) are assigned non-teratogenic status. Points are labeled as true positive, true negative, false positive, or false negative based on known teratogenic risk. **d** Confusion matrices and comparative performance metrics for FATE-MAP, TeraTox, and mEST on the benchmark panel. **e** Two blinded compounds without prior human pregnancy data (ulixertinib and nirogacestat) were predicted and experimentally confirmed to fall within cluster 11. **f** Zebrafish teratogenicity assays show that both compounds, like the positive control isotretinoin, induce qualitative and quantitative defects consistent with failure modes in embryonic development, including abnormal body curvature (BA), yolk edema YE, necrosis NC, craniofacial edema CE, scoliosis SC, snout and jaw defects SJ, reduced lateral heart area LH, and decreased lateral body length BL, as indicated by the color-coded bars.

phenotypes suggests that morphospace location can be leveraged to identify additional high-risk compounds directly from chemical structure.

To benchmark this structure-to-phenotype framework against established teratogenicity assays, we assembled a curated panel of 33 FDA-approved drugs with well-characterized teratogenicity outcomes (positive or negative), supported by human clinical data (as reported in FDA labeling) and/or in vivo embryo–fetal development (EFD) studies in rats or rabbits, and for which results from both the TeraTox iPSC multilineage differentiation assay and the mouse embryonic stem cell test (mEST) have been reported[36]. For each compound, FATE-MAP first generated a 150-dimensional predicted phenotype vector from its chemical structure and then projected this vector into the

experimentally defined morphospace using the neural network approximation of the t-SNE embedding (Fig. 3b).

Teratogenicity was inferred directly from morphospace location using the same cluster boundaries applied to experimental data: compounds whose predicted embeddings fell within the failure-mode region (clusters C1, C11, and C12) were classified as teratogenic, while those mapping to canonical patterning clusters (C2–C10) were classified as non-teratogenic (Fig. 3c). Notably, only 4 of the 33 benchmarking compounds were included in our experimental dataset. Across the full 33-drug panel, FATE-MAP achieved an overall accuracy of 72.7%, outperforming the mEST assay (51.5%) and matching the TeraTox assay (72.7%), while exhibiting higher balanced accuracy (74.8% for FATE-MAP vs. 66.3% for TeraTox)

(Fig. 3d, and Supplementary Data 1). Because FATE-MAP assigns phenotypes to clusters using fixed morphospace boundaries derived from experimental data, classification does not require re-testing each compound, enabling scalable, fully in silico teratogenicity prediction.

To further assess predictive performance relative to existing in silico developmental toxicity models, we evaluated a set of 20 compounds held out from training across three models: FATE-MAP, the CAESAR developmental toxicity predictor, and the Procter & Gamble (P&G) teratogenicity model. For CAESAR and P&G, we used the reported model outputs provided in their published evaluations[36], ensuring a direct comparison without retraining or re-tuning either model. Across this compound set, FATE-MAP achieved an accuracy of 75.0%, matching the CAESAR model (75.0%) and substantially outperforming the P&G model (35.0%), while also demonstrating the highest balanced accuracy (85.7% for FATE-MAP vs. 81.2% for CAESAR) (Supplementary Fig. 7a, b; and Supplementary Data 2). Together, these results show that FATE-MAP captures developmental toxicity signals that generalize beyond the training distribution and performs competitively with both experimental and computational state-of-the-art approaches.

To evaluate FATE-MAP's ability to prospectively identify teratogens, we selected two clinical-stage compounds, ulixertinib (an ERK1/2 inhibitor) and nirogacestat (a γ-secretase inhibitor), from our initial screen. These compounds were both experimentally confirmed to reside in cluster 11 (Supplementary Fig. 8a, b) and had no established developmental risk data in humans. Importantly, they were also excluded from model training, yet were still accurately predicted to fall within cluster 11 based solely on chemical structure (Fig. 3e). Notably, the predicted failure-mode phenotypes for both compounds are consistent with the known essential roles of ERK and Notch signaling in embryogenesis, where disruption of either pathway leads to severe developmental defects and embryo–fetal loss in animal models[37,38]. While direct validation in human embryos is not possible, we applied a zebrafish-based developmental toxicity assay as an independent in vivo model to test these predictions, alongside isotretinoin as a positive control. Zebrafish are widely used in pharmaceutical developmental toxicology pipelines and offer higher-throughput, cost-effective testing compared to traditional mammalian models. Prior studies have shown that zebrafish assays achieve comparable or superior predictive performance to mouse-based testing (74.19% accuracy and 87.50% sensitivity for zebrafish versus 67.74% accuracy and 75.00% sensitivity for mice)[39]. At sub-micromolar to micromolar doses, all three compounds induced characteristic teratogenic phenotypes, including craniofacial defects, body axis shortening, and yolk sac edema (Fig. 3f, and Supplementary Fig. 8c–h). Although ulixertinib and nirogacestat produced the expected posterior truncation in zebrafish, axial tissues such as somites and notochord remained present, consistent with partial, rather than complete, disruption of primitive streak–derived mesoderm and reflecting species-specific differences in pathway sensitivity. Overall, these results confirm that FATE-MAP can prospectively identify previously unrecognized teratogens, offering a scalable in silico alternative to in vivo developmental toxicity testing

## Cell density shapes mesoderm positioning and fate
We next turned our attention to determine the causes of variance in the canonical patterning region. From previous observations on the importance of 2D gastruloid density in determining patterns[28,40–42], we hypothesized that mesoderm band width could be regulated by cell density. Because cell seeding produces inherent variation in cell density across colonies, BMP4-treated controls provide a natural spectrum for analyzing density-dependent effects. Indeed, the distribution of these control colonies in morphospace is explained by their density (Supplementary Fig. 9a), with a moderate correlation between density

and mesoderm peak position ($R^2 = 0.69$). Ranking colonies by density further revealed a clear expansion of mesoderm toward the colony center as density decreased (Supplementary Fig. 9b). This trend also held across cluster averages: lower-density clusters showed more central mesoderm expression, with a stronger correlation ($R^2 = 0.96$) when plotted by cluster mean (Supplementary Fig. 9c). Ranking clusters by edge-to-center BRA profiles reinforced this relationship (Supplementary Fig. 9d), indicating that cell density explains a major source of phenotypic variance.

Given the impact of cell density on patterning, we hypothesized that it might be strong enough to override the effects of drug treatment. If correct, we would expect colonies to be rescued back into the canonical patterning region by increasing density or driven into failure modes by decreasing density. Supporting this, we observed (-)-Blebbistatin-treated colonies, typically found in failure mode clusters C1 and C12, regained canonical phenotypes at higher densities (Supplementary Fig. 9e, left). Conversely, DEL-22379–treated colonies, normally found at the edge of the canonical patterning region, were pushed into failure mode C1 at lower densities (Supplementary Fig. 9e, right). These results suggest that density is not only correlated with but also causal in shaping gastruloid patterning outcomes, and may serve as a target for rescuing disrupted developmental trajectories.

## Modeling density-dependent morphogen signaling dynamics
Various studies have developed models of the effects of individual morphogens on cell fates[43–45]. While combinatorial morphogen approaches have recently been applied to model cell fate decisions in non-patterned stem cells[46] and 2D gastruloids[47], we lack a comprehensive model that incorporates combinatorial signaling dynamics and how those dynamics are decoded into cell fates. Such a model would enable an unbiased analysis of patterning outcomes in the gastruloid based on their fate patterning. Building off of previous PDE models of morphogen signaling[47], we utilized a reaction-diffusion description of BMP, Wnt, and Nodal dynamics (Fig. 4a, Methods). To model the fate choices, we incorporated equations that describe how morphogens are decoded into each of the three cell fate markers, GATA3, BRA, and SOX2 (Fig. 4b, and Supplementary Fig. 10a, Methods). We note that existing models of cell fate in the 2D gastruloid incorporate entire signaling histories but only focus on a single morphogen[43], or incorporate multiple morphogens but neglect dynamics and make heuristic predictions based on instantaneous morphogen levels[47].

Based on our observations of the importance of cell density, we sought to include its effects into our mechanistic model. Because hESCs form a tight monolayer, but also cannot grow beyond the bounds of the micropatterned ECM onto which they are plated, multiple morpho-mechanical features of cells in the gastruloid change as a function of density (e.g., aspect ratio, cortical tension, receptor availability, volume, etc.). We hypothesized that Wnt signal secretion, with its reported sensitivity to changes in local mechanics[48], is regulated by cell density. Mathematically, we captured this by introducing a function, $h(d)$, that linearly decreases all sources of Wnt ligand creation as density, $d$, increases. See Methods for details of model implementation and justification of parameter choices.

To test if we accurately captured the effects of density, we simulated morphogen dynamics and cell fate determination of three colonies that range in densities, from 750 cells (-10th percentile) to 1150 cells (-90th percentile). Kymographs of simulated signaling revealed a faster Wnt wave that further permeates the colony interior as density decreases (Fig. 4c). As a result, Nodal, which is created as a function of Wnt, also follows this trend. Ultimately, this leads to a deeper penetration of BRA into the center of the colony with a corresponding decrease in SOX2 because BRA is a function of both Wnt and Nodal (Fig. 4d, Supplementary Movie 1). Experimental colonies with similar

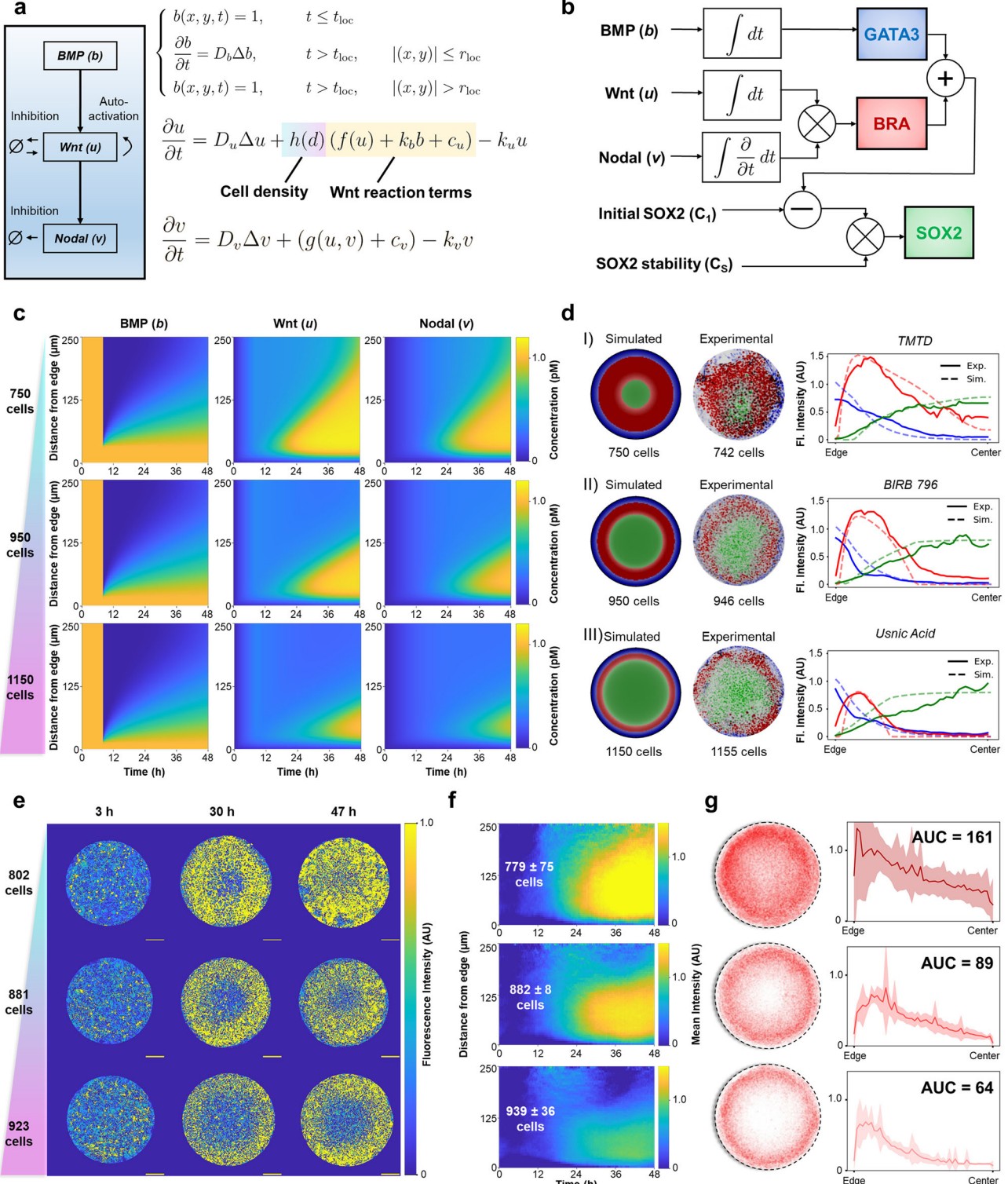

densities matched simulation results across a wide range of densities, even at the extremes of density and complete inhibition of a morphogen signal (Supplementary Fig. 10b, c). These findings suggest that our comprehensive model accurately captures the interactions between morphogen signaling, cell density, and cell fate markers.

A central implication of our model is that Wnt signaling is inversely proportional to cell density. To test this prediction, we sought to experimentally measure endogenous Wnt signaling in 2D gastruloids as a function of cell density. We leveraged a CRISPR-tagged β-Catenin (β-cat) hESC line[49] (Methods) to observe Wnt signaling in real time. We

varied initial seeding density and took live-cell, time-lapse images of 8 colonies per condition over the 48 hr differentiation (Fig. 4e). To extract the β-cat signaling activity we wrote a custom image processing algorithm that extracts the active, non-membrane β-cat from the inactive, membrane-bound pool, as has been done previously[47] (Methods; Supplementary Fig. 10d). Both timelapse videos from individual colonies (Supplementary Fig. 10e–g, and Supplementary Movie 2) and mean kymographs, subset by colony density, clearly demonstrate the impact of cell density on the Wnt wave (Fig. 4f). While the model captures density-dependent Wnt wave propagation,

**Fig. 4 | Modeling the impact of cell density on morphogen dynamics and cell fate determination in 2D gastruloids. a** Reaction-diffusion model describing the density-dependent spread of morphogens BMP, Wnt, and Nodal in 2D gastruloids. **b** Logic-based schematic diagram of cell fate determination rules based on integrated morphogen dynamics. Notably, cells respond to the integral of BMP and Wnt (concentration-dependent), but to the change in Nodal concentration (rate-dependent). Initial SOX2 levels ($C_1$) and SOX2 stability ($C_s$) modulate the response of cells to these signals, influencing differentiation outcomes. **c** Simulated concentration profiles of BMP, Wnt, and Nodal over time at varying cell densities (750, 950, and 1150 cells). The simulations reveal that lower cell densities lead to faster and deeper penetration of Wnt and Nodal signals into the colony. **d** Simulated versus experimental cell fate outcomes for different densities, demonstrating the model's ability to replicate patterns observed in our screen. Three examples of

simulations (left) compared to experimental images (right) with accompanying line plots show matching spatial distributions of cell fate markers (GATA3, BRA, SOX2). **e** Experimental investigation of density-dependent Wnt dynamics using CRISPR-tagged β-catenin (β-cat) hESC lines. Live-cell time-lapse imaging over 48 h shows the effect of cell density on Wnt signaling. Scale bars = 100 μm. **f** Mean kymographs of Wnt signaling activity across 8 colonies per cell density range, demonstrating that lower cell densities result in a faster and broader spread of the Wnt signal. **g** Quantification of BRA staining intensity profiles across different densities, taken from the same colonies used to generate the kymographs. Line plots represent the mean BRA intensity ± SD across colonies at each radial bin. The area under the curve (AUC) analysis confirms increased total BRA signal at lower densities, validating our model's prediction of density-induced variations in mesoderm dynamics. All fluorescence intensities are reported in arbitrary units (AU).

absolute signaling amplitudes differ from experiment, reflecting subtle differences in effective signaling timescales between simulation and experiment, as well as technical limitations in fully isolating active β-catenin. Finally, we fixed and stained the same colonies for which β-cat was imaged and confirmed that their BRA staining also decreased with increasing density (Fig. 4g). Overall, these experiments validate our choice of model architecture, confirm the effects of cell density on Wnt signaling, and suggest that our cell fate rules accurately capture the decoding process that occurs in the gastruloid.

## Parameterizing gastruloid morphospace

We previously trained a neural network $\psi(\vec{y})$ to approximate the original t-SNE embedding from the 150-dimensional phenotype vector. We now leverage this function to assign interpretability to morphospace by projecting simulated gastruloids with known developmental parameters into the same space. This allows us to understand how perturbations move colonies through morphospace, enabling mechanistic interpretation of drug action. Importantly, this analysis is independent of chemical structure and does not rely on the structure-to-phenotype model used elsewhere in the study.

To begin, we simulated gastruloids across a grid of two key parameters: cell density and SOX2 stability (Fig. 5a). SOX2 stability is a model parameter that reflects the resistance of SOX2 expression to mesodermal signals and captures how GATA and BRA modulate epiblast identity. Although not directly measurable, this parameter provides a mechanistically interpretable readout that broadly reflects drug mechanism of action and influences a colony's position within morphospace. We projected all simulated colonies into morphospace using $\psi(\vec{y})$ and found that these projected phenotypes formed a smooth grid spanning the canonical patterning region (Fig. 5b–e). The resulting structure reveals two interpretable axes aligned with these parameters, while not precluding additional contributing factors. To assess whether other coefficients could generate comparable trajectories, we systematically swept all model parameters over biologically plausible ranges; only density and SOX2 stability consistently produced extended, coherent axes in morphospace (Supplementary Figs. 11–12). Importantly, this organization persists across different t-SNE initializations, confirming that the parameter axes are not artifacts of a particular embedding (Supplementary Fig. 13).

We next asked whether this structure could be used to predict experimental outcomes directly from developmental parameters. To do so, we fit polynomials to the simulated grid to define density and SOX2 stability axes, then interpolated the SOX2 stability values for each drug-treated gastruloid from the location of its experimental centroid. We paired this interpolated value with the experimentally measured mean density of each drug treatment and used them as inputs to our morphogen model to simulate colony patterning from first principles. The predicted colonies were then projected into morphospace using $\psi(\vec{y})$ and compared to the original experimental centroids (Fig. 5f). Notably, the simulated colonies closely matched

both the morphospace locations and spatial phenotypes of the drug-treated gastruloids, validating the accuracy of the underlying model.

This approach demonstrates that the majority of phenotypic variance in the canonical patterning region can be described by just two parameters: cell density and SOX2 stability. By projecting simulated outcomes onto the same manifold as experimental data, we not only assign interpretable axes to morphospace, but also validate the predictive power of our mechanistic model, providing a framework to uncover how specific perturbations reshape developmental trajectories.

## Linking black-box and mechanistic models in development

While our AI-based structure-to-phenotype model predicts 2D gastruloid phenotypes with high accuracy (Fig. 6a), it functions as a black box and does not directly reveal which developmental processes are perturbed. To provide mechanistic insight, we developed a complementary model grounded in morphogen diffusion and fate transitions (Fig. 6b). By projecting simulated phenotypes into the same learned manifold as experimental data, this model assigns interpretable structure to morphospace and enables causal reasoning about how perturbations modulate underlying parameters. In practice, the mechanistic model codifies the rules that govern the structured, canonical region of morphospace and, when supplied with prior mechanistic hypotheses (e.g., morphogen inhibition; Supplementary Fig. 10c), can also reproduce specific failure modes. The AI model, in turn, predicts where a compound will land in morphospace directly from chemical structure. Together, the two are complementary: the AI localizes compound effects, and the mechanistic model interprets whether they reflect canonical displacements or topology-altering failures.

Quantitatively, the black-box model (chemical structure → black box → phenotype) and the mechanistic model (initial conditions → PDE's of morphogen spread and cell fate transition → phenotype) generated predictions with similar accuracies (Black-box model: mean RMSE = 0.17, mean cosine similarity = 0.90, 72% of predicted features within 1 standard deviation of replicate variation; Mechanistic model: mean RMSE = 0.18, mean cosine similarity = 0.93, 50% of predicted features within 1 standard deviation of replicate variation; Supplementary Data 3). However, the mechanistic model allowed us to predict which molecules were affecting Wnt signaling and cell density versus which molecules were affecting SOX2 stability.

Even deviations between mechanistic model predictions and experimental results revealed valuable insights. The morphogen spread model assumes fixed rules for BMP, Wnt, and Nodal signaling. However, certain compounds appear to override these assumptions. For example, isoxazole 9 (ISX-9)–treated colonies had relatively high cell counts (mean = 1020 cells), which would predict moderate BRA expression in simulation; experimentally, however, they exhibited strong BRA expression (Supplementary Fig. 14a). This suggests an additional mechanism at play; indeed, isoxazole 9 has been shown to

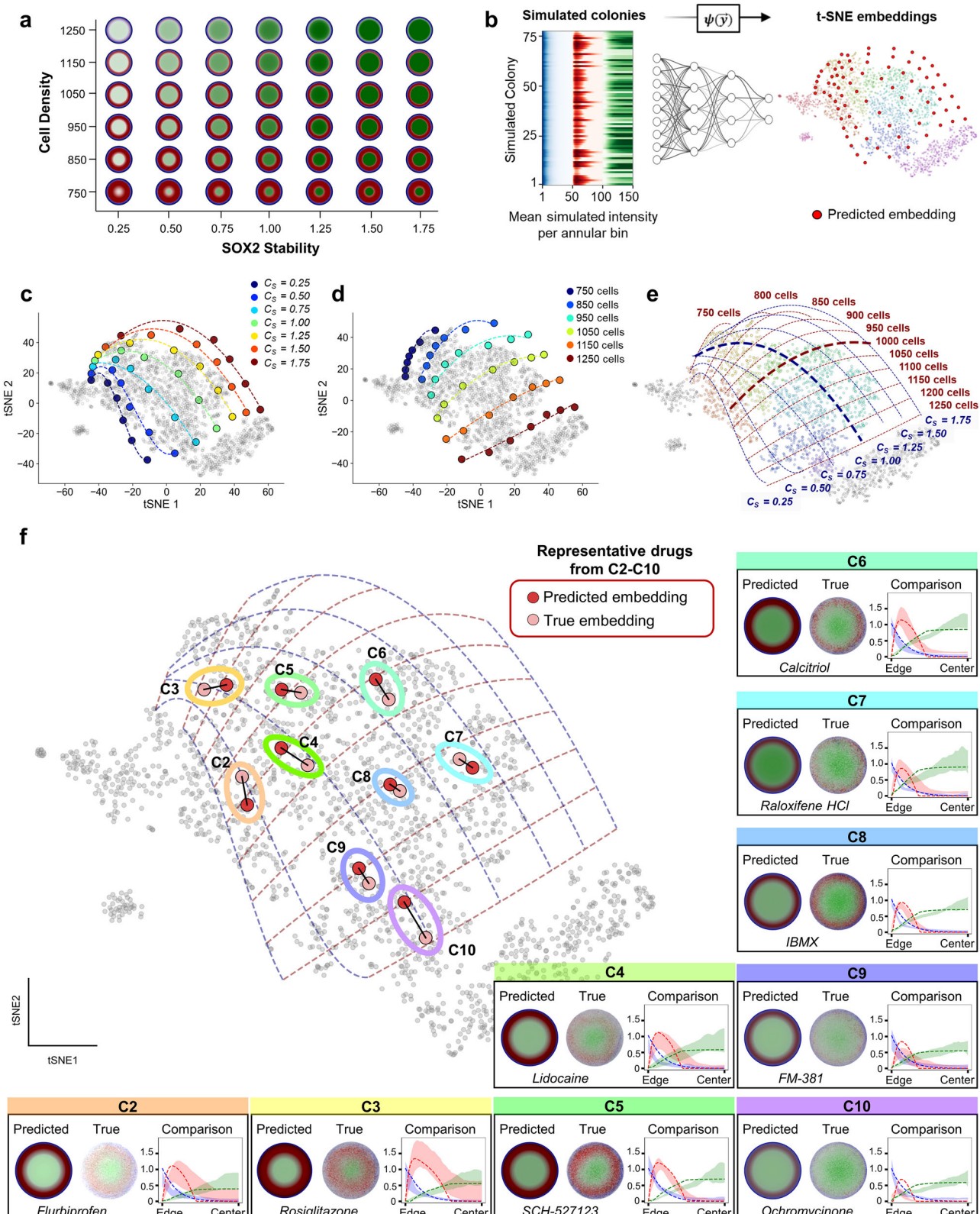

activate Wnt/β-catenin signaling by covalently binding Axin1 and stabilizing β-catenin[50], thereby enhancing mesodermal gene expression. Conversely, longdaysin-treated colonies had lower cell counts (mean = 892 cells), which would predict high BRA expression, but showed reduced BRA experimentally (Supplementary Fig. 14b), consistent with its reported Wnt-inhibitory action via disruption of LRP6 phosphorylation and reduced nuclear β-catenin accumulation[51]. These

discrepancies reveal when a perturbation alters the fundamental rules of morphogen signaling, enabling the model to identify biologically meaningful outliers rather than merely recapitulating the input.

These findings highlight an additional utility of the morphogen spread model as a tool for inferring mechanisms of action and identifying pathway-level effects of small molecules, offering a mechanistic

**Fig. 5 | Parameterizing gastruloid morphospace with developmental model parameters. a** Simulated gastruloid phenotypes spanning a grid of cell density and SOX2 stability values, a model-derived parameter describing resistance to mesodermal cues. **b** Schematic showing projection of simulated colonies into morphospace using a neural network trained to approximate the t-SNE embedding. **c**, **d** Individual SOX2 stability (**c**) and cell density (**d**) contours projected into morphospace reveal that these parameters form orthogonal, continuous axes spanning the canonical patterning region. **e** Full grid of parameterized simulations, showing smooth and interpretable structure in morphospace. **f** Model validation:

for each drug treatment from clusters C2–C10, experimental cell density and interpolated SOX2 stability values were used to simulate gastruloid outcomes and project them into morphospace. Predicted embeddings (dark red) closely match the true experimental locations (light red) and phenotypes, confirming the predictive power of the model. Insets show representative predicted and experimental colonies for each cluster, with radial intensity plots displaying model predictions (dashed lines) overlaid on shaded regions representing the mean ± SD of experimentally measured marker intensities across biological replicates at each radial bin. All fluorescence intensities are reported in arbitrary units.

complement to traditional phenotype-based screening in drug discovery.

## Discussion

Our successful identification of teratogens using FATE-MAP demonstrates the potential of stem cell-based in silico approaches to predict developmental toxicity at scale. Unlike animal models, which are limited by species-specific differences and are difficult to scale due to the need for in vivo dosing and animal husbandry[52], FATE-MAP enables rapid prediction of teratogenic risk directly from chemical structure, without the need to experimentally screen each compound. Although previous efforts have used mouse embryoid bodies[53] and micropatterned hESCs[36,54–56] for teratogen testing, these approaches require new compounds to be tested in vitro and lack a framework for linking observed phenotypes to underlying developmental mechanisms. In parallel, monolayer-directed differentiation protocols have also been implemented for teratogen detection, achieving high classification accuracy from single-marker readouts[57]. While compelling in their simplicity, these assays lack morphological context and mechanistic resolution, underscoring the added value of FATE-MAP in resolving developmental failure modes.

With over 80% of FDA-approved drugs lacking sufficient fetal risk data[58] and an estimated 1 in 16 pregnancies exposed to teratogenic compounds[59], there is a critical need for scalable, human-relevant screening platforms. FATE-MAP provides a proof-of-concept foundation for meeting this need, particularly as millions of new chemical compounds are synthesized each year, adding to a global chemical space estimated to contain up to $10^{60}$ possible small molecules[60], of which only a small fraction have been tested in any developmental context[61].

One of the central advantages of FATE-MAP lies in its integration of AI-based prediction and mechanistic modeling within a unified framework. While deep learning approaches offer scalable, prospective prediction of phenotype directly from chemical structure, their black-box nature limits mechanistic interpretability. Conversely, mechanistic models of morphogen dynamics provide causal insight but are typically constrained by simplified parameter spaces and limited scalability. For example, current models of morphogen signaling have shifted from static gradients to dynamic, wave-like propagation[47,62], and cell fate decoding has evolved from simple threshold models to more complex integral-based decoding[43,63]; however, most existing computational approaches lack a means to link these dynamical processes to phenotypic outcomes. FATE-MAP addresses this limitation by combining a mechanistic reaction-diffusion model of morphogen dynamics with a learned embedding function, enabling simulated gastruloids to be projected into the same morphospace as experimental data. This integration facilitates the identification of key signaling parameters, such as cell density-modulated Wnt signaling and SOX2 stability, that help organize developmental trajectories, bridging the gap between molecular inputs and biological processes.

Previous studies have established the importance of density and SOX2 dynamics in fate control in human 2D gastruloids. Etoc et al. showed that colony density both modulates morphogen accessibility (receptor relocalization and NOGGIN restrict TGF-β signaling to the

colony edge) and regulates BRA patterning (as density rises, a BRA band emerges and shifts toward the edge)[40]. Camacho-Aguilar et al. showed that SOX2 must decline for WNT to drive mesoderm and linked SOX2 loss to BMP/SMAD4 dynamics[46], while Teague et al. demonstrated that SOX2 acts as an integrator of BMP/SMAD4 input, with its rate of decrease proportional to the signaling integral, highlighting its role as a memory of morphogen exposure[43]. Building on this foundation, we show that WNT signaling is inversely proportional to cell density and that cell density and SOX2 stability emerge as orthogonal, interpretable axes spanning the canonical morphospace. This framework not only corroborates earlier findings but also supports prospective mapping of developmental trajectories and separation of canonical variation from topology-breaking failure modes.

Unlike in the embryo, where cells grow in a 3D environment, micropatterned colonies are confined to a 2D plane, causing them to spread at low density and crowd at high density, conditions that alter membrane tension[64] and receptor localization[40]. Our data show that even subtle changes in initial seeding density can shift colonies between canonical patterning and failure-mode trajectories, raising the possibility that Wnt signaling functions as a mechano-sensitive density detector to ensure robust patterning despite variability in starting conditions. Mouse studies further support the existence of a density-sensing module: gastrulation is delayed in embryos with reduced cell numbers, while aggregated double-sized embryos contain more cells at early time points but still initiate gastrulation at the same time as normal-sized embryos[65–67]. Consistently, whole-embryo imaging has shown that density variance is minimized at the primitive streak[68], suggesting active regulation of local cell density during development. Recent work in engineered mouse 3D gastruloids has similarly used live-cell signaling reporters to dissect the timing and interplay of BMP, Wnt, and Nodal signaling during early axis formation[69], highlighting the utility of these tools across species and model systems. While it remains to be determined whether aberrant density sensing directly contributes to failures of human gastrulation, our study of the 2D gastruloid suggests that this module can respond to a wide range of fluctuations and modulate germ layer fate patterning accordingly, at least in part through changes in Wnt signaling.

As a model-derived parameter, SOX2 stability governs the balance between epiblast maintenance and differentiation, capturing whether SOX2 is downregulated in response to mesoderm-inducing cues. While not directly measurable, it likely reflects epigenetic or post-translational regulation: SOX2 degradation is influenced by phosphorylation and methylation[70]. In our dataset (Supplementary Data 4), compounds associated with increased SOX2 stability included valproic acid, a histone deacetylase (HDAC) inhibitor widely used to enhance pluripotency and reprogramming efficiency[71]. In contrast, ROCK inhibitors such as hydroxyfasudil and ripasudil were among the strongest SOX2 destabilizers. While these effects may not stem from direct SOX2 targeting, ROCK inhibition disrupts apical junctions and epithelial architecture, indirectly enhancing BMP signaling, which promotes differentiation[40,72]. Recent studies have further shown that SOX2 levels reconfigure Wnt/β-catenin binding, thereby controlling whether Wnt signals promote pluripotency or mesoderm[73]. Together, these findings reinforce the idea that SOX2 stability encodes a pluripotency

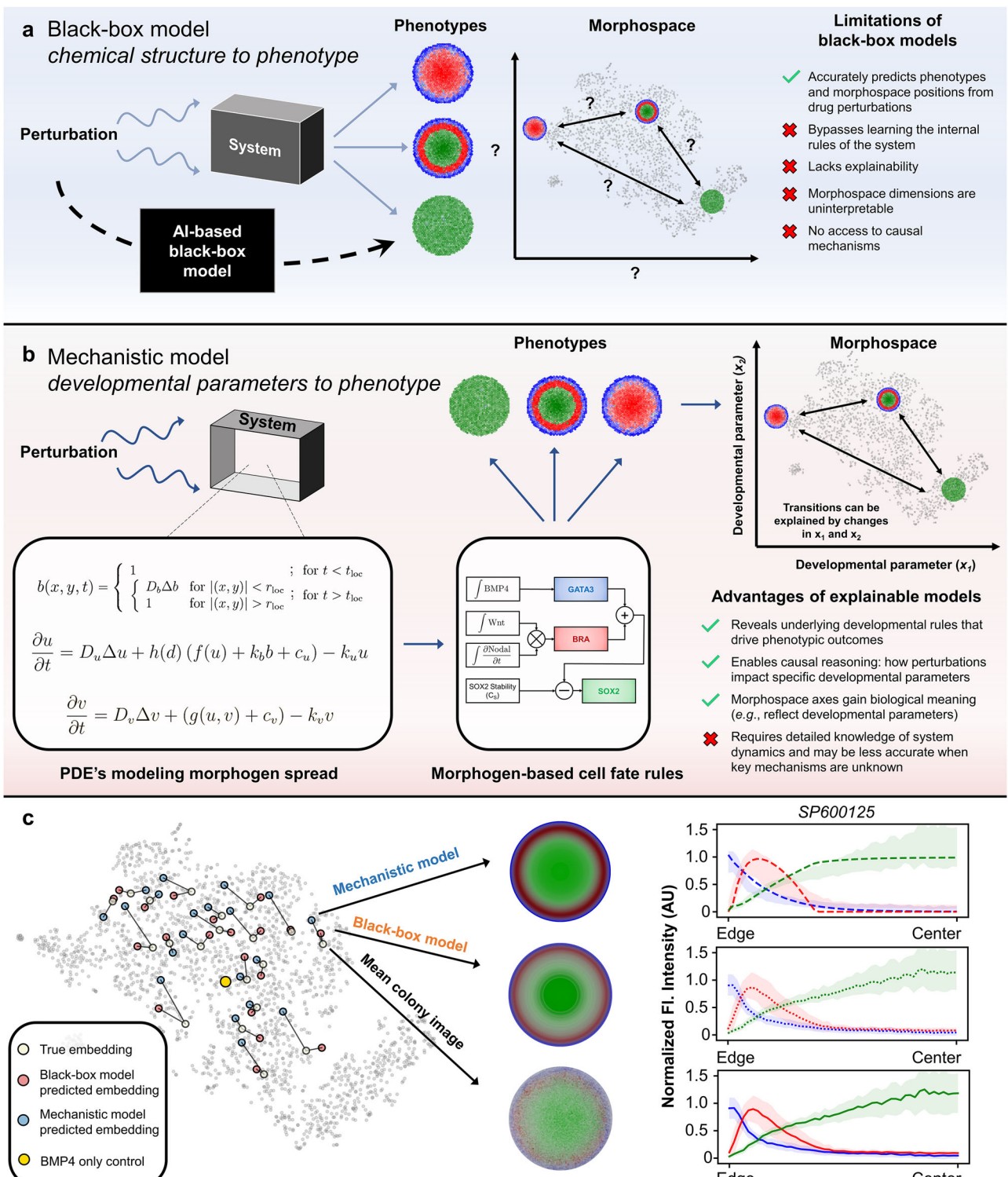

**Fig. 6 | Combining deep learning and mechanistic models reveals complementary strategies for decoding human development. a** AI-based black-box models predict gastruloid phenotypes and morphospace locations directly from chemical structure. While highly accurate, this approach bypasses the internal developmental rules of the system and lacks interpretability. **b** In contrast, a mechanistic model based on partial differential equations (PDEs) simulates morphogen diffusion and cell fate transitions from first principles. This enables developmental outcomes to be linked to specific parameters, such as cell density and SOX2 stability, thereby assigning biological meaning to morphospace axes and enabling causal reasoning. **c** Predicted morphospace embeddings from the black-box model (blue circles) and mechanistic model (red circles) align closely with experimental data (white circles), demonstrating strong agreement across representative compounds from each of the nine canonical patterning clusters. Both models also recapitulate the spatial phenotypes of drug-treated colonies with high fidelity. As an example, predictions for SP600125 are shown alongside its experimental phenotype. Radial intensity plots for GATA3 (blue), BRA (red), and SOX2 (green) compare model predictions (dashed lines) overlaid on shaded regions representing the mean ± SD across experimental replicates, illustrating how deep learning and mechanistic modeling provide complementary approaches for decoding human developmental outcomes. All fluorescence intensities are reported in arbitrary units (AU).

threshold, linking morphospace position to chromatin state, environmental responsiveness, and the broader logic of fate decisions during gastrulation. A recent study integrating time-lapse imaging of morphology with single-cell transcriptomics in 3D mouse trunk-like structures used machine learning to identify early features predictive of fate and found that metabolic state plays a key role in regulating differentiation outcomes[74]. Similar multimodal approaches could be applied to investigate the molecular determinants of SOX2 stability and its role in patterning outcomes in human gastruloids.

While FATE-MAP exhibits high accuracy (72.7%), balanced accuracy (74.7%), precision (88.9%), and specificity (80.0%) in identifying certain classes of teratogens, its sensitivity (69.6%) remains limited, and many known teratogens are not detected. Moreover, while zebrafish assays confirm that several predicted compounds disrupt conserved developmental signaling, the resulting phenotypes do not necessarily align one-to-one with those observed in gastruloids, and the gastruloid assay should be viewed as a human-relevant readout of early signaling disruption rather than a comprehensive or species-agnostic model of teratogenic outcomes. For example, compounds like valproic acid, which primarily affect neural tube development post-gastrulation[75], were not flagged by our assay. Additional signaling pathways, such as FGF[76,77], remain underexplored, and expanding the mechanistic model to include these will be critical to increasing coverage. Furthermore, all compounds were tested at a single active concentration (10 μM), which may underestimate failure modes occurring at higher doses. As a proof-of-principle, we performed titration experiments for two representative cluster 11 compounds (tretinoin and isotretinoin) and for an unannotated compound (IB-2744), observing clear dose-dependent phenotypes that mirrored their activity at 10 μM (Supplementary Fig. 15). These results demonstrate that our platform can resolve graded responses across a concentration range and highlight its potential for identifying lowest-observed-adverse-effect thresholds and for detecting mechanistic similarity among previously uncharacterized compounds. Future work will involve testing across a broader range of concentrations and expanding the set of known teratogens.

In addition, measuring just three transcription factors limited our ability to characterize certain cell fate decisions of rare cell types, including primordial germ cells, that emerge during gastrulation[78]. We also note that the absence of a definitive endoderm marker[26] (e.g., SOX17 or FOXA2) may mask potential defects in endoderm induction within otherwise canonical phenotypes. To characterize these rare, but important events, we plan to integrate additional protein markers and small molecule probes, such as spatial transcriptomics techniques[79]. For clarity, we note that FATE-MAP is used as an acronym for the framework and does not imply comprehensive mapping of all downstream lineage fates in its current form.

Overall, FATE-MAP demonstrates the promise of combining mechanistic models with AI-based prediction to build interpretable, predictive frameworks for human development. By embedding parameterized simulations into learned manifolds, this approach offers a generalizable strategy for understanding how molecular perturbations give rise to complex phenotypes, laying the foundation for predictive, scalable, mechanism-informed models of early human development.

## Methods

### Ethics statement
All experiments involving human pluripotent stem cells were performed in accordance with applicable regulations and the principles outlined in the 2025 ISSCR Guidelines for Stem Cell Research and Clinical Translation.

### Cell culture and differentiation
**Cell lines.** All experiments were conducted using H9 (WA09) human embryonic stem cells obtained from WiCell (Cat No. WB66446). For the initial drug screen, H9 cells were utilized to assess the impact of various drug treatments on patterning outcomes. Additionally, a CRISPR-Cas9 tagged β-Catenin (β-Cat) line was developed using H9 cells specifically for the quantification of Wnt signaling dynamics, enabling the study of active, non-membrane-bound β-Catenin in live cells.

**Maintenance and culture conditions.** For routine maintenance, H9 cells were cultured on Corning Matrigel hESC-Qualified Matrix, LDEV-free (Corning, Cat #: 354277) coated dishes, and grown in mTeSR Plus feeder-free maintenance medium (STEMCELL Technologies, Cat #: 100-0276). Matrigel-coated dishes were prepared by coating them overnight at 4 °C, followed by warming at room temperature for 1 h before cell seeding.

**Micropatterned gastruloid differentiation.** For micropattern cell culture, custom ordered 500 μm diameter micro-patterned 96 well glass bottom dishes (CYTOO Inc, Cat #: A500P650) were first coated with CellAdhere Laminin 521 (STEMCELL Technologies, Cat #: 200-0117) at 10 ug/mL for 2 h at room temperature. Established protocols for 2D gastruloid patterning were used[28]. Briefly, the wells are serially diluted with ice-cold calcium and magnesium free PBS (Thermo Fisher Scientific, Cat #: J67802.K2). Cells already resuspended in growth medium with 5 μM ROCK inhibitor (RI), Y-27632 (STEMCELL Technologies, Cat #: 72307) are then immediately plated upon removal of PBS. For cell seeding onto the micropattern, cells growing on Matrigel are lifted using ACCUTASE (STEMCELL Technologies, Cat #: 07920). Cells are centrifuged and $5 \times 10^5$ cells per well are resuspended in RI containing media and added to the micropatterned plate. After 4 h the medium is replaced with mTeSR plus without ROCK inhibitor and incubated overnight.

**Drug screening protocol.** For the drug screen, we utilized the Stem Cell Signaling Compound Library (Selleck Chemicals, Cat #: L2100), containing 210 distinct drug conditions. The unannotated compound IB-2744 was obtained separately from Specs (Cat. #AK-968/15360521). Each compound was provided at an initial concentration of 10 mM, either dissolved in DMSO or water, based on solubility requirements. The compounds were stored and handled according to the manufacturer's recommendations to maintain stability and activity. To prepare the working solutions, the compounds were aliquoted into 96-well plates and diluted to a final concentration of 10 μM in mTeSR Plus medium supplemented with BMP4 (STEMCELL Technologies, Cat #: 78211.1) at 50 ng/mL and without RI. Following the initial preparation of the micropatterned colonies, as described in the Micropatterning section, the cells were incubated overnight in mTeSR Plus medium. The next day, the media was carefully aspirated, and the BMP4 + drug mixture was added to the micropatterned wells, exposing the cells to both the differentiation cue (BMP4) and the specific drug conditions simultaneously. The plates were then incubated at 37 °C with 5% $CO_2$ for 48 h, allowing the cells to respond to the combined BMP4 and drug treatment.

Wells are fixed with 4% paraformaldehyde and rinsed twice with PBS then permeabilized with −80 °C ethanol for two min. Wells are then incubated with primary antibodies (Dilution ratio: 300:1): anti-mouse GATA3 (Thermo Fisher Scientific, Cat #: MA1-028, RRID: AB_2536713), anti-goat Brachyury (R&D Systems, Cat #: AF2085, RRID: AB_2200235), and anti-rabbit SOX2 (Cell Signaling Technology, Cat #: 3579, RRID: AB_2195767), at 4 °C overnight, washed three times in PBS for 5 min each, and incubated with secondary antibodies (Dilution ratio: 1000:1) Alexa Fluor™ 568 Donkey anti-Mouse (Thermo Fisher Scientific, Cat #: A10037, RRID: AB_11180865), Alexa Fluor™ 647 Donkey anti-Goat (Thermo Fisher Scientific, Cat #: A-21447, RRID: AB_2535864), and Alexa Fluor™ 488 Donkey anti-Rabbit (Thermo Fisher Scientific, Cat #: A-21206, RRID: AB_2535792) and DAPI nuclear counterstain (Thermo Fisher Scientific, Cat #: D3571) for 1 h at room temperature before being washed 3 times with PBS.

### Genetic engineering and reporter line generation

**CRISPR-Cas9 fluorescent tagging.** For CRISPR-Cas9 editing in H9 human embryonic stem cells, we utilized Lipofectamine Stem Transfection Reagent (Thermo Fisher Scientific, Cat #: STEM00015), to transfect a plasmid expressing a gRNA targeting the N-terminus of the *CTNNB1* gene, a homology template with ~180 bp of both upstream and downstream homology to the targeted insertion site flanking the tdmRuby2 construct (Supplementary Data 5), and the Cas9 enzyme (IDT) according to manufacturer's directions. Edited cells were passaged into a single cell suspension and sorted, into ROCKi (5 μM in mTeSR), for tdmRuby2 positive clones using a Sony SH800 FACS system. Clones we isolated and expanded prior to verification of successful edition through PCR followed by Sanger sequencing.

### Imaging and image analysis

**Imaging and quantification of gastruloid patterning.** Fixed cell imaging of drug-treated gastruloids was carried out using a Nikon W2 SoRa spinning-disk confocal microscope equipped with incubation chamber maintaining cells at 37 °C and 5% CO2. We imaged in the four channels corresponding to DAPI, Alexa488, Alexa555 and Alexa647 conjugated antibodies. Images were exported as .tiff files and analyzed using the custom MATLAB and Python software, enabling quantification of the spatial distribution and intensity of cell fate markers across various drug treatments.

We developed custom MATLAB software to extract both raw and radial intensity profiles from imaged colonies. The pipeline begins by isolating the DAPI channel, converting it to a binary mask, and removing background noise using morphological filtering. A standard nuclei detection algorithm involving fill, dilation, and connected component analysis is then applied to generate a mask and identify nuclear positions. This is followed by secondary segmentation using enhanced binarization and a regional maxima filter to refine nuclei detection. Intensity values for each germ layer marker are extracted by applying the nuclei mask to the corresponding channels. Colony edges are determined by computing the centroid of all nuclei and applying a boundary detection algorithm to identify each cell's proximity to the colony edge. The colony is divided into 50 concentric radial bins, and average marker intensities are computed per bin.

**Live imaging and quantification of density-dependent mesoderm specification.** Live-cell imaging of 2D gastruloids seeded at varying densities (10–100%) was performed using the CellVoyager CQ1 Benchtop High-Content Analysis System (Yokogawa Electric Corporation). Cells were seeded at $5 \times 10^5$ per well in ROCK inhibitor-containing media, diluted to the desired concentration. Eight colony positions were selected and saved per well to enable consistent tracking across time. Imaging was conducted under incubation conditions (37 °C, 5% CO₂), with z-stacks acquired every 15 minutes using brightfield and 561 nm laser illumination with autofocus enabled. Images were saved as .tiff files and analyzed using custom Python software to assess Wnt signaling dynamics.

Following live imaging, colonies were fixed, immunostained for BRA, and re-imaged using the previously saved positions to directly compare live signaling dynamics with endpoint fate outcomes. Fixed-cell imaging was conducted on the same CQ1 system, acquiring z-stacks under identical incubation and autofocus settings. Images were analyzed using custom Python and MATLAB software to quantify germ layer marker expression across seeding densities.

**Wnt signaling quantification and kymograph generation.** To quantify active Wnt signaling, we developed a custom Python-based image processing pipeline to isolate non-membrane (active) β-catenin from the membrane-bound pool. After background subtraction on .tiff images, a Gaussian blur reduced noise, and adaptive thresholding generated a binary membrane mask. This mask was dilated and subtracted from the original image to isolate cytoplasmic and nuclear β-catenin. Morphological operations further refined the mask, and mean fluorescence intensity was calculated to quantify Wnt signaling activity. To assess temporal dynamics, the same pipeline was applied to time-lapse images, with each colony divided into 50 radial bins. Mean intensities were computed per bin across frames to generate kymographs capturing spatial and temporal patterns of β-catenin signaling.

### Zebrafish teratogenicity assay

**Zebrafish maintenance and compound treatment.** All experiments were conducted using wild-type AB zebrafish (*Danio rerio*) maintained at 28 ± 1 °C with a 14-h light and 10-h dark cycle. All zebrafish husbandry, breeding, and experimental procedures were conducted by a commercial research organization (ZeClinics, Barcelona, Spain) in accordance with institutional guidelines and applicable national and European regulations governing the use of animals in research. Fertilized zebrafish embryos were collected in E3 1X medium (60X stock is prepared by dissolving 34.8 g NaCl, 1.6 g KCl, 5.8 g CaCl₂·2H₂O, 9.78 g MgCl₂·6H₂O in 2 L H₂O and adjusting the pH to 7.2 using NaOH) in Petri dishes, and abnormal or unfertilized embryos were discarded at 6 h post-fertilization (hpf). Healthy embryos were grown up to 96 hpf and exposed to ulixertinib, nirogacestat, and isotretinoin at five concentrations (0.01 μM, 0.1 μM, 1 μM, 10 μM, 100 μM) starting at 6 hpf. 4-Diethylaminobenzaldehyde (DEAB) (MilliporeSigma, Cat #: D86256-100G) was used as a positive control at concentrations of 0.1 μM, 0.3 μM, 1 μM, 3 μM, and 10 μM.

**Phenotype quantification and analysis.** Mortality was assessed at 24 h and 96 h, and larvae were imaged at 96 hpf using the automated VAST system (Union Biometrica). Dead, unhatched, and incorrectly detected larvae were excluded from analysis, resulting in a variable number of larvae analyzed per condition. Teratogenic effects were quantified by assessing both qualitative and quantitative phenotypes. Qualitative phenotypes included body curvature abnormalities, yolk edema, necrosis, and craniofacial defects, while quantitative phenotypes such as lateral area, eye diameter, pigmentation, and body length were measured using FIJI software. Continuous data were transformed to binary outcomes based on interquartile range thresholds.

LC50 and EC50 values, representing the concentration at which 50% of mortality is observed and the concentration at which 50% larvae population show a teratogenic phenotype, respectively, were calculated using dose-response models via Proast software, allowing for the determination of the teratogenic index (TI) as the ratio between LC50 and EC50 for the most sensitive phenotype. The Lowest Observed Effect Concentration (LOEC) was identified for each compound, with effects within 20% of biological range considered non-significant.

### Data analysis and computational modeling

**Dimensionality reduction and clustering.** Bin-wise features were normalized per plate using robust scaling (25th–75th percentiles) to mitigate batch effects and outliers. The normalized single-colony profiles were embedded into two dimensions with t-SNE. To define morphological phenotypes, we applied a Gaussian kernel density estimate (KDE) over the t-SNE space of all experimental colonies, identified local density maxima, and then performed watershed segmentation to partition the morphospace into discrete, contiguous regions (Supplementary Fig. 1g). This unsupervised procedure does not require pre-specifying the number of clusters and yields cluster boundaries that reflect the empirical structure of the embedded data. While alternative clustering methods (*e.g.*, Spectral or Leiden) produce similar numbers of clusters, the watershed-defined boundaries offer a key practical advantage in that they can be directly reused to assign new predicted phenotypes without re-running clustering, ensuring consistent and unbiased classification across experimental and in silico data.

**Simulation of reaction-diffusion models.** The reaction-diffusion models were simulated using a custom Python script to capture the dynamic interactions of signaling pathways in the system. The simulation involved creating a 2D spatial grid to represent the environment and updating the concentrations of key signaling molecules over time. Standard numerical techniques were applied to compute diffusion, reaction, and degradation processes. Custom functions handled boundary conditions and interactions between signaling molecules.

**Simulation of cell fate specification.** Cell fate simulations were conducted using custom Python scripts designed to integrate the dynamics of morphogen signaling with rules for determining cell fate outcomes. The process involved calculating fate maps based on our cell fate equations to determine the expression levels of GATA3, BRA, and SOX2 proteins. Visualization of fate maps was performed by generating heatmaps of normalized expression levels using custom color maps to represent each fate (GATA, BRA, and SOX). Separate images were created for each morphogen, and a composite image was generated to illustrate the combined spatial distribution of all fates within the colony. Additionally, radial intensity profiles were computed by binning the colony into concentric rings and averaging the protein expression levels within each ring.

### Neural network-based modeling

**Chemical structure-to-phenotype prediction (ChemBERTa).** To predict gastruloid phenotypes directly from chemical structure, we used a two-stage model based on molecular embeddings generated from ChemBERTa. ChemBERTa was downloaded from the Hugging-Face repository seyonec/PubChem10M_SMILES_BPE_450k, which was pretrained on over 10 million SMILES strings using masked language modeling and byte pair encoding (BPE). SMILES strings were tokenized and embedded with the HuggingFace Transformers library, yielding a fixed-length molecular representation for each compound. The dataset was split 90/10 into training and blind test sets, with each compound represented by a ChemBERTa embedding paired to its corresponding radial phenotype vector.

In the first stage, we trained a neural network classifier to map ChemBERTa embeddings to logits over morphological clusters. The classifier consisted of two fully connected layers (512 and 256 units) with ReLU activations and dropout, followed by a linear output layer. Training used the Adam optimizer (learning rate = 0.001) with a combined cross-entropy and cost-sensitive loss designed to penalize misclassification of failure-mode clusters. Optimization was performed using 5-fold cross-validation with an early-stopping criterion based on validation accuracy (patience = 30 epochs). Final cluster assignments were obtained from fold-averaged logits. Each compound was then assigned the mean 150-dimensional phenotype vector of its predicted cluster.

In the second stage, we modeled structure-dependent deviations from these cluster means. For each cluster, a ridge regression model was trained to predict the residual between the cluster mean and the observed 150-dimensional phenotype vector. The final output phenotype vector for each compound was obtained by adding the predicted residual to the cluster mean, and accuracy was evaluated by comparing predicted and experimental radial profiles and morpho-space locations.

**Approximation of t-SNE embedding.** To approximate the t-SNE embedding of high-dimensional phenotypic data, we trained a neural network, $\psi(y^-)$, to predict low-dimensional projections from bin-wise fate marker distributions. The network comprised three fully connected layers with 128, 64, and 2 neurons, respectively, and ReLU activations. Input data were split 80/20 into training and validation sets.

Training was performed for up to 1000 epochs using the Adam optimizer (learning rate 0.001) with mean squared error loss. A learning rate scheduler was used to reduce the step size when validation loss plateaued, and early stopping (patience = 50 epochs) prevented overfitting. Model accuracy was assessed by comparing predicted versus true t-SNE embeddings using Jensen-Shannon Divergence (JSD) and visual overlay of density plots.

### Mathematical modeling

**Reaction-diffusion model for morphogen spread.** Inspired by recent studies examining BMP-Wnt-Nodal dynamics in 2D gastruloids[47], we developed a simplified reaction-diffusion model tailored to our experimental observations. The model specifically addresses the influence of cell density on mesoderm patterning, as we observed that lower cell density shifts the mesoderm peak inward and causes the mesoderm region to expand outward. This density-dependent modulation was incorporated into our model, accounting for the dynamic interactions between BMP, Wnt, and Nodal, and aligning with observed spatial and temporal morphogen patterns.

BMP was modeled with the following equation:

$$\begin{cases} b(x,y,t) = 1, & t \le t_{loc} \\ \frac{\partial b}{\partial t} = D_b \Delta b, & t > t_{loc}, \quad |(x,y)| \le r_{loc} \\ b(x,y,t) = 1, & t > t_{loc}, \quad |(x,y)| \le r_{loc} \end{cases} \quad (1)$$

The BMP signaling dynamics are modeled as a diffusive species within a localized region, $r_{loc}$, which varies based on cell density. Initially, BMP concentration is set to 1 across the entire domain, representing the uniform activation phase. After $t_{loc}$, BMP diffusion is restricted within $r_{loc}$, simulating the feedback inhibition observed experimentally, where BMP signaling diminishes centrally but remains active at the periphery[40]. In this context, $D_b$, represents the diffusion coefficient of BMP, $r_{loc}$, denotes the radius defining the localized BMP diffusion modulated by cell density, and $t_{loc}$ indicates the time point when BMP signaling becomes localized. The boundary for BMP signaling, $r_{loc}$, is defined by $r_{rad}*(1 - e^{-a*d})$, where and $r_{rad}$ is the radius of the stem cell colony. Here, $a$ is a constant that scales the effect of cell density, and $d$ represents the cell density; this formulation reflects our observation that the outermost cells spread more at lower densities, expanding the peripheral region in which BMP signaling remains active. The spread of Wnt was modeled by:

$$\frac{\partial u}{\partial t} = D_u \Delta u + h(d)\big(f(u) + k_b b + c_u\big) - k_u u \quad (2)$$

Where:

$$h(d) = 1 - c_m\left(\frac{d}{c_d} - 1\right) \quad (3)$$

$$f(u) = \frac{s_u u^2}{1 + K_u u^4} \quad (4)$$

The Wnt signaling dynamics incorporate diffusion and autocatalytic activation, modulated by the density-dependent scaling function h(d). The term f(u) captures the autocatalytic production of Wnt, with inhibition effects included through a saturation term in the denominator to limit excessive Wnt production at high concentrations. Here, $D_u$, is the diffusion coefficient of Wnt, h(d) is the density-dependent scaling function, $s_u$ denotes the rate of Wnt autocatalytic production, $K_u$ is the saturation constant controlling Wnt production, $k_b$ represents the contribution of BMP to Wnt production, $c_u$ is a constant source term for Wnt, and $k_u$ is the degradation rate on Wnt. The scaling function h(d) modulates Wnt reaction terms based on cell

density, where $c_m$ is a scaling factor, $d$ is the cell density, and $c_d$ is a constant representing the mean cell density from the screen. The spread of Nodal was modeled by:

$$\frac{\partial v}{\partial t} = D_v \Delta v + g(u, v) - k_v v + c_v \qquad (5)$$

Where:

$$g(u, v) = \frac{s_v v}{1 + K_v v^2} u \qquad (6)$$

The Nodal signaling dynamics incorporate diffusion and production influenced by Wnt activity, as captured by the term g(u,v), which includes a saturation effect to reduce Nodal production at high concentrations. Unlike previous models, we chose not to model Nodal as autocatalytic due to evidence indicating that Nodal spreads primarily through a relay mechanism rather than an autocatalytic loop[80]. In this context, $D_v$, denotes the diffusion coefficient of Nodal, $s_v$, is the rate of Nodal production influenced by Wnt, $K_v$ is the saturation constant reducing Nodal production at high concentrations, $c_v$ represents the constant source term for Nodal, and $k_v$ is the degradation rate of Nodal. All parameter values used in the model can be found in Supplementary Data 6.

**Morphogen-based cell fate determination model.** We developed cell-fate determination rules based on the integration of BMP, Wnt, and Nodal signaling dynamics. Each fate map was generated by simulating the morphogen concentrations over time, followed by applying specific computational rules to map the spatial location of fate-markers, GATA3, BRA, and SOX2, throughout the colony. Specifically, GATA3 expression was modeled by:

$$GATA3 = \frac{\int b(x, y, t) dt}{1 + e^{-k_1 \left( \sqrt{x^2 + y^2} - r_{loc} \right)}} \qquad (7)$$

The expression of GATA3 is modeled as a response to the integral of BMP signaling over time, reflecting the cumulative impact of BMP exposure on cells[43]. The denominator accounts for the preferential availability of transforming growth factor β receptors at the edge of the colony[40], with the steepness parameter modulating the influence of distance from the colony edge. Here, $\sqrt{x^2 + y^2}$ represents the radial position within the colony, $r_{loc}$ defines the localized region where BMP signaling is most concentrated, and $k_1$ adjusts the drop-off rate of GATA3 expression as distance increases from the colony edge increased. BRA expression was modeled by:

$$BRA = \int \frac{1}{1 + e^{-k_2 \left( u(x, y, t) - Wnt_{thresh} \right)}} \frac{\partial v(x, y, t)}{\partial t} dt \qquad (8)$$

BRA expression is governed by the combined influence of Wnt and the rate of change of Nodal signaling, capturing the dynamics where Nodal alone does not induce mesoderm formation[81]. Here, the Wnt concentration must reach a certain threshold, $Wnt_{thresh}$ in order for the cells to respond to changes in the rate of Nodal concentration. Specifically, unlike BMP, which is concentration-dependent, our experimental observations indicate that many Nodal targets depend on the rate of concentration change rather than its absolute levels. Here, $k_2$ represents the steepness of the BRA activation response to Wnt, $Wnt_{thresh}$ denotes the threshold level of Wnt required for BRA expression, and $\frac{\partial v(x, y, t)}{\partial t}$ is the rate of change of Nodal concentration over time. SOX2 expression was modeled by:

$$SOX2 = C_s \left( c_1 - c_2 GATA3 - c_3 BRA \right) \qquad (9)$$

SOX2 expression is modeled as inversely proportional to the levels of GATA3 and BRA, aligning with our observations that SOX2 expression typically decreases as cells begin to differentiate. The stability constant, $C_s$, modulates the persistence of SOX2 under varying conditions, reflecting our experimental findings that certain drug treatments allow cells to maintain high SOX2 expression despite concurrent BRA expression. In this equation, $C_1$, $C_2$, and $C_3$ are constants that adjust the relative influence of GATA3 and BRA on SOX2 suppression. All parameter values used in the model can be found in Supplementary Data 7.

**Materials Availability.** Plasmids generated in this study are available upon request to Maxwell Wilson (mzw@ucsb.edu) without restrictions.

### Reporting summary
Further information on research design is available in the Nature Portfolio Reporting Summary linked to this article.

## Data availability
The raw and processed gastruloid images and corresponding morphospace data (radial intensity profiles and cluster assignments) used in all analyses are publicly available at (https://max-wilson.mcdb.ucsb.edu/research/gastruloid-morphospace and https://github.com/MZW-Lab/gastruloid_morphospace). These datasets include the binned radial fluorescence intensity profiles for each colony and constitute the complete dataset used to generate the results reported in this study. Additional phenotype categorizations are provided in the Supplementary Information/Source Data file.

## Code availability
All code generated during this study has been deposited at the Wilson Github repository (https://github.com/MZW-Lab/gastruloid_morphospace).

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

## Acknowledgements

This study was supported by UCSB start-up funds. J.R. is supported by NICHD of the National Institutes of Health via a Ruth L. Kirschstein Postdoctoral Individual National Research Service Award (1F32HD114539-01). The authors acknowledge the use of the Micro-fluidics Laboratory within the California NanoSystems Institute, supported by the University of California, Santa Barbara and the University of California, Office of the President. Results related to the zebrafish teratogenicity assay were performed in collaboration with ZeClinics. The authors also thank Angela Pitenis and Dennis Clegg for their valuable feedback and discussions on experimental and figure design.

## Author contributions

Conceptualization, J.R., C.Q., D.H., and M.Z.W.; methodology, J.R., C.Q., D.H., N.B., G.D., and M.Z.W.; software and data analysis, J.R., C.Q., and M.Z.W.; modeling and simulations, J.R. and M.Z.W.; investigation, J.R., C.Q., D.H., N.B., G.D., J.D., and M.Z.W.; writing–original draft, J.R. and M.Z.W.; editing–final draft, J.R., F.W., J.J.C., and M.Z.W.; funding acquisition, J.R. and M.Z.W.

## Competing interests

F.W. and M.Z.W. are employees, shareholders, and board members of Integrated Biosciences Inc. J.J.C. is the founding Scientific Advisory Board chair of Integrated Biosciences. The remaining authors declare no competing interests.

## Additional information

[1]Department of Molecular, Cellular, and Developmental Biology, University of California Santa Barbara, Santa Barbara, CA, USA. [2]Center for BioEngineering, University of California Santa Barbara, Santa Barbara, CA, USA. [3]Neuroscience Research Institute, University of California Santa Barbara, Santa Barbara, CA, USA. [4]Integrated Biosciences, Inc., Redwood City, CA, USA. [5]Infectious Disease and Microbiome Program, Broad Institute of MIT and Harvard, Cambridge, MA, USA. [6]Institute for Medical Engineering and Science and Department of Biological Engineering, Massachusetts Institute of Technology, Cambridge, MA, USA. [7]Wyss Institute for Biologically Inspired Engineering, Harvard University, Boston, MA, USA. ✉e-mail: mzw@ucsb.edu

