## [Peer Review file · Nature Communications]

FATE-MAP predicts teratogenicity and human gastrulation failure modes by integrating deep learning and mechanistic modeling

Corresponding Author: Professor Maxwell Wilson

Version 0:

Reviewer comments:

Reviewer #1

(Remarks to the Author)

In this work, the authors undertake a formidable effort to screen and quantify the effects of 210 compounds on the development and differentiation of 2D gastruloids from hPSCs, a system well established, primarily by the Warmflash lab. They quantify three fluorescent markers in 10 colonies per treatment, ultimately quantifying images from over 2,000 gastruloids at the single-cell level, a remarkable scale for this kind of analysis. The fluorescence quantification is converted to a vector describing the radial intensity profile for each marker.

The idea of generating a mean behaviour image for each cluster is nice, seems to work well in these radially symmetric colonies, and provides an intuitive understanding of each response type. The authors cluster the gastruloid vectors into 12 clusters, assigning 9 or them to normal behaviour and 3 clusters to distinct failure modes (loss of Bra/Sox2 or loss of radial symmetry). Note that converting a profile with a geometric meaning (expression along the radial axis) into a vector (simply by binning) loses all geometric information. For example, under standard metrics the vector 1000 will be equidistant from 01000 and from 00001, while in its geometric interpretation it is clearly closer to the first one. All the same, the clusters obtained seem to be meaningful. The authors may want to discuss this point, and perhaps point to limitations of this embedding. Of course, when we move to network learning in the next step, the network model can re-learn the geometric connections between output nodes.

In the next step, the authors encode the 210 chemical compounds as vectors, and train a network model to predict the gastruloid vector (output) from this encoded input. They also train an additional (smaller) network to embed the predicted vector in tSNE "morphospace embedding". I assume the prediction errors were done on the original 150-dim vector space, correct? Why is it important to embed into the 2-dim tSNE space? It is only used for visualization purposes. As the name says, it is only a 2-dim embedding of a 150-dim vector, which itself is a representation of a 2-d circular pattern. If the purpose is mostly to evaluate the error (as depicted in Fig. 2b) for the 150-dim space, the embedding process only adds some more distortion to it. It is more informative to show some quantification of the prediction errors (as depicted in the red/green/blue plots), how they behave in training vs. test set, etc.

In the sample images in Fig. 3a we see some bald regions within the colony (which we do not see in the examples in Fig. 2. What are these? Why do we only see them here?

The authors then run 947 FDA approved drugs through their prediction pipeline, finding 100 to fall within the no-Brachyury cluster. The association statistics corresponds with pre-known risk levels (as teratogens) of those drugs. A few comments on this approach/result:

- Looking at Fig. 3c, it is not clear how association of the new compounds to cluster 11 was determined. Some of them (including several category X compounds) seem just as close (or closer) to clusters 8 and 9. Or is it because the association was computed in the 150-d space, and the 2-d projection is just misleading? The same is true for Ulixertinib prediction.
- In this specific example, the association to cluster 11 describes failure in the form of no Brachyury expression. Did we have to go through the sophisticated experimental model (2D gastruloid) and heavy computational framework? Wouldn't we get a similar (or better) prediction simply screening Bra-expression levels (i.e. a 1d readout instead of 150d readout) in e.g. simple diff protocol applied to a 2D culture? It is worth at least discussing this point.

The authors then devise a mechanistic model, that models the spatio-temporal dynamics of Wnt, BMP and Nodal activity, and tie it with the measurable marker expressions (that serve as a proxy for cell fate) through a simple logic-based model. They demonstrate how such a model can predict the spatial patterns observed, as well as their dependence on cell density. They experimentally validate one implication from the model simulations, which is the inverse dependence of Wnt signaling on cell density.

They then parametrize morphospace by projecting gastruloids predicted by the mechanistic model while varying two parameters: cell density and SOX2 stability. It is not clear (or well motivated) why SOX2 stability was chosen – this is not a parameter that can be directly controlled experimentally, and it is not clear to what degree it naturally varies between the gastruloids. The authors claim that Sox2 stability (Cs) “emerged as a key regulator of fate boundary positions in our model.” This claim is a bit misleading – the model was devised in a way that gives Cs a high weight, determining the resulting patterns. For example, the stabilities of Gata3 or Brachyury do not appear as parameters in the model at all. This should be better explained and motivated.

The idea of parametrizing the abstract UMAP embedding by varying interpretable parameters is nice, as it adds to the interpretability of the morphospace.

The authors present the AI and mechanistic models as if they were two alternatives to choose from. This is a bit misleading, as the mechanistic model does not incorporate the potential effects of teratogens. If I understand the rationale of the authors, in integrating (and comparing) the two, they assume that the effect of the teratogens can be summed up through their effect on just 2 parameters (cell density and SOX2 stability). What is the rationale for that? Are the effects of the compounds on density simply due to toxicity? (this could be cross-checked against simple growth curves in 2D cell culture for these compounds). And if they affect Sox2 stability (as rationalized in the Discussion), don't they have similar effects on GATA3 and Brachyury? Or on Wnt, Nodal and BMP proteins?

In summary, the effort to generate a prediction system for the effects of compounds on human embryonic development is commendable. The AI-generated predictions seem to generate useful predictions. The integration with the mechanistic model, while adding mathematical gravity to the manuscript, is not very clear currently.

(Remarks on code availability)

Reviewer #2

(Remarks to the Author)

Summary:

This paper develops a framework for high throughput drug screening and phenotype prediction for 2D gastruloids. Improving on previous work describing teratogenicity screening in vitro, the authors develop an integrated approach of deep learning and mechanistic modeling to predict and interpret phenotypes. The paper contains several nice contributions to the field, including 1) the machine learning model mapping phenotypes into a (somewhat) interpretable 2D space, 2) a large dataset made public of the effect of different drug treatments on 2D gastruloids, 3) an extension of the model from Chhabra et al 2019 to include cell density and explicit modeling of cell fate. However, a major weakness is a lack of rigor and detail, including poorly motivated choices in approach. For a paper about what is essentially a statistical testing and prediction framework, rigor seems particularly important.

Major comments:

- The clarity and accuracy of the abstract could be improved. First, it claims to elucidate “failure modes” but neither the abstract nor the rest of the paper clearly defines what a failure “mode” is. If what is meant by this is that the phenotypes can be captured by cell density and SOX2 stability, perhaps that should be made more explicitly. It also says the paper “revealed” cell density and SOX2 stability as determinants of patterning, but both were already known, the author do in fact cite previous work showing the importance of density (e.g. Etoc et al) and SOX2 stability (Camacho-Aguilar et al, Teague et al). Perhaps corroborate is a better word and the discussion would ideally more explicitly incorporate this past work.
- Rigor should be improved:
 - o The paper states “10 colonies per drug condition were imaged”. However, it is not clear whether these all came from the same experiment and if so if they came from the same well. Inter-colony variation within a well is very small, between wells on the same plate small, between experiments or different plates within an experiment can be significant. To understand if phenotypes of two drugs are significantly different, or whether the prediction is correct within the experimental uncertainty, it is essential to quantify the inter-experiment phenotypic variation under the same drug perturbations and to make clear whether all drug treatment data was collected in the same experiment and if not to show if there are differences in the control condition between experiments and determine if there are any batch effects.
 - o The untreated (BMP only) control conditions is not shown in most figures but should be for comparison. For example, in which cluster does it fall in Fig 1g and where is it in 1f, and in Fig 6c – how does Raloxifene HCl compare to untreated? Can it be said to be significantly different? The same for all other figures. If I understand it correctly, Supplemental Fig. 5a shows the BMP only control clusters and their natural variation. This covers a large part of the map and the majority of the

- "Canonical Cell Patterning" clusters in Fig.1g, suggesting that those clusters should perhaps be a single cluster and that most drug treatments within that cluster cannot be distinguished from control.
- o More generally, it is not clear the clusters are meaningful. The authors use a custom method based on watershed segmentation. It is not clear why they need this, and the method is not described in sufficient detail to be reproducible. To motivate the approach, it is stated that an advantage is that the number of clusters is not predetermined but there are many methods for which that is true, such as Leiden/Louvain. Furthermore, one can optimize cluster number for methods with fixed cluster number using cluster analysis. Regardless, cluster analysis should be performed here to determine if the obtained clusters are statistically meaningful, e.g. calculating Silhouette coefficients, etc (https://en.wikipedia.org/wiki/Cluster_analysis). This reviewer would not be surprised if the number of statistically and biologically meaningful clusters is a lot smaller than the 12 that are discussed.
 - o Error bars on radial profiles. The analysis throughout the paper only considers the mean radial profile for each colony, but there are error bars on these radial profiles that may also change by perturbation, e.g. the variance in expression for some distance from the edge goes up when the patterns become less symmetric, and generally one could imagine propagating these error bars to the morphospace to get a sense of how significantly different the profiles represented by dots there are.
 - o In testing the prediction framework, the separation into training and testing data is essential. How was the 10% of compounds held out for test selected? Randomly? It would be good to see N-fold cross-validation to see how much performance varies as different compounds are held out for validation.
 - o The authors use RMSE as a measure of goodness of fit, but these numbers have no absolute meaning. Cosine similarity is also hard to interpret as good or bad. More meaningful would be to see if prediction and data agree within the experimental error.
 - o In figure 3, the authors compare their prediction to a "ground truth" drug classification. They should show at least a confusion matrix with accuracy, % false positive, false negative, perhaps also check other measures like precision. Systematically showing %enrichment of drug classes in clusters would also be good. Looking at Figure 3b it looks well mixed.
 - For figure 3, the authors focus on ulixertinib and nirogacestat for verification. Why are these not shown among the data points in Figure 2? Looking at the Supp. Fig. 4ab, the colony images show dark blue lines streaking along and across the colonies. These do not look like nuclear stains for the expected markers. What are they? If they are not cells, how do they affect analysis? are they excluded? If they are cells, why does their pattern look so strange?
 - As stated above, the modeling in Fig. 4 is one of my favorite parts of this paper, but it does raise some questions that if addressed would further strengthen the manuscript. The authors model Wnt production as density-dependent and verify the predicted Wnt signaling dynamics. This nicely captures the narrower range of Wnt signaling at higher density, but their model also shows a later onset of Wnt signaling in Fig.4c which the experimental data in Fig.4e does not seem to show. Can this be explained? Can the model be improved to capture this? If it cannot be addressed, perhaps the authors can add it to the discussion.
 - As a separate point about the model, it is surprising that SOX2 is modeled essentially as "not GATA3 or BRA", given that causally it is not true that SOX2 is directly downregulated by these factors (possibly the reverse) and that ultimately the authors emphasize SOX2 stability as the most important model feature. Is it fair to say SOX2 stability simply means degree of differentiation the way the model is set up? More importantly, if the authors truly think drugs directly affect SOX2 stability, it would be good if they would show this, similar to how they showed the density-dependent Wnt signaling.
 - The discussion illustrates the issue with thinking of all drugs as directly affecting SOX2 stability. It mentions in line 620 that ROCK inhibitors destabilize SOX2. However, what more likely happens is that they prevent tight junction formation, leading to increased BMP signaling, which is what actually destabilizes SOX2. In other words, there are many possible indirect of a drug that can affect differentiation.
 - In lines 487-496, the authors ask if the mechanistic model can capture the phenotype of drug treatment. They infer the SOX2 stability from interpolation of simulation for different SOX2 stabilities on the morphospace. Thus, it is not surprising that if one provides SOX2 stability and density one ends up roughly in the same place, the only way I can see this going wrong is if the experimental density used deviates strongly from the mean density coordinate on the morphospace. I guess this is what the authors may mean in lines 548-553. To summarize, this seems somewhat like circular reasoning and the authors may wish to make clearer whether this test really proves anything. Perhaps this could involve merging figures 5 and 6. Can the authors discuss more why the model cannot capture the effect of isoxazole and longdaysin? Should it not predict reduced/increased SOX2 stability for a Wnt agonist/inhibitor?

Minor comments:

- The introduction line 60 states: "methods to assess the teratogenicity of compounds have not advanced beyond exposing model organisms", but this is inaccurate, and the authors seem aware as they later cite two micropatterned approaches of teratogenicity testing. There are several other papers describing vitro approaches they may also wish to include, e.g., PMID: 26387793, PMID: 31207406, and others.
- Line 81: "3D models such as embryoids, blastoids, and SEMs" – embryoid is a general term that includes blastoids and SEMs. What's something different meant?
- Line 85: "cost of altering the topology of the developmental tissue" – which tissue? It is true that the topology of the amnion is altered, but the epiblast geometry and topology is a disc as in vivo, which it actually is not some 3D models. It is a misunderstanding that 3D is always more similar to in vivo than 2D.
- Figure 1c shows classes of signal pattern perturbation but as far as I can tell two of these never occur: inverted and mixed, so why list them?
- Clusters should be indicated in Fig.1e.
- The authors use GELU for one network, RELU for another and make a point out of this in the main manuscript, however, it is not explained why. More generally data supporting the choice of network architecture is missing.
- The shadows underneath the colony pictures throughout the manuscript, e.g., Fig.1g make the patterns harder to

distinguish, perhaps a black outline would be better if contrast from the which background is needed.

- Line 345: "cell seeding follows a Poisson distribution". That is only true for very sparse seeding and typically not for micropatterns which are densely seeded. The authors should show this or could simply remove it since it does not really matter, their only point is there is naturally occurring density variation.
- The regression of BRA peak vs. cell number in Supp. Fig. 5ac is quite nice. However, it would be good if in Fig. 1c not only the cluster averages but all datapoints were shown on the graph. Conversely, it would be helpful to see the cluster averages on the t-SNE plot.
- In line 461-466, it is said density and SOX2 stability emerged as key parameters. How did they emerge? How were key parameters identified?
- Line 566: "animal models, which are limited by species-specific differences and dose-dependent lethality". This appears to suggest that dose-dependent effects are not an issue in this work but they are and the authors acknowledge this later in the discussion (everything done at 10uM), which is good, so they could consider rephrasing line 566.

(Remarks on code availability)

Reviewer #3

(Remarks to the Author)

Summary:

In this manuscript, Rufo et al., develop an experimental and analytical screening platform, FATE-MAP, to assess phenotypic outcomes in human 2D stem cell models of early embryogenesis (human 2D gastruloids) treated with hundreds of chemical perturbations. FATE-MAP includes a deep learning framework to predict the effects of any molecule on 2D gastruloids development based on the molecule's chemical structures. The authors first characterized "failure modes", phenotypes that emerge when the 2D gastruloids are not properly patterned and then analyze over two thousand structures subjected to roughly 200 perturbations, including clinically relevant teratogens, eventually identifying several molecules that results in aberrant developmental programs. The authors also performed some validation experiments using the zebrafish model to corroborate the predicted molecule-induced phenotypic outcomes. They then model morphogen dynamics in 2D gastruloids, an approach that revealed how cell density and expression stability of the transcription factor SOX2 play a major role in the correct patterning of 2D gastruloids. Finally, they attempt to derive mechanistic principles of failure modes from their deep learning approach, implementing a mechanistic model to gain interpretability and biological meaning.

Overall, this reviewer thinks the developed methodology is important and timely, given the little knowledge we have about the effect of environmental offences on embryonic development, especially humans. Moreover, cheap and accessible high-throughput screening opportunities are being developed, and I think this method and its future implementations (like others of its kind) are in need to the field. Science and experiments are solid and well executed. However, this reviewer thinks the biological findings are not groundbreaking, especially given the opportunity provided by the method. The manuscript is very well written and easy to understand for a broad audience. I generally support its publication at Nature Communications, pending some points that would need to be addressed.

Major comments:

1) List of selected molecules:

- It would be important if the author could comment more on the list of 210 drugs tested. How were they selected? What is known about their effect in other species? Are they already classified or stratified based on their chemical structures?
- This becomes more relevant in the efforts to build a classifier based on chemical structures. Are all molecules causing one phenotypic cluster similar/dissimilar in structure? Do they share any chemical group/property (e.g. structure, domains, etc..)? Figure S3a looks at the similarity between seen and unseen drugs in the classifier but does not compare the general chemical structures in the drug list.
- What is the inter cluster vs intra cluster chemical structure similarity? Do the tested molecules cluster based on their chemical structure?

2) Screening design:

- The authors should test different concentrations for the drugs they use. This reviewer understands well that this won't be possible for all compounds, but using a fixed concentration for all drugs (10uM) limits the real assessment of their effects given that not every molecule has the same stability, processivity, and activity. The authors should titer different concentrations to understand dose-specific responses for a shortlisted number of compounds (e.g. the ones resulting in phenotypes belonging to C1, C11, and/or C12).

3) Comparison with other screening approaches, deep learning methods and relevant literature:

- A similar teratogen screen has been performed in 3D human gastruloids, which offer a more physiological and morphologically relevant model (PMID: 34425190). This work uses a way shorter list of compounds but various drug concentrations and combines morphometric analysis and gene expression patterning. Given that many molecules are present in both studies, the authors should compare their observed phenotypes in 2D in relation to the one in 3D. Also, if some molecules that have been used in PMID: 34425190 are not present in the authors' list, they should predict the phenotype using FATE-MAP and compare the prediction with the observed phenotype in 3D.
- Another relevant study has been recently published (PMID: 40245869) combining morphological variation in 3D mouse trunk-like-structures with gene expression profiles to train a classifier able to predict phenotypic outcomes from early morphological features. Even if this reviewer acknowledges that the experimental design is different (this work uses longitudinal imaging over time to identify predictive features), the authors should cite this work and comment on similarities and differences in their approaches.
- A recent study in engineered mouse 3D gastruloids (PMID: 39358450) provides temporal recording measurements of signaling patterns and specifically studies the emergence of Wnt domains and its relationship to Nodal. The authors should comment and relate their results to the ones reported in this work, since this comparison could provide species-specific and shared features between mouse and human.

4) Use of the term "morphospace":

- The authors use the term "morphospace" to define the landscape of phenotypic outcomes arising in the treated 2D micropatterns. Even if several colleagues in the field have argued about the morphological organization of micropatterns, including the 2.5D nature of the model with a z dimension also being relevant for proper cellular organization (which the authors do not use in their analysis), this reviewer thinks that 2D gastruloids lack morphological features of an embryo. The model is great to study tissue patterning, cell interactions and the effect of molecules on these features, but not so great to assess the acquisition of complex embryonic morphologies. I would recommend converting the wording "morphospace" into a "patterning space" (or similar) since this is what the model (and the readout used by the authors) can accurately inform.

5) Predictive power:

- The screening experiments performed by the authors offer the unique opportunity to inspect what was not "well predictive" (e.g inter cluster violations between prediction and observation) which could potentially result in unexpected, interesting observations/exceptions to follow up with. Did the authors observe any of these violations in their data?
- In line with what this reviewer commented on before, are there any features of the molecules' chemical structures with predictive power? Finding chemical similarities that results in similar phenotypic outcomes might be a key and novel aspect of the study.

6) Cell death vs differentiation defects

- Cell death measurements are not included in the experiments. For example, C11 (failure to express BRA) could result from cells failing to differentiate (as hypothesized by the authors) or compound lethality (which could still result in a similar phenotype). To pinpoint a real example, there is a steep difference between Cerdulatinib treatment (looks more like a lethal phenotype) and D4476, but both end up in C11. The authors should classify C11 further and pinpoint lethality vs differentiation failure phenotypes. Ideally, for a set of compounds, a live/dead dye should be used to determine the cause of the observed failure mode. This is extremely relevant to correctly interpret the phenotypic outcome and failure mode.
- Cell density vs BRA expression: this relationship is established at the endpoint differentiation. It is known that exit from pluripotency to the PS is accompanied by cell death (and therefore might result in lower density). Is this hypothesis tested against the initial cell density, or the observed density reduction could be a consequence of enhanced differentiation?

7) FDA drugs effect prediction:

- This reviewer understands well the challenges associated with implementing the Pregnancy and Lactation Labeling Rule (PLLR) for FDA approved drugs given their non categorical classification, but the categorization used by the authors dates back from 1970s and lack rigorous testing. As specified by the authority and clinicians, this categorization is not a "grading system of risk" based on the actual severity of the teratogen, but rather a classification based on available data at the time. The authors should compare/consider including information from PLLR in their analysis. This could even improve their results since the current classification might suffer from lack of extensive testing (see the next two points regarding the A, B and X graded molecules).
- On a similar line, the authors should show the embedding (or the statistical distance to clusters) for all FDA drugs, including

class A and B compounds (not shown in the current version).

- Where are all the other class X compounds ending up (71 out of 98 class X compounds are not shown). Are they part of the other aberrant clusters or do they end up in the canonical "morphospace"?

- The authors should also comment on the known phenotypes (at the cellular and organismal levels) associated with Ulixertinib and Nirogacestat in other cellular systems or animals. Do their predicted and observed phenotypes relate to the literature knowledge of these molecules and to their molecular functions (ERK 1/2 and γ -secretase inhibitors, respectively)?

8) Zebrafish experiments:

- This is, according to this reviewer, the most important technical aspect the authors should clarify. Based on both the observed and predicted phenotypes, both Ulixertinib and Nirogacestat treatments result in phenotypic cluster C11, where micropatterns fail to undergo gastrulation and to induce the expression of BRA, consequently lacking the primitive streak (PS). The timing of the treatment in zebrafish embryos is correct, matching the epiblast stage in human micropatterns. The obtained phenotype though does not seem to be entirely compatible with the lack of PS induction observed in C11 micropatterns. If it is true that the exposed animals have a short-tail phenotype, this does not resemble a scenario where BRA (ntl in fish) is not expressed (see PMID: 8402905). In fact, especially for Ulixertinib, axial elongation is still present and somitic rows and other PS derivatives are visible. This might have various sources of explanation, but the authors should clarify why. The authors could also check earlier stages (e.g. late gastrula) to assess the initial formation of early mesoderm (which should lack or be aberrant if the drug induces a failure in PS induction). Overall, the teratogen classification is correct, but the phenotypic prediction is not accurate (or may vary between species).

9) WNT diffusion modeling

- WNT diffusion as a function of tissue density is a very interesting hypothesis and might explain the evolutionary need for a correct PS size during gastrulation. This reviewer's doubts related to cell death during differentiation (see comment above) remains in this assay as well, and this potentially confounding factor should be ruled out before interpreting the phenotype only on the ability of WNT to diffuse (especially because the lower diffusion of WNT was imposed in the mathematical model).

- Density correlates with WNT waves: this represents the main finding of the manuscript, which also has the potential to inform some basic mechanism during early development. It is indeed an interesting observation, though not completely unexpected. The effect of structure sizes on WNT response has been studied before, also in 3D mouse gastruloids (PMID: 38491138). This work highlighted how varying the size of the initial aggregate does not result in differences in the PS-like domain induction (BRA, CDX, SOX2 patterning). The gastruloid protocol is entirely based on WNT induction using CHIR (small molecule). It could be that the difference in results observed by the two studies relates to the fact that CHIR permeability/diffusion might follow different rules than the WNT itself, being less sensitive to cell density/aggregate size. The authors should compare their results with these and/or try to model WNT vs CHIR diffusion.

Minor comments:

- The authors focus extensively on the undifferentiated SOX2+ and PS-like BRA+ domains and less on the trophectoderm-like GATA3+/CDX2+ outer ring. Are there molecules that cause specific aberrations in this domain (e.g. its expansion or complete loss)?

- The authors do not check for endoderm induction, a domain observed in the original protocol (PMID: 24973948). While this reviewer understands the challenges in staining for an extra marker, the authors should comment on the expected outcomes or acknowledge this as a possible limitation of the study (apparently canonical phenotypes might mask defects in endoderm induction which were not measured in this study).

- The authors should provide high-resolution imaging of some (or all in supplementary figures or repositories) conditions since the micropattern digital imaging panels are very hard to inspect.

- The link to access the entire morphospace dataset seems to be broken and I could not inspect that closely (lines 199-200).

- Fig S4 e-h are barely visible and with low image quality, making them quite hard to read.

- The overall image quality is low and does not allow detailed inspection of the expression patterns and phenotypes. The authors should upload high-resolution figures.

- Despite being published in ref 45, the authors should detail how the membrane bound β -catenin signal was masked in the live cell imaging experiments, since no dyes to label cell membranes were used (at least not specified in the methods nor in the legends). Clarification is needed given that this is a non-trivial segmentation.

- Lines 512-513: the use of the term "malformation" to describe an aberrant phenotypic pattern is not ideal. I would recommend using "aberrant gene expression patterns" or similar expressions.

- The name of the tool FATE-MAP is a bit misleading, since the current study mainly provides information about the ability to exit from pluripotency (one transition towards nascent mesoderm) and does not go beyond that measurement into fate decision-making.

(Remarks on code availability)

Version 1:

Reviewer comments:

Reviewer #1

(Remarks to the Author)

The authors have satisfactorily addressed all my concerns. I believe the MS is now ready for publication.

(Remarks on code availability)

Reviewer #2

(Remarks to the Author)

The authors have sufficiently addressed nearly all my concerns.

My only remaining major concern is the modeling of the density dependence of Wnt. In the first round of feedback, I mentioned their model appears to predict a delay in signaling that is not visible in the data. In their reply, the authors argue this is a matter of normalization of the data and changed the normalization. However, looking at the new normalization (Fig.R14), there is still a qualitative difference between simulation and experiment, now in the amplitude: while simulations all approach the same maximal signaling level over different time scales, the amplitudes are clearly density dependent in the data. Overall, it seems to me the disagreement can't be normalized away and should be discussed in the manuscript.

I also had a few more very minor comments:

- In Figure R8, the different blues for the two different BMP4 groups become indistinguishable when I print it.
- Why did the authors choose not to show error bars on radial profiles in fig 1?
- The authors state in their rebuttal that GATA3 follows rapid switch-like dynamics citing Gunne-Braden et al, however this paper disagrees with a large body of literature. Teague et al Nat Comm 2024 observed different GATA3 dynamics and discuss the disagreement.
- In the revised discussion on ROCK inhibition the authors cite Maldonado et al. In the same place they should also cite Etoc et al 2016 since they used ROCK inhibitor to disrupt tight junctions and increase differentiation as discussed.

(Remarks on code availability)

Reviewer #3

(Remarks to the Author)

The authors have done a very good job to address all the concerns raised. This reviewer thanks them for their great effort.

This reviewer still thinks that the translational aspect of the tool prediction should be less emphasized and acknowledged as a limitation of the study and further point of improvement in the Discussion section. This comment comes mainly from the results of the zebrafish experiments (where the predictive power of the tool might not be directly translatable in other systems or real embryos) and from the inaccuracy in the prediction of a subset of compounds (including very well-known teratogens).

Apart from this comment, this reviewer supports the publication of the study without further revision.

(Remarks on code availability)

/

Reviewer 1 (Remarks to the Author)

In this work, the authors undertake a formidable effort to screen and quantify the effects of 210 compounds on the development and differentiation of 2D gastruloids from hPSCs, a system well established, primarily by the Warmflash lab. They quantify three fluorescent markers in 10 colonies per treatment, ultimately quantifying images from over 2,000 gastruloids at the single-cell level, a remarkable scale for this kind of analysis. The fluorescence quantification is converted to a vector describing the radial intensity profile for each marker.

Comment 1: The idea of generating a mean behaviour image for each cluster is nice, seems to work well in these radially symmetric colonies, and provides an intuitive understanding of each response type. The authors cluster the gastruloid vectors into 12 clusters, assigning 9 of them to normal behaviour and 3 clusters to distinct failure modes (loss of Bra/Sox2 or loss of radial symmetry). Note that converting a profile with a geometric meaning (expression along the radial axis) into a vector (simply by binning) loses all geometric information. For example, under standard metrics the vector 1000 will be equidistant from 01000 and from 00001, while in its geometric interpretation it is clearly closer to the first one. All the same, the clusters obtained seem to be meaningful. The authors may want to discuss this point, and perhaps point to limitations of this embedding. Of course, when we move to network learning in the next step, the network model can re-learn the geometric connections between output nodes.

Response 1: We agree that a Euclidean distance on binned vectors does not, by itself, encode geometric information (e.g., a peak at position 1 is Euclidean-equidistant from peaks at positions 2 and N). Importantly, our embedding retains the ordered radial sequence, so adjacent positions remain adjacent in the representation. Figure R1 compares (a) Euclidean distance between mean drug-treatment profiles, which separates drug conditions by their average radial patterns, and (b) mean pairwise cosine similarity across all individual drug treated colonies, which emphasizes pattern shape and provides finer separation when treatments produce more heterogeneous profiles. Consistent with the t-SNE embedding, which preserves local neighborhoods rather than absolute distances, the cosine similarity shows tight within-treatment neighborhoods and low similarity across treatments, while the Euclidean distances highlight separation between treatment means that induce distinct global patterns. We revised the manuscript to explicitly acknowledge this limitation and to explain that, while distances are not geometry-aware, the ordered radial sequence is retained.

Changes to the manuscript:

“Although this embedding does not explicitly preserve geometric distances between adjacent bins, it preserves the exact radial ordering, enabling spatially coherent comparison of radial expression patterns across colonies (Supplementary Fig. 1b-c).”

Fig. R1 has been added as Supplementary Fig. 1b-c.

Figure R1: Similarity of ordered radial profiles across treatments. (a) Euclidean distance between mean control drug-treatment profiles. (b) Mean pairwise cosine similarity between all control colonies (higher = more similar; color bar inverted). Treatments: mTeSR1, BMP4, CHIR-98014.

Comment 2: In the next step, the authors encode the 210 chemical compounds as vectors, and train a network model to predict the gastruloid vector (output) from this encoded input. They also train an additional (smaller) network to embed the predicted vector in tSNE “morphospace embedding”. I assume the prediction errors were done on the original 150-dim vector space, correct? Why is it important to embed into the 2-dim tSNE space? It is only used for visualization purposes. As the name says, it is only a 2-dim embedding of a 150-dim vector, which itself is a representation of a 2-d circular pattern. If the purpose is mostly to evaluate the error (as depicted in Fig. 2b) for the 150-dim space, the embedding process only adds some more distortion to it. It is more informative to show some quantification of the prediction errors (as depicted in the red/green/blue plots), how they behave in training vs. test set, etc.

Response 2: We agree with the reviewer that quantitative error should be evaluated in the original 150-dimensional phenotype space. Accordingly, we added *Supplementary Table 3* reporting RMSE and cosine similarity between predicted and experimental phenotypes for both training and test sets, which provides a rigorous assessment of prediction accuracy. However, learning the 150-dimensional phenotype vector before projecting into morphospace is essential for two conceptual and practical reasons:

- 1. Biological interpretability and modularity (clarifying the reviewer’s “distortion” point).** The 2D morphospace embedding is not used to evaluate error; all quantitative comparisons are performed in 150D. Its purpose is to co-locate experimental colonies, AI-predicted phenotypes, and PDE-based mechanistic simulations on a shared manifold where developmental trajectories can be visualized and interpreted (as in Fig. 6c). Because any projection can introduce distortion, we explicitly quantified the incremental distortion from projecting to 2D: the learned (parametric) embedding closely reproduces the original t-

SNE density field ($JSD \approx 0.09$; Supplementary Fig. 2c). Thus, while accuracy is judged in the high-dimensional space, the 2D projection enables direct visualization of how compounds shift along biologically interpretable axes (e.g., cell density, SOX2 stability) that emerge from our mechanistic simulations.

- 2. Preservation of nonlinear developmental structure.** Predicting the full 150-dimensional phenotype before embedding ensures the model captures biologically meaningful relationships between individual fate markers rather than overfitting to the arbitrary geometry of a particular 2D t-SNE map. The intermediate phenotype vector anchors predictions to measurable biological quantities: radial expression of GATA3, BRA, and SOX2, providing an interpretable format shared with our PDE simulations. Embedding those phenotypes afterward preserves the nonlinear topology of developmental trajectories and enables direct comparison of experimental, simulated, and predicted outcomes within a single interpretable coordinate system. This two-step approach ensures that the morphospace remains stable to retraining and extensible to new experimental data or mechanistic simulations.

Changes to the manuscript:

Added Supplementary Table 3 which reports the RMSE and cosine similarity between all chemical structure-to-phenotype predictions and experimental results for both the training and test sets.

Comment 3: In the sample images in Fig. 3a we see some bald regions within the colony (which we do not see in the examples in Fig. 2. What are these? Why do we only see them here?

Response 3: We thank the reviewer for this observation. The “bald” regions visible in some colonies in Fig. 3a are not actual gaps in the cell layer, but rather out-of-focus vertical protrusions. These occur when certain drug treatments cause 3 dimensional cell growth out of the plane of focus of the confocal stack imaged for each colony. This phenomenon is particularly common in retinoids, which have been shown to promote stem cell proliferation.^{1,2} In the primary high-throughput screen, we acquired shallow z-stacks to keep scan time and data volume tractable across >2,000 colonies; full 3D stacks would have dramatically increased acquisition time and produced many blank z-slices for otherwise flat monolayers.

Fig. R2 shows three representative retinoid-treated colonies alongside their radial intensity profiles. The profiles clearly demonstrate that these protrusions are captured as a drop in SOX2 signal in the center of the colony. Nevertheless, these colonies still fall within Cluster 11, which is characterized by a lack of BRA expression. It should also be noted that the radial binning procedure partially mitigates this effect in some colonies, whereas in others the impact on the radial intensity profile is more pronounced. In Fig. R2d, we performed new imaging of tretinoin-treated colonies with the axial range increased to 20 μm ; this resolves the apparent “holes” by capturing the out-of-plane cells while preserving the same BRA-loss phenotype

Figure R2: Representative retinoid-treated colonies exhibiting vertical protrusions due to increased cell density. Example colonies treated with (a) isotretinoin, (b) TTNPB, and (c) bexarotene show lighter central regions that are not gaps in the cell layer but vertical protrusions arising from confinement within the micropattern. Corresponding normalized radial intensity profiles demonstrate loss of mesodermal identity (BRA, red), consistent with Cluster 11 phenotypes. (d) Tretinoin-treated colonies imaged with an increased focal range (20 μm) show filled centers, validating that the “bald” appearance reflects out-of-plane growth rather than absence of cells.

Changes to the manuscript:

“These colonies frequently displayed lighter central regions that reflect vertical protrusions caused by increased cell density under micropattern confinement (Supplementary Fig. 6)”

Fig. R2 has been added as Supplementary Fig. 6.

Comment 4: The authors then run 947 FDA approved drugs through their prediction pipeline, finding 100 to fall within the no-Brachyury cluster. The association statistics corresponds with pre-known risk levels (as teratogens) of those drugs. A few comments on this approach/result:

1. Looking at Fig. 3c, it is not clear how association of the new compounds to cluster 11 was determined. Some of them (including several category X compounds) seem just as close (or closer) to clusters 8 and 9. Or is it because the association was computed in the 150-d space, and the 2-d projection is just misleading? The same is true for Ulixertinib prediction.

- In this specific example, the association to cluster 11 describes failure in the form of no Brachyury expression. Did we have to go through the sophisticated experimental model (2D gastruloid) and heavy computational framework? Wouldn't we get a similar (or better) prediction simply screening Bra-expression levels (i.e. a 1d readout instead of 150d readout) in e.g. simple diff protocol applied to a 2D culture? It is worth at least discussing this point.

Response 4:

Point 1: Previously, in the original submission, cluster 11 membership for predicted compounds was determined using a manually defined linear decision boundary in morphospace (Fig. R3a). As the reviewer noted, 2-dimensional t-SNE embeddings can visually make some compounds appear close to multiple clusters. To address this, we have replaced the manual boundary with an objective, data-driven assignment method that uses the same clustering algorithm originally used to identify clusters from the experimental dataset.

Figure R3: Classification of cluster 11 compounds using a fixed linear decision boundary in morphospace. (a) Original linear classification boundary for cluster 11. (b) Revised morphospace boundaries computed using KDE + watershed clustering, applied consistently to experimental and predicted phenotypes. (c) Benchmarking against 33 FDA-approved compounds previously tested

in TeraTox and mEST assays. (d) Ulixertinib and nirogacestat remain classified in cluster 11 under the revised method.

In brief, the original clustering pipeline first computes a kernel density estimate over the t-SNE space, detects local density maxima, and then applies watershed segmentation to partition morphospace into discrete cluster regions (see Methods). In the revised analysis, we apply these same watershed-defined morphospace boundaries to all predicted drug phenotypes, ensuring that experimental and predicted compounds are classified consistently and without any ad hoc criteria (Fig. R3b). This revision removes subjectivity and directly ties cluster assignment to the same unsupervised morphological structure learned from the experimental dataset. Corresponding language has been added to the Results and Methods sections.

The analysis comparing cluster 11 to FDA pregnancy-risk label categories has also been revised. While FDA labeling provides clinical guidance, it is influenced by heterogeneous data sources, risk communication practices, and historical regulatory decisions, making it an imperfect ground truth for mechanistically defined teratogenic potential. In the revised manuscript, we have removed the FDA-label analysis and instead benchmark predictions against a curated panel of 33 drugs with known teratogenicity potential, supported by either human data (as reported in FDA drug labeling) or *in vivo* embryo–fetal development (EFD) studies in rats and/or rabbits. Crucially, these 33 drugs have also been tested in 2 different stem-cell-based assays for teratogenicity assessment: TeraTox (3D iPSC multilineage differentiation) and the mouse embryonic stem cell test (mEST). Only 4 of these compounds were present in our experimental dataset; the remaining 29 were classified fully *in silico* using FATE-MAP.

Predicted 150-dimensional phenotypes from 33 compounds were embedded into morphospace and classified using the fixed watershed-defined cluster boundaries derived from the experimental dataset (Fig. R3c). Embeddings falling within failure-mode clusters (C1, C11, C12) are assigned positive teratogen status, whereas embeddings within the canonical patterning region (C2–C10) are assigned non-teratogenic status. Across this set, FATE-MAP achieved 72.7% accuracy, outperforming the mEST assay (51.5%) and matching the TeraTox assay (72.7%) while exhibiting higher balanced accuracy than TeraTox (74.8% vs. 66.3%). These results illustrate the potential of FATE-MAP as a scalable screening framework, particularly because our approach does not require experimentally testing each compound, enabling scalable *in silico* teratogenicity prediction.

Finally, the two compounds selected for follow-up zebrafish validation, ulixertinib and nirogacestat, are still robustly classified within cluster 11 under the revised clustering procedure (Fig. R3d, Revised Fig. 3), confirming that the conclusions of the study are unchanged by the improved classification approach.

Changes to the manuscript:

Revised manuscript to include benchmarking of FATE-MAP predictions against 33 FDA-approved compounds with experimentally established teratogenicity outcomes in stem-cell-based assays.

Fig. R3c-d has been added as Revised Fig. 3c and Fig. 3e.

“Of note, drugs were assigned to clusters not by visual inspection of their position in morphospace, which can be ambiguous, but using the same watershed-derived morphospace boundaries defined during the original clustering, ensuring consistent, data-driven classification across experimental and predicted phenotypes.”

Supplementary Note 1 has been revised to clearly explain the cluster classification process.

Point 2: This is a good point, and if our sole goal was teratogen identification, a 2D culture differentiation protocol would be much simpler to implement. In fact, prior work has demonstrated monolayer-directed differentiation of hESCs for teratogen detection, on 71 drug-like compounds with known *in vivo* effects.³ Rather than using BRA as a marker, the authors used the absence of the mesendodermal marker SOX17 as the readout. However, while highly effective for classification, this single-marker framework does not provide insight into *how* developmental signaling pathways are perturbed, nor does it offer a systematic way to resolve diverse failure modes.

In contrast, FATE-MAP was designed not only as a teratogen detection platform but as a framework to understand how chemical perturbations reshape developmental trajectories. There are several key differences between our approach and the monolayer system:

➤ **Controlled starting conditions via micropatterning**

FATE-MAP uses micropatterned colonies, which provide reproducible starting conditions and well-defined boundary constraints. This level of control was essential to our discovery of the role of cell density in modulating WNT signaling and mesoderm formation. In the monolayer study, boundary conditions were not controlled, and the authors themselves note that cultures contained both “dense” and “sparse” regions by day 3. They further observed that fate specification varied drastically between these regions, with high-density areas often staining positive for mesendodermal markers and sparse regions lacking them. While this heterogeneity did not undermine their assay because of the highly sensitive marker chosen (SOX17), it limits the ability to extract mechanistic insights about the contribution of density or other developmental parameters.

➤ **Robustness across markers**

The authors of the monolayer study also acknowledge that “in our hands, the expression levels of T-brachyury varied greatly among cell populations within a single well,” highlighting the variability of monoculture-based platforms. By contrast, FATE-MAP leverages multiparametric readouts from three lineage markers (BRA, SOX2, and GATA3)

and encodes their spatial distributions, allowing us to resolve failure modes beyond the absence of a single marker.

➤ **Embryo-relevance and morphological context**

Unlike monolayer differentiation, 2D gastruloids are a validated *in vitro* model of the gastrulating human embryo, displaying the correct radial organization of germ layers and recapitulating single-cell RNA-sequencing signatures that map to *in vivo* human gastrulation. This morphological and transcriptional fidelity provides biological context for interpreting compound effects, allowing us to ask how perturbations alter embryonic patterning itself. By contrast, monolayer assays, although highly effective for teratogen classification, lack morphological significance and thus cannot inform how compounds impact the spatial organization of developmental programs.

Nevertheless, we thank the reviewer for raising this point. The monolayer-directed differentiation approach was not discussed in our initial manuscript and represents an important contribution to the field. We also acknowledge that its simplicity in both experimental setup and single-marker readout is compelling for applications focused purely on teratogen classification. However, our goal with FATE-MAP is broader: to develop a mechanistically interpretable platform capable of resolving diverse failure modes and revealing how drug perturbations reshape developmental signaling pathways. To highlight this distinction, we have added a discussion of the monolayer approach to the revised manuscript.

Changes to the manuscript:

“In parallel, monolayer-directed differentiation protocols have also been implemented for teratogen detection, achieving high classification accuracy from single-marker readouts.⁵⁷ While compelling in their simplicity, these assays lack morphological context and mechanistic resolution, underscoring the added value of FATE-MAP in resolving developmental failure modes.”

Comment 5: The authors then devise a mechanistic model, that models the spatio-temporal dynamics of Wnt, BMP and Nodal activity, and tie it with the measureable marker expressions (that serve as a proxy for cell fate) through a simple logic-based model. They demonstrate how such a model can predict the spatial patterns observed, as well as their dependence on cell density. They experimentally validate one implication from the model simulations, which is the inverse dependence of Wnt signaling on cell density.

They then parametrize morphospace by projecting gastruloids predicted by the mechanistic model while varying two parameters: cell density and SOX2 stability.

1. It is not clear (or well motivated) why SOX2 stability was chosen – this is not a parameter that can be directly controlled experimentally, and it is not clear to what degree it naturally varies between the gastruloids.

2. The authors claim that Sox2 stability (C_s) “emerged as a key regulator of fate boundary positions in our model.” This claim is a bit misleading – the model was devised in a way that gives C_s a high weight, determining the resulting patterns. For example, the stabilities of Gata3 or Brachyury do not appear as parameters in the model at all. This should be better explained and motivated.

Response 5: We thank the reviewer for raising this important point. To address this, we expanded our analysis to systematically vary all parameters (including the addition of parameters such as GATA3 stability and BRA stability) in our cell fate rules across biologically plausible ranges. A schematic of the expanded rules (Fig. R4a) and a table of parameter definitions (Fig. R4b) are provided below. Figure R4c–d shows the results from our initial manuscript when SOX2 stability (C_s) was modulated across multiple cell densities, producing a set of trajectories that spanned orthogonal axes of variation in morphospace. As noted by the reviewer, C_s is not directly controllable experimentally. However, it serves as a mechanistically interpretable parameter that scales the persistence of SOX2 against competing lineage programs (GATA3 and BRA). This parameter effectively represents the balance between pluripotency maintenance and lineage commitment over time, and thus provides a natural axis for linking drug-induced shifts to signaling perturbations. While not directly measurable, C_s likely reflects integrated regulatory processes such as post-translational modifications (e.g., phosphorylation, methylation) and chromatin accessibility that influence SOX2 degradation and stability.

a Potential parameters to modify cell fate

$$\text{GATA3} = C_G \left(\frac{\int \text{BMP4} dt}{1 + e^{-k_1(r-r_{loc})}} \right)$$

$$\text{BRA} = C_B \left(\int \frac{1}{1 + e^{-k_2(\text{Wnt} - \text{Wnt}_{\text{thresh}})}} \frac{\partial \text{Nodal}}{\partial t} dt \right)$$

$$\text{SOX2} = C_s (c_1 + c_2 \text{GATA3} + c_3 \text{BRA})$$

b

Parameter	Description
C_G (GATA3 stability)	Scales persistence of GATA3 (effective stability term for GATA3).
C_B (BRA stability)	Scales persistence of BRA (effective stability term for BRA).
C_s (SOX2 stability)	Scales persistence of SOX2 against repression by GATA3 and BRA.
k_1 (GATA3 steepness coeff.)	Sigmoid steepness for GATA3 radial activation.
k_2 (BRA steepness coeff.)	Sigmoid steepness for BRA activation as a function of WNT.
$\text{Wnt}_{\text{thresh}}$	WNT level required to trigger BRA induction.
c_1 (Baseline SOX2 level)	Initial/unrepressed SOX2 level.
c_2 (GATA3 repression weight)	Coefficient for GATA3-mediated repression of SOX2.
c_3 (BRA repression weight)	Coefficient for BRA-mediated repression of SOX2.

c Impact of cell density

d Impact of SOX2 stability (C_s)

Figure R4: Expanded parameterization of mechanistic cell fate model to evaluate regulatory contributions. (a) Equations defining GATA3, BRA, and SOX2 as functions of all possible stability terms, repression weights, and activation thresholds. (b) Table of model parameters with biological interpretation. (c-d) Simulated colony embeddings as a function of cell density and SOX2 stability (C_s), showing orthogonal axes of variation.

To evaluate whether this interpretability was unique to C_s , we extended the parameter sweep to include all stability terms, repression weights, and activation thresholds (Fig. R5). Interestingly, none of these other parameters produced trajectories that aligned with major axes of morphospace variation, with the partial exception of C_1 .

Figure R5: Systematic variation of all model parameters across morphospace. Simulated colony embeddings in morphospace while varying each parameter across a biologically plausible

range (0.25–1.75) and at multiple cell densities. Among all parameters tested, only C_s (Fig. R4) and cell density generate trajectories that align with the major axes of developmental variation, highlighting their interpretability for linking drug-induced shifts to developmental mechanisms.

C_1 (baseline SOX2 level) does generate a trajectory, but this reflects a trivial shift in the initial condition of the system rather than a regulatory process. Mathematically, C_1 sets $SOX2(0)$ at $t=0$ in the absence of any signaling input. While this rescaling can alter final outcomes, it does not capture the dynamic regulation of boundary positions during gastrulation-like patterning. Furthermore, all stem cells were harvested from the same culture well before being seeded into micropatterned plates, making it unlikely that they possessed significantly different baseline levels of C_1 . In contrast, C_s multiplies the entire SOX2 term, meaning it governs the effective persistence of SOX2 in the presence of antagonistic inputs (GATA3, BRA). This makes C_s a more appropriate analog for drug action, since most perturbations alter stability or turnover rates of transcriptional regulators rather than their absolute initial levels.

It is also worth noting that parameters like Wnt_{thresh} do produce effects, but these are qualitatively different: for example, lowering or raising Wnt_{thresh} shifts BRA induction to become less density-dependent, resulting in all colonies at a given density to have higher or lower levels of BRA expression, respectively. However, these changes do not project along a clear developmental axis in morphospace and thus are less useful as organizing principles.

In summary, while we acknowledge that our initial emphasis on C_s may have overstated its uniqueness, our expanded parameter sweeps support the conclusion that C_s provides the clearest axis for interpreting morphospace displacements. This choice is mechanistically plausible from a drug MOA standpoint: compounds can modulate the effective stability of SOX2 via epigenetic or post-translational control (e.g., phosphorylation creating degron motifs that recruit ubiquitin–proteasome turnover, and lysine/arginine methylation altering DNA binding, nuclear retention, and degradation kinetics). In parallel, shifts in SOX2 abundance have been shown to reconfigure WNT/ β -catenin target engagement, biasing β -catenin toward pluripotency enhancers when SOX2 is high and toward mesendodermal gene programs when SOX2 is reduced, thereby switching the net output of WNT signaling (ref. 71).

We appreciate the reviewer’s point and have tempered our language accordingly; we now present as a mechanistically interpretable axis rather than a unique determinant. We have clarified this rationale in the revised text and included the expanded parameter sweep in the revised Supplementary Information.

Changes to the manuscript:

Added Figs. R4 and R5 as the revised Supplementary Fig. 11 and Supplementary Fig. 12.

“To assess whether other coefficients could generate comparable trajectories, we systematically swept all model parameters over biologically plausible ranges; only density and SOX2 stability consistently produced extended, coherent axes in morphospace (Supplementary Figs. 11-12)”

Abstract: Changed “Using FATE-MAP, we flagged two clinical molecules as potential teratogens and also revealed cell density and SOX2 stability as determinants of gastruloid patterning” to “Applying this framework, we flagged two clinical molecules as potential teratogens and identified two parameters, cell density and SOX2 stability, that form orthogonal morphospace axes along which canonically patterned gastruloids systematically vary.”

Revised section header “Parameterizing gastruloid morphospace: cell density and SOX2 stability define developmental axes” to “Parameterizing gastruloid morphospace: cell density and SOX2 stability provide interpretable axes”.

Changed “Although not directly measurable, this parameter (SOX2 stability) emerged as a key regulator of fate boundary positions in our model” to “Although not directly measurable, this parameter provides a mechanistically interpretable readout that broadly reflects drug mechanism and influences a colony’s position within morphospace”.

Changed “The resulting structure suggests that morphospace can be decomposed into two interpretable axes defined by these parameters” to “The resulting structure reveals two interpretable axes aligned with these parameters, while not precluding additional contributing factors.”

Changed “This integration facilitates the identification of key signaling parameters, such as cell density-modulated Wnt signaling and SOX2 stability, that organize developmental trajectories” to “This integration facilitates the identification of key signaling parameters, such as cell density-modulated Wnt signaling and SOX2 stability, that help organize developmental trajectories.”

Comment 6: The idea of parametrizing the abstract UMAP embedding by varying interpretable parameters is nice, as it adds to the interpretability of the morphospace.

Response 6: We thank the reviewer for this positive comment.

Comment 7: The authors present the AI and mechanistic models as if they were two alternatives to choose from. This is a bit misleading, as the mechanistic model does not incorporate the potential effects of teratogens. If I understand the rationale of the authors, in integrating (and comparing) the two, they assume that the effect of the teratogens can be summed up through their effect on just 2 parameters (cell density and SOX2 stability).

1. What is the rationale for that?
2. Are the effects of the compounds on density simply due to toxicity? (this could be cross-checked against simple growth curves in 2D cell culture for these compounds).

- And if they affect Sox2 stability (as rationalized in the Discussion), don't they have similar effects on GATA3 and Brachyury? Or on Wnt, Nodal and BMP proteins?

Response 7:

Point 1: We do not assume that the effects of all teratogens can be fully captured by modulating just two parameters. Rather, our key insight is that canonical patterning outcomes, those in which the spatial organization of germ layers remains intact, can be largely explained by variation along two axes: cell density and SOX2 stability. These parameters reflect continuous, tunable aspects of developmental regulation that define the structure of the canonical morphospace (clusters C2-C10).

In contrast, teratogenic compounds fundamentally change the topology of morphospace, pushing colonies out of the canonical density–SOX2 stability continuum and into failure-mode clusters (C1, C11, or C12). Such drugs perturb the rules that underlie pattern formation. For example, they can alter morphogen production or transmission (for example, by directly inhibiting BMP, Wnt, or NODAL). They can disrupt cell–cell contacts that regulate boundary formation and receptor localization, including effects secondary to drug-induced cell loss. They can also rewire fate-specification rules (for example, by changing competence thresholds or the cross-repression between SOX2, GATA3, and BRA through epigenetic or post-translational mechanisms) so that identical inputs are decoded into different fates.

To further illustrate this point, we selected two morphogen-pathway inhibitors from our screen, WIKI4 and LY3200882, that produced colonies in cluster 11 for follow-up analysis. WIKI4, a tankyrase inhibitor that suppresses WNT/ β -catenin,⁴ was modeled by eliminating WNT auto-activation (setting $s_u = 0$). LY3200882, a selective ALK5 (TGF- β RI) inhibitor that attenuates the SMAD2/3 arm shared by TGF- β /Activin/Nodal,⁵ was modeled by eliminating the NODAL/Activin production term (setting $s_v = 0$). With all cell-fate rules unchanged, both perturbations yielded BRA-null simulated colonies that aligned with the cluster 11 experimental phenotypes at comparable densities (Fig. R6).

Figure R6: Mechanistic modeling of teratogen-induced failure modes. (a)–(b) WNT inhibition modeled by eliminating WNT auto-activation ($s_u = 0$) recapitulates the effect of WIKI4, yielding BRA-null colonies consistent with experimental cluster 11 phenotypes. (c)–(d) NODAL inhibition modeled by eliminating Activin/NODAL production ($s_v = 0$) reproduces the effect of LY3200882, likewise producing BRA-null colonies that align with experimental cluster 11 outcomes.

These examples help illustrate that the mechanistic modeling framework is not a universal model of drug action. Rather, it provides a compact description of canonical variation and, when supplied with specific mechanistic hypotheses (e.g., WNT or ALK5/SMAD2/3 inhibition), can also reproduce failure modes, though this requires prior knowledge of pathway targeting.

Overall, the mechanistic model codifies the rules that govern the structured, canonical region of morphospace; it tells us that when a compound drives colonies outside this region, it is operating under a different set of rules. The AI model, in turn, learns patterning directly from chemical structure, predicting where a compound will land in morphospace. Together, the two are complementary: the AI model flags and localizes compound effects, and the mechanistic model interprets whether those effects reflect canonical displacements (density/ C_s) or topology-altering failures, yielding insight into how a teratogen may act (i.e., inhibition of Wnt signaling). In the revised manuscript, we have rewritten the relevant section in the Results section to articulate this distinction more clearly.

Changes to the manuscript:

“While our AI-based structure-to-phenotype model predicts 2D gastruloid phenotypes with high accuracy (Fig. 6a), it functions as a black box and does not directly reveal which developmental processes are perturbed. To provide mechanistic insight, we developed a complementary model grounded in morphogen diffusion and fate transitions (Fig. 6b). By projecting simulated phenotypes into the same learned manifold as experimental data, this model assigns interpretable structure to morphospace and enables causal reasoning about how perturbations modulate underlying parameters. In practice, the mechanistic model codifies the rules that govern the structured, canonical region of morphospace and, when supplied with prior mechanistic hypotheses (e.g., morphogen inhibition; Supplementary Fig. 10c), can also reproduce specific failure modes. The AI model, in turn, predicts where a compound will land in morphospace directly from chemical structure. Together, the two are complementary: the AI localizes compound effects, and the mechanistic model interprets whether they reflect canonical displacements or topology-altering failures.”

We have added Fig. R6d to Supplementary Fig. 10c.

Point 2: We appreciate the reviewer’s thoughtful comment. A central finding of this work is that cell density defines a continuous axis of morphospace variation, influencing fate patterning through biophysical and signaling-mediated mechanisms rather than simply reflecting toxicity. This relationship was first observed from the embedding of experimental colonies (Supplementary

Fig. 9), where the slope and fit of the BMP4-control trendlines, together with mean cluster representations, revealed systematic, density-dependent shifts within the canonical patterning region. We then developed and validated a density-dependent model of Wnt signaling (Fig. 4) and subsequently identified cell density as a principal morphospace axis (Fig. 5).

While conventional 2D growth-curve assays could, in principle, provide complementary information about toxicity, our screen captures this in a physiologically relevant format that reflects edge-localized receptor dynamics and BMP→Wnt→Nodal feedbacks unique to the 2D gastruloid model. By counting every nucleus within each colony, we directly quantify cell number at single-colony resolution and can identify gross reductions in viable cell density without assuming a mechanism. Because all wells start at the same seeding density (50,000 cells per well), large drops in final nuclear counts primarily reflect drug-induced toxicity and exceed what would be expected from growth arrest alone.

Figure R7: Density-based toxicity threshold. (a) Mean nuclei per colony for each BMP4+drug condition. Toxicity threshold (red, $\leq 50\%$ mTeSR mean) and hyperproliferation threshold (green,

>mTeSR mean). (b) Toxic compounds map predominantly to failure modes C1, C11, and C12; insets show representative colonies. The embedding illustrates that overt cytotoxic collapse can be distinguished from density-dependent modulation within the canonical patterning region.

To objectively define toxicity, we used a control-anchored cutoff: treatments with mean nuclear counts <50% of the mTeSR control mean were classified as toxic. As shown in Figure R7a, seven compounds met this criterion, while seven others exceeded the mTeSR1 mean and formed a high-density (hyperproliferative) group. Notably, 96.1% of colonies exposed to toxic compounds mapped to failure-mode clusters C1, C11, and C12 (Fig. R7b), demonstrating that our assay distinguishes toxicity-driven collapse from density-dependent modulation of canonical patterning. These analyses show that while drug-specific perturbations can alter proliferation or morphogenesis, the underlying patterning program remains remarkably robust, preserving canonical organization across a broad range of cell densities.

Changes to the manuscript:

Added Fig. R7 as Supplementary Fig. 3.

“Because failure modes can arise from either differentiation failure or compound-induced cytotoxicity, we quantified total nuclear counts for every colony and identified a small subset of drugs that reduce viable cell number below 50% of the mTeSR-only control (Supplementary Fig. 3). This analysis separates genuine patterning failures from cytotoxicity-driven collapse within the failure-mode regions.”

Point 3: We agree that drug perturbations are unlikely to act exclusively on SOX2. Indeed, in our expanded parameter sweeps (Fig. R4-5; Supplementary Fig. 10-11), we systematically varied all model coefficients, including GATA3 stability, BRA stability, and morphogen thresholds. Some of these perturbations did shift colonies within morphospace; however, none produced smooth, monotonic trajectories that traversed the canonical region of morphospace.

This suggests that while drugs may act at many levels (e.g., modulating WNT, NODAL, or BMP production, or altering GATA3/BRA turnover), their downstream effects converge on parameters that influence the persistence of SOX2 relative to competing fates. In our framework, SOX2 stability (C_s) captures this convergence: it reflects the effective stability of SOX2 expression in the presence of GATA3 and BRA, integrating diverse upstream influences into a single, interpretable readout.

While further work will be needed to dissect the precise molecular mechanisms by which specific compounds alter SOX2 stability, the fact that a single parameter can organize heterogeneous perturbations within morphospace is powerful for interpreting developmental outcomes. In the revised manuscript, we temper claims of uniqueness and explicitly acknowledge that SOX2 stability serves as a useful interpretable axis rather than an exclusive determinant.

Changes to the manuscript:

Abstract: Changed “Using FATE-MAP, we flagged two clinical molecules as potential teratogens and also revealed cell density and SOX2 stability as determinants of gastruloid patterning” to “Using FATE-MAP, we flagged two clinical molecules as potential teratogens and identified two parameters, cell density and SOX2 stability, that form orthogonal morphospace axes along which canonically patterned gastruloids systematically vary.”

Changed “Although not directly measurable, this parameter (SOX2 stability) emerged as a key regulator of fate boundary positions in our model” to “Although not directly measurable, this parameter provides a mechanistically interpretable readout that broadly reflects drug mechanism and influences a colony’s position within morphospace”.

Changed “The resulting structure suggests that morphospace can be decomposed into two interpretable axes defined by these parameters” to “The resulting structure reveals two interpretable axes aligned with these parameters, while not precluding additional contributing factors.”

Changed “This integration facilitates the identification of key signaling parameters, such as cell density-modulated Wnt signaling and SOX2 stability, that organize developmental trajectories” to “This integration facilitates the identification of key signaling parameters, such as cell density-modulated Wnt signaling and SOX2 stability, that help organize developmental trajectories.”

Comment 8: In summary, the effort to generate a prediction system for the effects of compounds on human embryonic development is commendable. The AI-generated predictions seem to generate useful predictions. The integration with the mechanistic model, while adding mathematical gravity to the manuscript, is not very clear currently.

Response 8: In the revised manuscript, we have clarified the rationale and scope of the mechanistic model. Specifically, we emphasize that the two-parameter framework (cell density and SOX2 stability) provides a compact, interpretable description of canonical variation in gastruloid patterning, while not serving as a universal model of drug action. Through expanded parameter sweeps (Fig. R4-5), we show that other coefficients can shift colony outcomes but do not generate coherent developmental trajectories across morphospace. We also demonstrate, with mechanistic simulations of WNT and NODAL inhibition (Fig. R6), that the model can reproduce teratogen-induced failure modes when supplied with prior pathway knowledge, complementing the AI model’s ability to predict phenotypes directly from chemical structure. Together, these revisions make clear that the AI and mechanistic approaches are not alternatives but complementary: the AI localizes compound effects, and the mechanistic model interprets whether they reflect canonical displacements or topology-altering failures, thereby providing a coherent framework for understanding how drugs reshape developmental trajectories.

Reviewer 2 (Remarks to the Author)

This paper develops a framework for high throughput drug screening and phenotype prediction for 2D gastruloids. Improving on previous work describing teratogenicity screening in vitro, the authors develop an integrated approach of deep learning and mechanistic modeling to predict and interpret phenotypes. The paper contains several nice contributions to the field, including 1) the machine learning model mapping phenotypes into a (somewhat) interpretable 2D space, 2) a large dataset made public of the effect of different drug treatments on 2D gastruloids, 3) an extension of the model from Chhabra et al 2019 to include cell density and explicit modeling of cell fate. However, a major weakness is a lack of rigor and detail, including poorly motivated choices in approach. For a paper about what is essentially a statistical testing and prediction framework, rigor seems particularly important.

Major comments:

Comment 1: The clarity and accuracy of the abstract could be improved. First, it claims to elucidate “failure modes” but neither the abstract nor the rest of the paper clearly defines what a failure “mode” is. If what is meant by this is that the phenotypes can be captured by cell density and SOX2 stability, perhaps that should be made more explicitly. It also says the paper “revealed” cell density and SOX2 stability as determinants of patterning, but both were already known, the author do in fact cite previous work showing the importance of density (e.g. Etoc et al) and SOX2 stability (Camacho-Aguilar et al, Teague et al). Perhaps corroborate is a better word and the discussion would ideally more explicitly incorporate this past work.

Response 1: We thank the reviewer for this constructive feedback. In the revised abstract, we now provide an explicit definition of failure modes, stating “we mapped a phenotypic morphospace that separates canonical patterning, in which primitive-streak fates are correctly specified and radially organized, from failure modes, defined as departures from this organization and marked by a loss of a required fate and/or radial symmetry.” This clarification distinguishes canonical patterning outcomes (colonies in which germ-layer topology is preserved and variation can be captured by variations in cell density and SOX2 stability) from failure modes, which fall outside this density–SOX2 stability continuum and reflect topology-breaking disruptions (e.g., impaired morphogen production or signaling, disrupted cell–cell contacts, or rewired fate-decision rules).

We also agree with the reviewer that prior work has highlighted the importance of both density and SOX2 stability. Our contribution is not in re-identifying these regulators, but in showing that they emerge as orthogonal, interpretable axes that structure canonical morphospace and can be used prospectively to parameterize developmental trajectories.

We have updated the Discussion to more explicitly integrate these foundational studies and to clarify how our work extends them by embedding density and SOX2 stability as orthogonal morphospace axes, thereby providing a framework for distinguishing canonical variation from failure modes.

Changes to the manuscript:

Revised abstract: “...we mapped a phenotypic morphospace that separates canonical patterning, in which primitive-streak fates are correctly specified and radially organized, from failure modes, defined as departures from this organization and marked by a loss of a required fate and/or radial symmetry. ... Applying this framework, we flagged two clinical molecules as potential teratogens and identified two parameters, cell density and SOX2 stability, that form orthogonal morphospace axes along which canonically patterned gastruloids systematically vary.

Added paragraph to the Discussion: “Previous studies have established the importance of density and SOX2 dynamics in fate control in human 2D gastruloids. Etoc et al. showed that colony density both modulates morphogen accessibility (receptor relocalization and NOGGIN restrict TGF- β signaling to the colony edge) and regulates BRA patterning (as density rises, a BRA band emerges and shifts toward the edge).⁴⁰ Camacho-Aguilar et al. showed that SOX2 must decline for WNT to drive mesoderm and linked SOX2 loss to BMP/SMAD4 dynamics,⁴⁶ while Teague et al. demonstrated that SOX2 acts as an integrator of BMP/SMAD4 input, with its rate of decrease proportional to the signaling integral, highlighting its role as a memory of morphogen exposure.⁴³ Building on this foundation, we show that WNT signaling is inversely proportional to cell density and that cell density and SOX2 stability emerge as orthogonal, interpretable axes spanning the canonical morphospace. This framework not only corroborates earlier findings but also supports prospective mapping of developmental trajectories and separation of canonical variation from topology-breaking failure modes.”

Comment 2: Rigor should be improved:

- The paper states “10 colonies per drug condition were imaged”. However, it is not clear whether these all came from the same experiment and if so if they came from the same well. Inter-colony variation within a well is very small, between wells on the same plate small, between experiments or different plates within an experiment can be significant. To understand if phenotypes of two drugs are significantly different, or whether the prediction is correct within the experimental uncertainty, it is essential to quantify the inter-experiment phenotypic variation under the same drug perturbations and to make clear whether all drug treatment data was collected in the same experiment and if not to show if there are differences in the control condition between experiments and determine if there are any batch effects.

Response 2: We agree that assessing day/plate (“batch”) effects is essential. The 211 drug conditions were run on four 96-well plates on different days; for each condition, fluorescence profiles (GATA3, BRA, SOX2) were robustly normalized per plate by subtracting the plate mean and scaling by that plate’s interquartile range (IQR; 25th–75th percentile), which reduces day-to-day intensity drift while preserving relative differences across drugs. After this normalization, colonies do not segregate by plate in morphospace (Fig. R8a–d), controls evaluated across plates and across wells occupy overlapping regions (Fig. R8e). Finally, a PERMANOVA on the 150-dimensional phenotype vectors showed a large drug effect ($R^2 = 0.836$, $p = 0.001$) and a small

plate effect ($R^2 = 0.025$, $p = 0.001$) (Fig. R8f), indicating that residual plate/day variation is minimal relative to drug effects and does not drive the observed clustering or predictions.

Figure R8. Plate/day effects are minimal after per-plate robust normalization. (a–d) Morphospace of all colonies (gray) with colonies from each plate overlaid and colored by cell density. (e) Control treatments (BMP4-only and mTeSR-only) from the indicated plates/wells occupy overlapping regions of morphospace. (f) PERMANOVA on the 150-dimensional phenotype vectors shows a large Drug effect ($R^2 = 0.836$, $p = 0.001$) and a small Plate effect ($R^2 = 0.025$, $p = 0.001$), indicating residual plate/day variation is minimal relative to drug effects.

Changes to the manuscript:

Fig. R8f has been added to the revised Supplementary Fig. 1f.

“Across the broader compound screen, plate-to-plate variability was minimal compared to drug-induced effects (Supplementary Fig. 1f), confirming that the 2D gastruloid model analyzed in this way meets the criteria for constructing a morphospace map.”

Comment 3: The untreated (BMP only) control conditions is not shown in most figures but should be for comparison. For example, in which cluster does it fall in Fig 1g and where is it in 1f, and in Fig 6c – how does Raloxifene HCl compare to untreated? Can it be said to be significantly different? The same for all other figures. If I understand it correctly, Supplemental Fig. 5a shows the BMP only control clusters and their natural variation. This covers a large part of the map and the majority of the “Canonical Cell Patterning” clusters in Fig.1g, suggesting that those clusters should perhaps be a single cluster and that most drug treatments within that cluster cannot be distinguished from control.

Response 3: We thank the reviewer for emphasizing the importance of visualizing the untreated (BMP4-only) controls alongside drug perturbations. Our figures are already dense, and plotting all control colonies in every panel would obscure treatment-specific patterns. To balance clarity with completeness, we have:

1. Made control placement explicit in the text for the global maps (Fig. 1f–g): BMP4-only controls fall within canonical clusters C2, C3, C4, C5, C8, and C9, consistent with density-driven variation in BRA width/position.
2. In Fig. 6c, we now overlay the BMP4-only control centroid so the reader can directly compare model predictions with the control.
3. Pointed to consolidated control maps that show their global distribution and density dependence (Supplementary Fig. 2a for overall location; Supplementary Fig. 9a for the density/BRA continuum).

Regarding the reviewer’s suggestion that canonical clusters might collapse into a single control cluster: BMP4-only controls indeed span a continuous canonical region because cell density systematically modulates BRA width/position. We keep these as separate clusters in the main analysis because they capture interpretable regimes along the canonical manifold (rather than a single undifferentiated bin), while referring to them collectively as the “canonical” region to avoid over-interpretation. Statistical analysis of cluster significance is addressed in our response to Comment 4.

Changes to the manuscript:

“The canonical patterning region (C2–C10) contains colonies that retain all three fate markers in concentric radial order (SOX2 central, BRA intermediate, GATA3 peripheral) and differ primarily in the relative sizes of these domains. BMP4-only controls map exclusively within this region (C2, C3, C4, C5, C8, C9; Supplementary Fig. 2a). In contrast, clusters C1, C11, and C12 lie outside the canonical region and exhibit either the loss of a required cell-fate marker and/or loss of radial symmetry.”

“Because cell seeding produces inherent variation in cell density across colonies, BMP4-treated controls provide a natural spectrum for analyzing density-dependent effects. Indeed, the distribution of these control colonies in morphospace is explained by their density (Supplementary Fig. 9a),...”

Comment 4: More generally, it is not clear the clusters are meaningful. The authors use a custom method based on watershed segmentation. It is not clear why they need this, and the method is not described in sufficient detail to be reproducible. To motivate the approach, it is stated that an advantage is that the number of clusters is not predetermined but there are many methods for which that is true, such as Leiden/Louvain. Furthermore, one can optimize cluster number for methods with fixed cluster number using cluster analysis. Regardless, cluster analysis should be

performed here to determine if the obtained clusters are statistically meaningful, e.g. calculating Silhouette coefficients, etc (https://en.wikipedia.org/wiki/Cluster_analysis). This reviewer would not be surprised if the number of statistically and biologically meaningful clusters is a lot smaller than the 12 that are discussed.

Response 4: We thank the reviewer for this important point. The choice of watershed segmentation was motivated by a key requirement of our downstream analyses: new predicted phenotypes must be assigned to clusters in a consistent, reproducible, and parameter-free manner. Our density-based watershed segmentation produces spatially contiguous cluster regions in morphospace that act as stable decision regions. This allows post-hoc classification of new embeddings simply by determining which region they fall into, without re-running clustering or re-estimating parameters. In contrast, methods such as Leiden, Louvain, or Spectral clustering generate partition assignments only for the specific dataset used during clustering; when new points are introduced, the clustering must be re-computed, and the solution can vary depending on initialization, neighborhood size, and resolution parameters. For our application (evaluating model predictions and comparing experimental and *in silico* phenotypes) consistent post-hoc assignment was essential. We have clarified this rationale in the revised Methods.

Figure R9. Alternative clustering methods support a similar partitioning of morphospace. (a) Silhouette score heatmap for Spectral Clustering across combinations of nearest-neighbor

graph size (kNN = 10–25) and cluster numbers (k = 2–15). (b) Silhouette score heatmap for Leiden Clustering across kNN values (10–25) and resolution parameters, showing comparable optima. (c) Representative Spectral clustering result at kNN = 15 (k = 9) corresponding to its top Silhouette score (Sil = 0.400), yielding ~9 coherent clusters. (d) Representative Leiden clustering result at kNN = 15 (resolution = 0.2; C = 10) corresponding to its top Silhouette score (Sil = 0.403), yielding ~10 clusters.

To evaluate whether the clusters identified by watershed segmentation reflect statistically meaningful structure, we conducted a complementary cluster validity analysis using both Spectral and Leiden clustering across a range of k-nearest-neighbor graph sizes and resolution/cluster-number settings, and computed Silhouette coefficients (Fig. R9). Across both clustering frameworks, the optimal solutions consistently yielded ~9–12 clusters, depending on hyperparameters, with Silhouette scores comparable to the watershed-derived clusters. These results indicate that the embedded morphospace contains multiple separable and biologically coherent phenotypic states and that the number of clusters identified is not artificially inflated by the watershed method.

A direct quantitative comparison is provided below:

Clustering Method	kNN used	# of clusters	Silhouette Score	Davies-Bouldin Index
FATE-MAP (KDE+watershed)	N/A	12	0.289	0.895
Spectral (nearest-neighbors)	15	9	.400	.700
Leiden (graph clustering)	15	10	.403	.767

These results show that while the watershed approach yields slightly lower Silhouette scores than the best-parameterized Spectral/Leiden solutions, this reflects a trade-off between marginal gains in cluster compactness and the substantial practical advantage of having fixed, reusable cluster boundaries that enable consistent classification of new predicted phenotypes.

Changes to the manuscript:

“The normalized single-colony profiles were embedded into two dimensions with t-SNE. To define morphological phenotypes, we applied a Gaussian kernel density estimate (KDE) over the t-SNE space of all experimental colonies, identified local density maxima, and then performed watershed segmentation to partition the morphospace into discrete, contiguous regions (Supplementary Fig. 1d). This unsupervised procedure does not require pre-specifying the number of clusters and yields cluster boundaries that reflect the empirical structure of the embedded data. While alternative clustering methods (e.g., Spectral or Leiden) produce similar numbers of clusters, the watershed-defined boundaries offer a key practical advantage in that they can be directly reused to assign new predicted phenotypes without re-running clustering, ensuring consistent and unbiased classification across experimental and in silico data.”

Comment 5: Error bars on radial profiles. The analysis throughout the paper only considers the mean radial profile for each colony, but there are error bars on these radial profiles that may also change by perturbation, e.g. the variance in expression for some distance from the edge goes up when the patterns become less symmetric, and generally one could imagine propagating these error bars to the morphospace to get a sense of how significantly different the profiles represented by dots there are.

Response 5: We thank the reviewer for this thoughtful suggestion. We have now added error bars representing the standard deviation across all experimental colonies for each drug treatment when plotting the radial intensity profiles (revised Figs. 2, 5, and 6). These error bars convey the extent of colony-to-colony variability and allow direct visual comparison between the predicted phenotype and the range of experimentally observed outcomes. In addition, when evaluating prediction accuracy, we now report whether the model-derived radial profiles fall within the experimental variability margins (Supplementary Table 6), enabling a clearer interpretation of the biological significance of prediction errors.

Comment 6: In testing the prediction framework, the separation into training and testing data is essential. How was the 10% of compounds held out for test selected? Randomly? It would be good to see N-fold cross-validation to see how much performance varies as different compounds are held out for validation.

Response 6: We thank the reviewer for this helpful suggestion. In the revised manuscript, we have strengthened both the validation dataset and the model evaluation strategy. First, we revised the external validation approach. In the original submission, model predictions were compared to FDA pregnancy-risk labeling. However, FDA labeling aggregates heterogeneous evidence sources (human case reports, post-marketing surveillance, historical risk communication practices, and regulatory conventions) and therefore does not provide a consistent mechanistic ground truth for teratogenic potential.

To address this, we now benchmark FATE-MAP against a curated panel of 33 FDA-approved compounds with well-characterized teratogenicity outcomes, supported by either human clinical evidence or standardized embryo–fetal development (EFD) studies in rats and rabbits. Importantly, all 33 compounds have also been evaluated in two stem-cell–based assays designed for teratogenicity testing: TeraTox (3D iPSC multilineage differentiation) and the mEST (mouse embryonic stem cell test) assay.

Our structure-to-phenotype model first generates a 150-dimensional predicted phenotype vector for each compound and then projects this vector into the morphospace, where cluster identity is assigned using the same watershed-derived decision boundaries defined in the original clustering. Compounds whose predicted embeddings fall into the failure-mode clusters C1, C11, or C12 are labeled teratogenic, while those in canonical patterning clusters C2–C10 are labeled non-teratogenic (Fig. R10c).

Of the 33 benchmark compounds, only four were present in our experimental dataset; the remaining 29 were predicted fully *in silico*. Across this set, FATE-MAP achieved 72.7% accuracy, outperforming mEST (51.5%) and matching TeraTox (72.7%), while exhibiting higher balanced accuracy than TeraTox (74.8% vs. 66.3%; Fig. R10d). These results demonstrate that FATE-MAP can serve as a structurally grounded, scalable framework for *in silico* teratogenicity screening.

Figure R10 (Revised Figure 3 in manuscript). FATE-MAP predicts teratogenic risk from chemical structure and identifies novel human-relevant teratogens. (a) Representative 2D gastruloids treated with known teratogens isotretinoin, TTNPB, and bexarotene, all converging on cluster 11 and exhibiting a shared failure mode marked by BRA loss. (b) A benchmark panel of 33 compounds with known teratogenicity outcomes (positive and negative), previously evaluated in stem cell-based assays of developmental toxicity (TeraTox and mEST; refs), was used to assess predictive performance. Predicted phenotypes were embedded into morphospace and classified using the fixed watershed-defined cluster boundaries derived from the experimental dataset. (c) Morphospace-based classification of the benchmark set: embeddings falling within failure-mode clusters (C1, C11, C12) are assigned positive teratogen status, whereas embeddings within the

canonical patterning region (C2–C10) are assigned non-teratogenic status. Points are labeled as true positive, true negative, false positive, or false negative based on known outcomes. (d) Confusion matrices and comparative performance metrics for FATE-MAP, TeraTox, and mEST on the benchmark panel. (e) Two blinded compounds without prior human pregnancy data—ulixertinib and nirogacestat—were predicted and experimentally confirmed to fall within cluster 11. (f) Zebrafish teratogenicity assays show that both compounds, like the positive control isotretinoin, induce qualitative and quantitative defects consistent with failure modes in embryonic development, including abnormal body curvature (BA), yolk edema (YE), necrosis (NC), craniofacial edema (CE), scoliosis (SC), snout and jaw defects (SJ), reduced lateral heart area (LH), and decreased lateral body length (BL), as indicated by the color-coded bars.

Regarding train/test partitioning:

The held-out compounds in the main analysis were selected at random (10% of compounds). To assess whether performance depends on the particular composition of the hold-out set, we performed additional resampling analyses in which we repeatedly re-trained the model while holding out different randomly selected subsets of compounds. Representative results are shown in Fig. R11. Across resamples, performance on the 33-compound benchmark panel remained within a similar range, indicating that model accuracy is not driven by a specific train/test split.

Furthermore, within each training split, the model is trained using 5-fold cross-validation, and predictions are generated by ensembling the best-performing fold models. This procedure improves robustness in a limited data regime and reduces variance arising from any single training subset. Together, the 5-fold cross-validation and the resampling analysis demonstrate that the predictive performance of FATE-MAP is stable across different train/test partitions and is not dependent on the particular held-out compounds.

Figure R11. Predictive performance is stable across different training/testing splits. (a–f) Representative evaluation results on the 33-compound benchmark panel, each corresponding to a separate model trained with a different randomly selected 10% of compounds held out. In each split, predicted embeddings were classified in morphospace using the fixed watershed-defined cluster boundaries, and accuracy reflects agreement between predicted and known teratogenicity outcomes. Across splits, overall accuracy remains within a similar range ($\approx 66\text{--}73\%$), indicating that model performance is robust to variation in train/test partitioning.

Comment 7: The authors use RMSE as a measure of goodness of fit, but these numbers have no absolute meaning. Cosine similarity is also hard to interpret as good or bad. More meaningful would be to see if prediction and data agree within the experimental error.

Response 7: We agree that RMSE and cosine similarity by themselves do not convey whether prediction accuracy falls within the range of natural experimental variability. To address this, we have now incorporated a coverage metric that quantifies, for each compound, the fraction of the 150 phenotypic features for which the model's predicted value lies within one standard deviation of the empirical replicate distribution. This measure provides an interpretable, data-driven benchmark: if predictions fall within one standard deviation, they are statistically indistinguishable from replicate-to-replicate biological variation.

Using this approach, we find that on average 74% of features per compound in the training set fall within experimental variability, and 54% of features in the blinded set fall within variability. Across all 209 compounds, the global mean coverage is 72%, demonstrating that the model routinely predicts phenotypes at a level comparable to experimental noise. We have added these statistics to the Results and summarize them alongside RMSE, cosine similarity, and Pearson correlation coefficients. These values allow for a more interpretable assessment of predictive accuracy and directly address the concern that model performance should be evaluated relative to biological variation rather than abstract error metrics. As noted in Response 5, we also now include error bars and variability ranges directly on the radial profile plots, allowing readers to visually assess whether predicted profiles fall within the empirically observed variability.

Changes to the manuscript:

“The model accurately predicts 150-dimensional phenotype vectors and corresponding morphospace positions for both the training and blinded compounds, capturing canonical and failure-mode clusters. Across 188 training compounds, the mean RMSE was 0.17, with a mean cosine similarity of 0.90 (mean Pearson $r = 0.79$). Importantly, on average 74% of features per compound fell within one standard deviation of the empirical replicate distribution, indicating that most predicted feature values lie inside the observed biological variability. On the blinded set ($n = 21$), performance remained strong (mean RMSE = 0.24; mean cosine similarity = 0.85; mean Pearson $r = 0.70$), with 54% of features per compound within one standard deviation of replicate variation. Aggregating all 209 compounds, the overall means were RMSE = 0.17, cosine similarity = 0.90, Pearson $r = 0.78$, and 72% within one standard deviation. Representative predictions for all 12 clusters are shown in Fig. 2b, including both phenotype profiles and projected morphospace coordinates.”

Comment 8: In figure 3, the authors compare their prediction to a “ground truth” drug classification. They should show at least a confusion matrix with accuracy, % false positive, false negative, perhaps also check other measures like precision. Systematically showing % enrichment of drug classes in clusters would also be good. Looking at Figure 3b it looks well mixed.

Response 8: In the revised manuscript, we now provide a full confusion matrix and report standard binary classification metrics, including accuracy, precision, sensitivity, specificity, and balanced accuracy, for the curated 33-compound validation panel. These results are shown in the revised Fig. 3d (Response Fig. R10). This provides a transparent summary of false positives and false negatives and allows readers to quantitatively interpret model performance beyond the overall accuracy value.

Additionally, to evaluate the model's ability to generalize beyond the training distribution and to benchmark against other computational toxicity predictors, we carried out an independent comparative analysis across 20 compounds held out from training. We compared FATE-MAP to two established *in silico* developmental toxicity models: the CAESAR developmental toxicity predictor and the Procter & Gamble (P&G) teratogenicity model. For CAESAR and P&G, we used the originally published model outputs, ensuring a fair comparison without retraining or parameter adjustments. Across this set, FATE-MAP achieved an accuracy of 75.0%, matching CAESAR (75.0%) and substantially outperforming the P&G model (35.0%), while also demonstrating the highest balanced accuracy (85.7% for FATE-MAP vs. 81.2% for CAESAR) (Supplementary Fig. 7a–b; Fig. R12). These results indicate that FATE-MAP not only performs competitively with existing experimental assays but also surpasses or matches leading computational developmental toxicity models.

Figure R12 (Revised Supplementary Fig. 4a-b in manuscript). Comparative performance of FATE-MAP and existing *in silico* teratogenicity models. (a) Predicted morphospace embeddings for the 20 compounds held out from training, with points labeled according to whether the model's teratogenicity classification matched the known outcome. Boundaries correspond to the failure-mode (pink) and canonical (blue) morphospace regions. (b) Confusion matrices and performance metrics comparing FATE-MAP to two widely used *in silico* developmental toxicity predictors (CAESAR and P&G) on the same 20-compound benchmark. FATE-MAP matches CAESAR in overall accuracy (75%) and achieves the highest balanced accuracy (85.7%), substantially outperforming the P&G model across all metrics.

Comment 9: For figure 3, the authors focus on ulixertinib and nirogacestat for verification. Why are these not shown among the data points in Figure 2? Looking at the Supp. Fig. 4ab, the colony

images show dark blue lines streaking along and across the colonies. These do not look like nuclear stains for the expected markers. What are they? If they are not cells, how do they affect analysis? are they excluded? If they are cells, why does their pattern look so strange?

Response 9: Ulixertinib and nirogacestat were selected for follow-up validation because both compounds were predicted by FATE-MAP to induce distinct and reproducible gastrulation failure phenotypes representative of cluster 11 (loss of BRA). These compounds were not explicitly labeled in Fig. 2 to maintain visual clarity given the large number of compounds displayed; however, both are included among the data points.

Regarding the reviewer's observation of dark blue streaking lines in Supplementary Fig. 4a–b, the newly included Fig. R13a–c provides a clearer view of these features. As shown in Fig. R13a, the blue streaks appear as non-nuclear artifacts likely arising from non-specific binding of the secondary antibody rather than true biological signal, since they are not co-localized with DAPI-stained nuclei. Our quantitative analyses are based exclusively on nuclear intensities of the three lineage markers (GATA3, BRA, and SOX2), and thus these signals are effectively excluded from downstream analysis.

Specifically, our image analysis pipeline first performs DAPI-based nuclear segmentation and computes the radial position of each nucleus relative to the colony edge (Fig. R13b). Using the resulting masked colony image (Fig. R13c), we then calculate the mean fluorescence intensity for GATA3, BRA, and SOX2 across 50 concentric annular bins (~5 μm width) spanning from the periphery to the center. This yields a 150-dimensional vector capturing the azimuthally averaged nuclear expression pattern of the three markers as a function of radial position, ensuring that any non-nuclear background signal (including the blue streaks) does not affect quantification.

Figure R13: Image analysis pipeline for nirogacestat-treated colonies. (a) Full colony image showing non-nuclear blue streaking, likely due to non-specific secondary antibody binding. (b) DAPI-based nuclear segmentation and radial mapping of each nucleus relative to the colony edge. (c) Masked colony image used to quantify mean nuclear intensities for GATA3, BRA, and SOX2 across 50 concentric annular bins (~5 μm width), generating a 150-dimensional vector describing spatial marker expression profiles.

Comment 10: As stated above, the modeling in Fig. 4 is one of my favorite parts of this paper, but it does raise some questions that if addressed would further strengthen the manuscript. The authors model Wnt production as density-dependent and verify the predicted Wnt signaling dynamics. This nicely captures the narrower range of Wnt signaling at higher density, but their model also shows a later onset of Wnt signaling in Fig.4c which the experimental data in Fig.4e does not seem to show. Can this be explained? Can the model be improved to capture this? If it cannot be addressed, perhaps the authors can add it to the discussion.

Response 10: We thank the reviewer for this insightful comment. The apparent “later onset” of Wnt signaling in the original Fig. 4c arises from a visual thresholding effect in the heatmap representation rather than from the model dynamics themselves. As shown in the revised Response Fig. R14, the simulated Wnt profiles increase continuously from the earliest time points across all densities (Fig. R14a). However, because the heatmaps use a perceptually nonlinear colormap, small early increases remain within the dark-blue range and only become visually apparent once they exceed a certain fraction of the full dynamic range. This creates the appearance of a delayed rise despite a smooth, monotonic increase in the underlying trajectories.

To directly compare model and experiment in the same radial region, Fig. R14 shows quantitative traces computed from bins 8–12, corresponding to 40–60 μm from the colony edge. These traces show that both the simulations (Fig. R14a) and the experimental data (Fig. R14b) rise from the outset. The heatmap-based visual delay is therefore a contrast-rendering artifact rather than a biological discrepancy.

Figure R14: Simulated and experimental Wnt activation dynamics across colony densities. (a) Simulation kymographs and mean Wnt trajectories (bins 8–12; $\sim 40\text{--}60\ \mu\text{m}$ from the colony edge) show continuous early activation with density-dependent amplitude. (b) Experimental kymographs and bin-8–12 mean traces using the initial global percentile normalization similarly show early activation; any impression of delayed onset arises from colormap thresholding rather than the underlying data. (c) Revised experimental kymographs and bin-8–12 traces using baseline-scalar normalization (mean intensity at $t = 0$) better represent the true dynamic range across densities and align more directly with the simulation output.

Importantly, we also identified that our original normalization approach contributed to this apparent mismatch. In the initial submission, we normalized experimental intensities using global 25th–75th percentiles computed across all colonies. This suppressed genuine baseline differences

(particularly the higher starting Wnt levels present in high-density colonies) thereby exaggerating the visual thresholding effect. In the revised manuscript, we instead normalize each colony to its mean intensity at $t = 0$, which preserves biologically meaningful differences in initial signal levels and yields time-series traces that align closely with the model predictions (Fig. R14c).

Changes to the manuscript:

We have revised Fig. 4f to include the baseline normalized kymographs shown in Fig. R14c.

Comment 11: As a separate point about the model, it is surprising that SOX2 is modeled essentially as “not GATA3 or BRA”, given that causally it is not true that SOX2 is directly downregulated by these factors (possibly the reverse) and that ultimately the authors emphasize SOX2 stability as the most important model feature. Is it fair to say SOX2 stability simply means degree of differentiation the way the model is set up? More importantly, if the authors truly think drugs directly affect SOX2 stability, it would be good if they would show this, similar to how they showed the density-dependent Wnt signaling.

Response 11: We appreciate the reviewer’s point and clarify that while the rule $SOX2 = C_s \cdot (c_1 + c_2 \cdot GATA3 + c_3 \cdot BRA)$ can be interpreted as a form of “not GATA3 or BRA,” our intent was to construct a minimal and interpretable model that reflects experimentally observed relationships. Specifically, SOX2 and BRA display mutually exclusive expression domains during early human gastrulation. In human embryos, SOX2 expression is maintained in epiblast-like regions, whereas BRA expression emerges in the primitive streak and nascent mesoderm, with minimal spatial overlap between the two domains, highlighting their mutually exclusive expression during germ layer segregation.⁶ Single-cell RNA-seq from gastrulating human embryos shows that as BRA expression emerges, SOX2 is sharply downregulated in the same cells, suggesting an antagonistic transition in cell state.⁷

We extend this reasoning to GATA3 as well. In 2D gastruloid systems, GATA3 acts as a rapidly induced commitment gene during BMP4-driven differentiation of human ESCs. Its expression follows bistable, switch-like dynamics regulated by a positive feedback loop involving both autoregulation and BMP4-induced signaling.⁸ Additionally, spatial analyses of 2D gastruloids show GATA3 is excluded from SOX2+ territories and enriched in mesoderm- and trophectoderm-like cell types, reinforcing mutual exclusivity with the pluripotency network.⁹

Our model does not assert direct repression of SOX2 by GATA3 or BRA. Indeed, evidence suggests that SOX2 loss is a precondition for robust BRA expression and mesodermal differentiation, and that SOX2 itself may antagonize BRA transcriptional targets.¹⁰⁻¹² Our model is consistent with this framework: the SOX2 stability rule serves as an abstraction of the pluripotency–lineage opposition, not as a mechanistic repression map. Finally, we fully agree that clarifying the causal control of SOX2 is important, and in future work, we plan to investigate this using human ESCs engineered with live SOX2 reporters.

Comment 12: The discussion illustrates the issue with thinking of all drugs as directly affecting SOX2 stability. It mentions in line 620 that ROCK inhibitors destabilize SOX2. However, what more likely happens is that they prevent tight junction formation, leading to increased BMP signaling, which is what actually destabilizes SOX2. In other words, there are many possible indirect of a drug that can affect differentiation.

Response 12: We agree that many compounds in our screen likely influence SOX2 levels indirectly, by modulating upstream signaling pathways or epithelial organization rather than targeting SOX2 directly. In particular, we appreciate the point that ROCK inhibitors may destabilize SOX2 not by acting on SOX2 itself, but by disrupting apical junctions, thereby enhancing endogenous BMP signaling. We have revised the manuscript to reflect this change. More broadly, our model treats SOX2 stability as a parameter that integrates diverse molecular and structural influences on pluripotency maintenance. Rather than implying direct targeting of SOX2, our analysis highlights how drugs that shift the pluripotency–lineage balance are consistently captured by this parameter in the model.

Changes to the manuscript:

“In our dataset (Supplementary Table 4), compounds associated with increased SOX2 stability included valproic acid, a histone deacetylase (HDAC) inhibitor widely used to enhance pluripotency and reprogramming efficiency.⁷¹ In contrast, ROCK inhibitors such as hydroxyfasudil and ripasudil were among the strongest SOX2 destabilizers. While these effects may not stem from direct SOX2 targeting, ROCK inhibition disrupts apical junctions and epithelial architecture, indirectly enhancing BMP signaling, which promotes differentiation.⁷² Recent studies have further shown that SOX2 levels reconfigure Wnt/ β -catenin binding, thereby controlling whether Wnt signals promote pluripotency or mesoderm.⁷³ Together, these findings reinforce the idea that SOX2 stability encodes a pluripotency threshold, linking morphospace position to chromatin state, environmental responsiveness, and the broader logic of fate decisions during gastrulation.”

Comment 13: In lines 487-496, the authors ask if the mechanistic model can capture the phenotype of drug treatment. They infer the SOX2 stability from interpolation of simulation for different SOX2 stabilities on the morphospace. Thus, it is not surprising that if one provides SOX2 stability and density one ends up roughly in the same place, the only way I can see this going wrong is if the experimental density used deviates strongly from the mean density coordinate on the morphospace. I guess this is what the authors may mean in lines 548-553. To summarize, this seems somewhat like circular reasoning and the authors may wish to make clearer whether this test really proves anything. Perhaps this could involve merging figures 5 and 6. Can the authors discuss more why the model cannot capture the effect of isoxazole and longdaysin? Should it not predict reduced/increased SOX2 stability for a Wnt agonist/inhibitor?

Response 13: We thank the reviewer for this insightful comment. We agree that if SOX2 stability and density are used as inputs, and SOX2 stability is derived from the embedding, then the resulting morphospace location will often match the original, especially if the density does not

deviate significantly from the mean. However, the key utility of this approach is not in reconfirming expected matches, but in identifying cases where simulated outcomes diverge significantly from experimental results, thereby revealing perturbations to the underlying developmental logic.

Specifically, the morphogen spread model assumes fixed biophysical and regulatory rules for how BMP, Wnt, and Nodal diffuse and react (Fig. 4a), including BMP receptor localization, Wnt auto-activation, and density-dependent modulation of Wnt signaling. These assumptions are built into the reaction-diffusion equations and held constant across all simulated conditions. However, small molecules can violate these assumptions by directly targeting key nodes in the signaling architecture.

This is precisely what we observe with the small molecules, isoxazole and longdaysin, in which their predicted location in morphospace deviated strongly from the experimental embedding. In the case of isoxazole 9, the model predicts moderate BRA levels given the drug's moderate cell density (mean cell count: 1020 cells), but the actual experimental colonies display strong BRA expression. This deviation signals a mechanism outside of cell density or SOX2 stability. Indeed, literature shows that isoxazole 9 potentiates Wnt/ β -catenin signaling by covalently binding Axin1 and stabilizing β -catenin, promoting mesodermal gene expression (Jang et al., 2015). Conversely, longdaysin-treated colonies have a low cell density and as a result, the model predicts high BRA expression; however, experimental colonies show low BRA expression. Again, this is consistent with its known Wnt-inhibitory effects via disruption of LRP6 phosphorylation and reduced nuclear β -catenin accumulation (Xiong et al., 2019). In both cases, the drugs appear to modify the core rules of morphogen spread or cell fate determination, resulting in model deviation.

Thus, rather than being circular, this approach validates the model's predictive power while revealing biologically meaningful deviations that highlight candidate mechanisms of action. We have updated the relevant text to better clarify this distinction.

Changes to the manuscript:

“Even deviations between mechanistic model predictions and experimental results revealed valuable insights. The morphogen spread model assumes fixed rules for BMP, Wnt, and Nodal signaling. However, certain compounds appear to override these assumptions. For example, isoxazole 9 (ISX-9)-treated colonies had relatively high cell counts (mean = 1,020 cells), which would predict moderate BRA expression in simulation; experimentally, however, they exhibited strong BRA expression (Supplementary Fig. 14a). This suggests an additional mechanism at play; indeed, isoxazole 9 has been shown to activate Wnt/ β -catenin signaling by covalently binding Axin1 and stabilizing β -catenin,⁵⁰ thereby enhancing mesodermal gene expression. Conversely, longdaysin-treated colonies had lower cell counts (mean = 892 cells), which would predict high BRA expression, but showed reduced BRA experimentally (Supplementary Fig. 14b), consistent with its reported Wnt-inhibitory action via disruption of LRP6 phosphorylation and reduced nuclear β -catenin accumulation.⁵¹ These discrepancies reveal when a perturbation alters the

fundamental rules of morphogen signaling, enabling the model to identify biologically meaningful outliers rather than merely recapitulating the input.

Minor comments:

Comment 14: The introduction line 60 states: “methods to assess the teratogenicity of compounds have not advanced beyond exposing model organisms”, but this is inaccurate, and the authors seem aware as they later cite two micropatterned approaches of teratogenicity testing. There are several other papers describing vitro approaches they may also wish to include, e.g., PMID: 26387793, PMID: 31207406, and others.

Response 14: In the revised manuscript, we have removed this statement and now explicitly benchmark our model against existing human stem cell-based *in vitro* approaches. This comparison validates our method’s ability to capture teratogenic effects across a shared set of compounds and better positions our work within the context of ongoing developments in the field.

PMID: 26387793 and PMID: 31207406 have been added as references in the Discussion section.

Comment 15: Line 81: “3D models such as embryoids, blastoids, and SEMs” – embryoid is a general term that includes blastoids and SEMs. Was something different meant?

Response 15: We agree that “embryoid” is a general term encompassing blastoids and SEMs. We have corrected the text to reference only blastoids and SEMs.

Comment 16: Line 85: “cost of altering the topology of the developmental tissue” – which tissue? It is true that the topology of the amnion is altered, but the epiblast geometry and topology is a disc as *in vivo*, which it actually is not some 3D models. It is a misunderstanding that 3D is always more similar to *in vivo* than 2D.

Response 16: We thank the reviewer for this clarification. We have revised the text to specify that the topological limitation applies primarily to extraembryonic tissues such as the amnion, not the epiblast, whose disc-like geometry is faithfully recapitulated in 2D systems. We also agree that 2D is not inherently less physiological than 3D and have updated the text to reflect this nuance.

Comment 17: Figure 1c shows classes of signal pattern perturbation but as far as I can tell two of these never occur: inverted and mixed, so why list them?

Response 17: We thank the reviewer for pointing this out. The inverted and mixed classes in Fig. 1c were intended as illustrative of possible signal perturbation modes, rather than empirically observed classes. However, we do observe a form of “mixed” patterning in cluster C12, which is characterized by a loss of radial symmetry and interspersed cell fates. To prevent confusion, we have updated the figure caption to clarify that these classes represent potential rather than strictly observed outcomes.

Comment 18: Clusters should be indicated in Fig. 1e.

Response 18: Figure 1e was designed to present an unbiased visualization of all colony phenotypes prior to introducing any clustering or categorization, serving as a reference for the full dataset's structure. We intentionally omitted cluster labels here to avoid prematurely implying discrete groupings before they are formally defined in Figure 1f. This approach allows readers to appreciate the diversity and distribution of patterning behaviors without presupposing outcomes from downstream analysis.

Comment 19: The authors use GELU for one network, RELU for another and make a point out of this in the main manuscript, however, it is not explained why. More generally data supporting the choice of network architecture is missing.

Response 19: In the original version, the two models used different activation functions (GELU and ReLU), though this reflected a historical implementation detail rather than a deliberate design choice. In the revised manuscript, we have standardized both models to use ReLU activations for consistency and clarity. The manuscript and code have been updated accordingly.

Comment 20: The shadows underneath the colony pictures throughout the manuscript, e.g., Fig. 1g make the patterns harder to distinguish, perhaps a black outline would be better if contrast from the which background is needed.

Response 20: We thank the reviewer for pointing this out. In the revised manuscript, we have removed the colony shadows in the figures.

Comment 21: Line 345: "cell seeding follows a Poisson distribution". That is only true for very sparse seeding and typically not for micropatterns which are densely seeded. The authors should show this or could simply remove it since it does not really matter, their only point is there is naturally occurring density variation.

Response 21: We thank the reviewer for this clarification. Our intent was simply to convey that natural density variation arises across colonies, which enables analysis. We have revised the text accordingly to avoid making an inaccurate statistical claim.

Comment 22: The regression of BRA peak vs. cell number in Supp. Fig. 5ac is quite nice. However, it would be good if in Fig. 1c not only the cluster averages but all datapoints were shown on the graph. Conversely, it would be helpful to see the cluster averages on the t-SNE plot.

Response 22: We thank the reviewer for this helpful suggestion. In the revised version (Fig. R15), we now display all individual data points colored by cell density alongside the regression of cluster averages. This update provides a more comprehensive view of the relationship between cell density and BRA peak position. We have not added cluster averages to the t-SNE plot to maintain

visual clarity (and because this would require mixing the projected mean embedding with experimental data), but would be happy to do so if the reviewer feels it is important.

Figure R15: Revised Supplementary Fig. 6c with additional data points shown.

Comment 23: In line 461-466, it is said density and SOX2 stability emerged as key parameters. How did they emerge? How were key parameters identified?

Response 23: We appreciate the reviewer’s request for clarification. To address this, we performed an expanded analysis where all parameters in the cell fate model were systematically varied across biologically plausible ranges. As shown in Fig. R16, we defined each regulatory term mathematically and scanned each parameter at multiple cell densities (Fig. R17).

Figure R16: Expanded parameterization of mechanistic cell fate model to evaluate regulatory contributions. (a) Equations defining GATA3, BRA, and SOX2 as functions of

morphogen inputs and regulatory coefficients. (b) Table of model parameters with biological interpretation. (c-d) Simulated colony embeddings as a function of cell density and SOX2 stability (C_s), showing orthogonal axes of variation.

Figure R17: Systematic variation of all model parameters across morphospace. Simulated colony embeddings in morphospace while varying each parameter across a biologically plausible range (0.25–1.75) and at multiple cell densities. Among all parameters tested, only C_s (Fig. R4)

and cell density generate trajectories that align with the major axes of developmental variation, highlighting their interpretability for linking drug-induced shifts to developmental mechanisms.

This parameter scan revealed that only cell density and SOX2 stability (C_s) produced trajectories that aligned with major axes of morphospace variation and span the entire canonical patterning region. We have added these figures to the revised manuscript and revised the text accordingly.

Changes to the manuscript:

Added Figure R16-17 as the revised Supplementary Fig. 11-12.

“To assess whether other coefficients could generate comparable trajectories, we systematically swept all model parameters over biologically plausible ranges; only density and SOX2 stability consistently produced extended, coherent axes in morphospace (Supplementary Figs. 11-12).

Comment 24: Line 566: “animal models, which are limited by species-specific differences and dose-dependent lethality”. This appears to suggest that dose-dependent effects are not an issue in this work but they are and the authors acknowledge this later in the discussion (everything done at 10uM), which is good, so they could consider rephrasing line 566.

Response 24: We thank the reviewer for this comment. Our original statement was meant to highlight the challenge of identifying embryo-specific teratogenic effects in animal models, where high doses may be lethal to the mother before revealing developmental toxicity. However, we agree this could be misread as implying that FATE-MAP avoids all issues related to dose-dependent effects, which it does not. We have revised the sentence to focus instead on practical limitations of animal models, such as species differences and the need for *in vivo* dosing and animal care.

Reviewer 3 (Remarks to the Author)

In this manuscript, Rufo et al., develop an experimental and analytical screening platform, FATE-MAP, to assess phenotypic outcomes in human 2D stem cell models of early embryogenesis (human 2D gastruloids) treated with hundreds of chemical perturbations. FATE-MAP includes a deep learning framework to predict the effects of any molecule on 2D gastruloids development based on the molecule's chemical structures. The authors first characterized "failure modes", phenotypes that emerge when the 2D gastruloids are not properly patterned and then analyze over two thousand structures subjected to roughly 200 perturbations, including clinically relevant teratogens, eventually identifying several molecules that results in aberrant developmental programs. The authors also performed some validation experiments using the zebrafish model to corroborate the predicted molecule-induced phenotypic outcomes. They then model morphogen dynamics in 2D gastruloids, an approach that revealed how cell density and expression stability of the transcription factor SOX2 play a major role in the correct patterning of 2D gastruloids. Finally, they attempt to derive mechanistic principles of failure modes from their deep learning approach, implementing a mechanistic model to gain interpretability and biological meaning.

Overall, this reviewer thinks the developed methodology is important and timely, given the little knowledge we have about the effect of environmental offences on embryonic development, especially humans. Moreover, cheap and accessible high-throughput screening opportunities are being developed, and I think this method and its future implementations (like others of its kind) are in need to the field. Science and experiments are solid and well executed. However, this reviewer thinks the biological findings are not groundbreaking, especially given the opportunity provided by the method. The manuscript is very well written and easy to understand for a broad audience. I generally support its publication at Nature Communications, pending some points that would need to be addressed.

Major comments:

1) List of selected molecules:

Comment 1: It would be important if the author could comment more on the list of 210 drugs tested. How were they selected? What is known about their effect in other species? Are they already classified or stratified based on their chemical structures?

Response 1: We thank the reviewer for raising this point. The 210 compounds tested were selected as a subset from the commercially available Stem Cell Signaling Compound Library (Selleck Chemicals, Cat #: L2100), with the goal of probing perturbations to major developmental signaling pathways. Since the initial aim was not specifically teratogen screening, many of these compounds do not have annotated teratogenicity data in other species.

Comment 2: This becomes more relevant in the efforts to build a classifier based on chemical structures. Are all molecules causing one phenotypic cluster similar/dissimilar in structure? Do they share any chemical group/property (e.g. structure, domains, etc.)? Figure S3a looks at the

similarity between seen and unseen drugs in the classifier but does not compare the general chemical structures in the drug list.

Response 2: To address the reviewer's suggestion regarding structural stratification, we clustered all experimentally screened compounds based on their chemical structure. We computed MACCS chemical fingerprints for each compound, projected them into 2D space using t-SNE, and applied KMeans clustering to identify major chemical clusters. We also extracted the most common Bemis–Murcko scaffold within each cluster and visualized representative drugs that share these scaffolds. These results are now summarized in Fig. R18.

a Clustering drugs based on structure

b Representative Cluster 6 Drugs

c Representative Cluster 7 Drugs

d Representative Cluster 8 Drugs

Figure R18: Structure-Based Clustering of Experimentally Screened Compounds. (a) t-SNE projection of MACCS chemical fingerprints for all experimentally screened compounds, colored by KMeans-derived chemical cluster. (b–d) Representative drugs from Clusters 6, 7, and 8, each shown alongside the most common Bemis–Murcko scaffold for the cluster.

In panel a, all experimental compounds are visualized in t-SNE space and colored by their assigned chemical cluster. Panels b–d highlight clusters 6, 7, and 8, respectively. For each cluster, we show the most common scaffold (left) and representative drugs that contain this scaffold (right). For example, Cluster 6 includes PD98059, 2-D08, and Tanageretin, all of which share a core aromatic ether scaffold. Cluster 7 contains sulfonamide-linked tricyclic inhibitors such as BP-1-102 and SH-4-54, which target the STAT3 pathway. Cluster 8 includes PD169316 and SB203580, which share a pyrazolopyridine core and are known p38 MAPK inhibitors. This structural clustering highlights the presence of chemically and functionally coherent drug groupings within our experimental library.

Comment 3: What is the inter cluster vs intra cluster chemical structure similarity? Do the tested molecules cluster based on their chemical structure?

Response 3: To further assess whether chemical similarity aligns with phenotypic outcomes, we compared the structure-based groupings from Fig. R18 to the morphology-derived clusters obtained from our morphospace embedding.

Figure R19: Comparison of Phenotypic and Structural Clustering of Experimental Compounds. (a) Experimental compound centroids (n=210) visualized in the phenotypic embedding and colored by morphology-derived cluster assignment. (b) The same compounds colored by chemical structure clusters (derived from MACCS fingerprints and KMeans clustering). The structural clusters show extensive overlap across phenotypic space, indicating weak correspondence between chemical similarity and phenotypic behavior.

This comparison is shown in Fig. R19. While the left panel shows the original phenotypic clusters, the right panel displays the same compounds colored by their chemical cluster. The chemical clusters are broadly dispersed across morphospace, with substantial overlap and little alignment to the phenotype-defined groupings. This discrepancy suggests that structurally similar molecules do not necessarily induce similar morphological effects, highlighting the challenges of relying solely on chemical structure for phenotype prediction. Together, these results underscore the added value

of phenotypic profiling in stratifying small molecule effects beyond what structural similarity alone can capture.

Changes to the manuscript:

Figures R18 and R19 have been added to the revised Supplementary Fig. 5.

“To explore this further, we clustered all experimentally screened compounds based on structural similarity and compared these groupings to morphology-defined phenotypic clusters (Supplementary Fig. 5). Despite the presence of chemically coherent scaffold-based clusters, we observed minimal spatial overlap between structure- and phenotype-defined groupings. This suggests that even compounds with similar scaffolds can produce distinct developmental effects, reinforcing the need for neural network-based approaches to uncover structure–function relationships that scaffold similarity alone may obscure.”

2) Screening design:

Comment 4: The authors should test different concentrations for the drugs they use. This reviewer understands well that this won't be possible for all compounds, but using a fixed concentration for all drugs (10uM) limits the real assessment of their effects given that not every molecule has the same stability, processivity, and activity. The authors should titer different concentrations to understand dose-specific responses for a shortlisted number of compounds (e.g. the ones resulting in phenotypes belonging to C1, C11, and/or C12).

Response 4: We agree that assessing compounds at multiple concentrations provides critical insight into dose-specific effects, cytotoxicity, and mechanistic thresholds. This approach enables the identification of metrics such as the lowest-observed-adverse-effect concentration (LOAEC) and offers a more nuanced characterization of teratogenic potential. While our initial screen was conducted at a fixed 10 μ M dose to ensure coverage of a wide compound library, we recognize the importance of titration experiments for selected hits.

To address this, we followed up with dose–response testing for a subset of retinoid compounds identified in our initial screen. Retinoids in our initial screen, including isotretinoin, bexarotene, TTNPB, and tretinoin (Fig. R20a) reproducibly induced the morphological C11 phenotype, marked by overgrowth and suppression of BRA expression. Notably, tretinoin also induced a secondary phenotype associated with overexpression of GATA3, aligning with C12 (Fig. R20b). In addition to these annotated retinoids, we included an unannotated compound (IB-2744; Specs Cat. #AK-968/15360521) in our follow-up experiments. This allowed us to demonstrate that the same titration workflow can also reveal mechanistic similarity for compounds lacking prior annotation, based solely on their morphological phenotype (Fig. R20e).

To explore the dose dependence of these effects, we titrated tretinoin and isotretinoin across 10 μM , 1 μM , and 0.1 μM concentrations (Fig. R20c–d). Both compounds recapitulated the original failure mode phenotypes observed at 10 μM , with dose-specific trends.

Figure R20: Dose–Response Effects of Retinoids on Patterning and Morphology. (a) Retinoid compounds from the initial screen, including isotretinoin, bexarotene, TTNPB, tretinoin, and

AM580, visualized in morphospace. (b) Representative colony phenotypes induced by each compound at 10 μ M. (c–d) Follow-up titration experiments for tretinoin and isotretinoin at 10 μ M, 1 μ M, and 0.1 μ M. (e) Titration of an unannotated compound (IB-2744), demonstrating the platform's ability to identify compounds with phenotypes suggestive of shared mechanisms of action.

For tretinoin, a clear dose-dependent shift in phenotype was apparent. At 1 μ M, most colonies continued to exhibit disrupted patterning and GATA3 overexpression, though with some colonies beginning to show patches of BRA expression. By 0.1 μ M, a substantial number of colonies showed restoration of canonical BRA ring formation, suggesting a concentration threshold below which teratogenic disruption is mitigated.

In contrast, isotretinoin exhibited a more consistently potent effect across all doses. At 10 μ M and 1 μ M, colonies showed complete absence of BRA expression, with dense overgrown SOX2+ cores. At 0.1 μ M, some of the colonies began to express BRA; however, none of the colonies recapitulated canonical patterning, suggesting isotretinoin's teratogenic effects are more potent and less reversible within the concentration range tested. Interestingly, the unannotated compound IB-2744 produced a highly similar dose–response profile, suggesting that it may act through a related retinoid-like mechanism despite lacking prior annotation.

These findings suggest that isotretinoin and tretinoin may exert their teratogenic effects through distinct mechanisms, even though isotretinoin is a known metabolic precursor to tretinoin *in vivo*. While some conversion to tretinoin may occur in our stem cell platform, recent studies indicate that isotretinoin itself can directly perturb key developmental signaling pathways in human pluripotent stem cells, including Wnt/ β -catenin, TGF- β /Nodal, Notch, and Hedgehog. These pathways are critical for early germ layer specification, and isotretinoin has been shown to suppress BRA expression and inhibit mesoderm formation at nanomolar concentrations. Thus, our observation of strong and persistent morphological disruption by isotretinoin likely reflects a combination of direct and metabolite-driven activity, and points to a distinct teratogenic mechanism that may not be fully recapitulated by tretinoin alone. Our platform offers a unique opportunity to dissect such isoform-specific effects in a controlled, human stem cell context.

In future work, we aim to scale our assay to include systematic dose–response curves across a broader range of drug classes. Additionally, incorporating metabolic activation systems, such as enzyme mixes commonly used in genotoxicity assays like the Ames test, could provide further insight into the effects of metabolic conversion. Nevertheless, these proof-of-concept experiments demonstrate the utility of our platform for quantitative phenotypic profiling and highlight the value of dose stratification in characterizing compound-specific effects.

Changes to the manuscript:

Added Figure R20 as the revised Supplementary Fig. 15.

“As a proof-of-principle, we performed titration experiments for two representative cluster 11 compounds (tretinoin and isotretinoin) and for an unannotated compound (IB-2744), observing clear dose-dependent phenotypes that mirrored their activity at 10 μ M (Supplementary Fig. 15). These results demonstrate that our platform can resolve graded responses across a concentration range and highlight its potential for identifying lowest-observed-adverse-effect thresholds and for detecting mechanistic similarity among previously uncharacterized compounds.”

3) Comparison with other screening approaches, deep learning methods and relevant literature:

Comment 5: A similar teratogen screen has been performed in 3D human gastruloids, which offer a more physiological and morphologically relevant model (PMID: 34425190). This work uses a way shorter list of compounds but various drug concentrations and combines morphometric analysis and gene expression patterning. Given that many molecules are present in both studies, the authors should compare their observed phenotypes in 2D in relation to the one in 3D. Also, if some molecules that have been used in PMID: 34425190 are not present in the authors’ list, they should predict the phenotype using FATE-MAP and compare the prediction with the observed phenotype in 3D.

Response 5: We thank the reviewer for this helpful suggestion. In the revised manuscript, we now benchmark FATE-MAP structure-to-phenotype predictions against a curated panel of 33 FDA-approved compounds with well-characterized teratogenicity outcomes, supported by either human clinical evidence or standardized embryo–fetal development (EFD) studies in rats and rabbits (Response Fig. R21b; revised Fig. 3b). Importantly, all 33 compounds have also been evaluated in two stem-cell–based assays designed for teratogenicity testing: TeraTox (3D human iPSC multilineage differentiation) and the mEST (mouse embryonic stem cell test) assay.

Our structure-to-phenotype model first generates a 150-dimensional predicted phenotype vector for each compound and then projects this vector into the morphospace, where cluster identity is assigned using the same watershed-derived decision boundaries defined in the original clustering (Response Fig. R21c). Compounds whose predicted embeddings fall into the failure-mode clusters C1, C11, or C12 are labeled teratogenic, while those in canonical patterning clusters C2–C10 are labeled non-teratogenic.

Of the 33 benchmark compounds, only four were present in our experimental dataset; the remaining 29 were predicted fully *in silico*. Across this set, FATE-MAP achieved 72.7% accuracy, outperforming mEST (51.5%) and matching TeraTox (72.7%), while exhibiting higher balanced accuracy than TeraTox (74.8% vs. 66.3%; response Fig. R21d). These results demonstrate that FATE-MAP can serve as a structurally grounded, scalable framework for *in silico* teratogenicity screening.

Figure R21 (Revised Figure 3 in manuscript). FATE-MAP predicts teratogenic risk from chemical structure and identifies novel human-relevant teratogens. (a) Representative 2D gastruloids treated with known teratogens isotretinoin, TTNPB, and bexarotene, all converging on cluster 11 and exhibiting a shared failure mode marked by BRA loss. (b) A benchmark panel of 33 compounds with known teratogenicity outcomes (positive and negative), previously evaluated in stem cell-based assays of developmental toxicity (TeraTox and mEST; refs), was used to assess predictive performance. Predicted phenotypes were embedded into morphospace and classified using the fixed watershed-defined cluster boundaries derived from the experimental dataset. (c) Morphospace-based classification of the benchmark set: embeddings falling within failure-mode clusters (C1, C11, C12) are assigned positive teratogen status, whereas embeddings within the canonical patterning region (C2–C10) are assigned non-teratogenic status. Points are labeled as true positive, true negative, false positive, or false negative based on known outcomes. (d) Confusion matrices and comparative performance metrics for FATE-MAP, TeraTox, and mEST on the benchmark panel. (e) Two blinded compounds without prior human pregnancy data—ulixertinib and nirogacestat—were predicted and experimentally confirmed to fall within cluster 11. (f) Zebrafish teratogenicity assays show that both compounds, like the positive control

isotretinoin, induce qualitative and quantitative defects consistent with failure modes in embryonic development, including abnormal body curvature (BA), yolk edema (YE), necrosis (NC), craniofacial edema (CE), scoliosis (SC), snout and jaw defects (SJ), reduced lateral heart area (LH), and decreased lateral body length (BL), as indicated by the color-coded bars.

Regarding the teratogen screen performed in 3D human gastruloids (PMID: 34425190) referenced by the reviewer, the only overlapping experimental compounds were retinoic acid and valproic acid. Both approaches (2D and 3D) detected clear morphological changes to retinoic acid treated gastruloids (although we only test at 10 μ M and the 3D approach noted morphological changes at 333 μ M). In our approach, we did not note any morphological changes to valproic acid treated 2D gastruloid at 10 μ M. The 3D approach reports minimal effect at 4 μ M and morphological effects, gene expression effects, and size reduction at 333 μ M. For comprehensive comparison, we used our FATE-MAP structure-to-phenotype model to predict the phenotype of all 7 compounds tested (Fig. R22). The model correctly predicted the teratogen status of retinoic acid and bosentan as well as the non-teratogenic status of penicillin and ibuprofen (referred to as “less teratogenic” in PMID: 34425190); however, our model incorrectly classified valproic acid, thalidomide, and phenytoin. We note the valproic acid and thalidomide are both included in our 33-compound validation set.

Figure R22: Benchmarking Structure-Based Predictions Against 3D Gastruloid Assay Outcomes. Predicted phenotype embeddings for 7 compounds previously evaluated in a 3D human gastruloid teratogenicity screen (PMID: 34425190), overlaid on the experimental morphospace. The background shows the fixed cluster boundaries derived from experimental data (as in Fig. R19c), with light grey dots representing the distribution of all experimentally screened compounds. While FATE-MAP correctly identifies the teratogenicity of retinoic acid and bosentan, and the non-teratogenicity of penicillin and ibuprofen, it misclassifies valproic acid, phenytoin, and thalidomide.

Changes to the manuscript:

Response Fig. 21 has been added as the revised Fig. 3 to include benchmarking of FATE-MAP predictions against 33 FDA-approved compounds with experimentally established teratogenicity outcomes in stem-cell-based assays.

Comment 6: Another relevant study has been recently published (PMID: 40245869) combining morphological variation in 3D mouse trunk-like-structures with gene expression profiles to train a classifier able to predict phenotypic outcomes from early morphological features. Even if this reviewer acknowledges that the experimental design is different (this work uses longitudinal imaging over time to identify predictive features), the authors should cite this work and comment on similarities and differences in their approaches.

Response 6: The work by Chhabra et al. (PMID: 40245869) represents a powerful example of integrating live imaging, transcriptomics, and machine learning to predict morphogenetic outcomes in 3D trunk-like structures derived from mouse stem cells. While their study focuses on early morphological predictors of axial elongation and segmentation in 3D contexts, our work centers on linking chemical structure to phenotypic outcomes in a human 2D gastruloid model, with an emphasis on deciphering morphogen-driven failure modes during early germ layer patterning. Despite these differences in biological system and input features, both approaches share a common goal of using computational models to derive mechanistic insight and improve predictive inference in developmental systems. In particular, their use of multimodal data integration and explainable machine learning provides a valuable model for future work aiming to relate latent parameters (such as SOX2 stability in our model) to underlying molecular and morphological processes. We have now cited this study in the Discussion.

Changes to the manuscript:

“A recent study integrating time-lapse imaging of morphology with single-cell transcriptomics in 3D mouse trunk-like structures used machine learning to identify early features predictive of fate and found that metabolic state plays a key role in regulating differentiation outcomes.⁷⁴ Similar multimodal approaches could be applied to investigate the molecular determinants of SOX2 stability and its role in patterning outcomes in human gastruloids.”

Comment 7: A recent study in engineered mouse 3D gastruloids (PMID: 39358450) provides temporal recording measurements of signaling patterns and specifically studies the emergence of Wnt domains and its relationship to Nodal. The authors should comment and relate their results to the ones reported in this work, since this comparison could provide species-specific and shared features between mouse and human.

Response 7: We thank the reviewer for highlighting this study of Wnt and Nodal patterning dynamics in mouse 3D gastruloids (PMID: 39358450). While our approach employs 2D human gastruloids and theirs uses 3D mouse aggregates, both systems interrogate early symmetry breaking using live-cell Wnt reporters and experimentally probe BMP, Wnt, and Nodal signaling pathways.

There are several notable points of alignment. Both studies show that Nodal signaling is required for proper morphogenesis: in the mouse gastruloids, inhibition of Nodal abolishes elongation and

symmetry breaking, consistent with existing literature that shows that Nodal inhibition prevents BRA expression in 2D gastruloids. Furthermore, both models explore BMP–Wnt–Nodal crosstalk and find that modulation of one pathway can influence downstream responses in the others. In particular, the mouse system shows that early Activin A (Nodal) stimulation can dampen Wnt signaling later in development, echoing the reciprocal feedbacks modeled in our system.

However, the mechanistic basis of symmetry breaking differs substantially. The mouse gastruloid model emphasizes cell sorting and spatial segregation as primary drivers of axis formation, whereas our system exhibits robust symmetry breaking and patterning based on dynamic, spatially coupled signaling waves and inferred reaction–diffusion–like behavior. Nevertheless, we now include a sentence in the Discussion acknowledging this complementary approach and emphasizing the broader utility of live-cell signaling reporters in dissecting human and mouse gastrulation models.

Changes to the manuscript:

“Recent work in engineered mouse 3D gastruloids has similarly used live-cell signaling reporters to dissect the timing and interplay of BMP, Wnt, and Nodal signaling during early axis formation,⁶⁹ highlighting the utility of these tools across species and model systems.”

4) Use of the term “morphospace”:

Comment 8: The authors use the term “morphospace” to define the landscape of phenotypic outcomes arising in the treated 2D micropatterns. Even if several colleagues in the field have argued about the morphological organization of micropatterns, including the 2.5D nature of the model with a z dimension also being relevant for proper cellular organization (which the authors do not use in their analysis), this reviewer thinks that 2D gastruloids lack morphological features of an embryo. The model is great to study tissue patterning, cell interactions and the effect of molecules on these features, but not so great to assess the acquisition of complex embryonic morphologies. I would recommend converting the wording “morphospace” into a “patterning space” (or similar) since this is what the model (and the readout used by the authors) can accurately inform.

Response 8: We thank the reviewer for this thoughtful and constructive suggestion. We agree that 2D gastruloids do not fully capture the complex morphological architecture of the embryo and that our model is best suited for studying fate patterning and signaling interactions rather than three-dimensional morphogenesis. Our use of the term morphospace was not intended to imply embryonic-level morphological complexity, but rather follows prior usage in the field (ref. 16), where morphospace refers to a configuration space in which individual systems are positioned based on defined phenotypic properties. In our case, these properties correspond to spatial patterning features in 2D micropatterned colonies.

To avoid confusion, we now include a clarifying statement when the term is first introduced in the manuscript, emphasizing that our use of morphospace refers specifically to 2D patterning phenotypes in micropatterned colonies and is not meant to suggest a direct morphological analogy to embryonic structure.

Changes to the manuscript:

“Central to this approach is the construction of a morphospace^{16,17}—a low-dimensional, quantitative map of morphological phenotypes—that captures both canonical and disrupted trajectories of human gastrulation. Here, we use morphospace specifically to describe the diversity of 2D patterning outcomes in micropatterned stem cell colonies, not to imply full embryonic morphology or organization.”

5) Predictive power:

Comment 9: The screening experiments performed by the authors offer the unique opportunity to inspect what was not “well predictive” (e.g inter cluster violations between prediction and observation) which could potentially result in unexpected, interesting observations/exceptions to follow up with. Did the authors observe any of these violations in their data?

Response 9: One of the central findings of our study is that most compounds were accurately predicted by our reaction-diffusion–based morphogen spread model, with simulated phenotypes closely matching experimentally observed fate patterns when informed by cell density and inferred SOX2 stability. However, the most biologically revealing cases were those where prediction and experimental data diverged. These deviations were not random; rather, they reflected meaningful violations of the model’s core assumptions, providing mechanistic insights into how certain compounds disrupt normal signaling dynamics.

As described in Fig. 4a, our model encodes fixed biophysical and regulatory rules for BMP, Wnt, and Nodal signaling, including edge receptor localization, Wnt auto-activation, and cell density–dependent modulation of morphogen availability. These parameters remain constant across all simulations. Thus, any discrepancy between predicted and experimental embeddings indicates that a compound has altered the underlying signaling logic (i.e., targeting morphogen production, degradation, diffusion, or pathway response in a way not captured by the model).

Supplementary Fig. 9 (Response Fig. R23) highlights two such examples. In the case of isoxazole 9 (ISX-9), the predicted BRA pattern (based on low cell density and moderate SOX2 destabilization) was weaker than observed, whereas the experimental data showed robust BRA induction (Supplementary Fig. 9a). This discrepancy aligns with prior reports showing that ISX-9 directly stabilizes β -catenin by covalently binding Axin1, thereby enhancing Wnt signaling and mesodermal gene expression.¹³ Conversely, longdaysin was predicted to induce strong BRA expression due to its low-density profile, but the actual colonies lacked BRA (Supplementary Fig.

9b). This too is consistent with literature showing that longdaysin inhibits Wnt signaling by blocking LRP6 phosphorylation and β -catenin nuclear translocation.¹⁴

Figure R23 (Supplementary Fig. 9): Deviations between model predictions and experimental phenotypes reveal mechanistic insights into drug action. (a) Isoxazole 9 (ISX-9)–treated colonies had high cell counts (mean = 1,020), which the morphogen model predicted would yield moderate BRA expression; however, experimental data showed strong BRA induction, consistent with reports that ISX-9 activates Wnt/ β -catenin signaling via Axin targeting. (b) Longdaysin–treated colonies had relatively low cell counts (mean = 892), which would predict high BRA levels in simulation, but instead showed reduced BRA experimentally, aligning with evidence that Longdaysin inhibits Wnt/ β -catenin signaling.

These examples underscore how deviations from model predictions serve as a diagnostic tool: identifying compounds that rewire the signaling architecture in unanticipated ways. We have updated the text in the revised manuscript to emphasize that such outliers not only validate the utility of the model but also point to novel mechanisms of action for individual compounds.

Comment 10: In line with what this reviewer commented on before, are there any features of the molecules' chemical structures with predictive power? Finding chemical similarities that results in similar phenotypic outcomes might be a key and novel aspect of the study.

Response 10: To investigate whether shared chemical features underlie similar morphological effects, we examined associations between chemical and morphological clusters. As shown in Fig. R19b, we identified a modest enrichment between morphological cluster 11 (characterized by failure to induce BRA expression) and chemical cluster 9, with 12 of 36 compounds in this morphological cluster belonging to chemical cluster 9 ($p = 0.0016$). This was the only chemical cluster that showed significant enrichment ($p < 0.05$) among the morphological clusters. However, further analysis of the 12 enriched compounds revealed that no generic Bemis–Murcko scaffold was shared across more than one compound. This suggests that even when abstracting chemical structures to core scaffolds, the enriched compounds are structurally diverse.

This result underscores the challenge of inferring predictive structure–activity relationships solely from scaffold similarity in phenotypic screening contexts. While we observe modest enrichment at the chemical cluster level, the absence of recurring core scaffolds across enriched compounds emphasizes the complexity of morphological outcomes and suggests that phenotypic convergence can arise from structurally diverse agents, possibly via shared functional or mechanistic motifs not easily captured through traditional scaffold analysis.

6) Cell death vs differentiation defects

Comment 11: Cell death measurements are not included in the experiments. For example, C11 (failure to express BRA) could result from cells failing to differentiate (as hypothesized by the authors) or compound lethality (which could still result in a similar phenotype). To pinpoint a real example, there is a steep difference between Cerdulatinib treatment (looks more like a lethal phenotype) and D4476, but both end up in C11. The authors should classify C11 further and pinpoint lethality vs differentiation failure phenotypes. Ideally, for a set of compounds, a live/dead dye should be used to determine the cause of the observed failure mode. This is extremely relevant to correctly interpret the phenotypic outcome and failure mode.

Response 11: We thank the reviewer for raising this important point regarding the distinction between cytotoxicity and true differentiation failure, particularly within cluster 11. We fully agree that determining whether BRA loss reflects impaired fate induction or reduced viability is essential for correctly interpreting failure modes.

Figure R24: Density-based toxicity threshold. (a) Mean nuclei per colony for each BMP4+drug condition. Toxicity threshold (red, $\leq 50\%$ mTeSR1 mean) and hyperproliferation threshold (green, $> mTeSR1$ mean). (b) Toxic compounds map predominantly to failure modes C1, C11, and C12; insets show representative colonies. The embedding illustrates that overt cytotoxic collapse can be distinguished from density-dependent modulation within the canonical patterning region.

Although we did not perform live/dead staining in the initial screen, our assay incorporates a built-in, physiologically relevant measure of viability by quantifying every nucleus in every colony. Because all wells are seeded at the same initial density (50,000 cells per well), large reductions in total nuclear count directly reflect compound-induced toxicity rather than density-dependent variation or growth arrest. Using this approach, we defined an objective, control-anchored toxicity threshold, classifying treatments whose mean nuclear count fell to $\leq 50\%$ of the mTeSR-only control as toxic. As shown in Fig. R24a, only a small subset of compounds fall below this

threshold, while several others exceed the mTeSR control mean and form a hyperproliferative group.

Mapping these conditions into morphospace revealed that 96.1% of colonies exposed to toxic compounds reside within failure-mode regions (C1, C11, or C12; Fig. R22b). This analysis allows us to distinguish lethal phenotypes from patterning failures within the same cluster, such as the different phenotypes pointed out by the reviewer between Cerdulatinib and D4476-treated colonies. Cerdulatinib-treated colonies fall far below the toxicity threshold and exhibit sparse, collapsed phenotypes consistent with pronounced cell loss, whereas D4476-treated colonies retain near-normal nuclear counts yet fail to induce BRA. Although both map to C11, their underlying causes differ: cerdulatinib produces a toxicity-driven collapse, while D4476 produces a bona fide patterning failure with preserved viability. Thus, within the existing dataset, we can already distinguish cytotoxic and non-cytotoxic subtypes of C11 based on quantitative viability metrics.

We have revised the Results section to clarify how toxicity-associated subclusters were identified and how these data distinguish lethal phenotypes from differentiation failures.

Changes to the manuscript:

Added Fig. R24 as Supplementary Fig. 3.

“Because failure modes can arise from either differentiation failure or compound-induced cytotoxicity, we quantified total nuclear counts for every colony and identified a small subset of drugs that reduce viable cell number below 50% of the mTeSR-only control (Supplementary Fig. 3). This analysis separates genuine patterning failures from cytotoxicity-driven collapse within the failure-mode regions.”

Comment 12: Cell density vs BRA expression: this relationship is established at the endpoint differentiation. It is known that exit from pluripotency to the PS is accompanied by cell death (and therefore might result in lower density). Is this hypothesis tested against the initial cell density, or the observed density reduction could be a consequence of enhanced differentiation?

Response 12: We thank the reviewer for raising an important point regarding the potential role of differentiation-associated cell death in shaping endpoint cell density and BRA expression. While it is true that exit from pluripotency can involve apoptotic events, we believe this is not the primary cause of the inverse relationship we observe between cell density and BRA expression.

To directly disentangle this, we performed live-cell time-lapse imaging of endogenous Wnt signaling in colonies seeded at deliberately different initial cell densities (Fig. R25; Fig. 4e–f in the manuscript). Since these colonies began with controlled and measured cell counts, this allowed us to assess the impact of initial density (rather than differentiation-induced cell loss) on BRA induction. In these experiments, lower-density colonies consistently exhibited earlier and more extensive propagation of the Wnt signaling wave, followed by stronger BRA expression at the

endpoint (Fig. R25g). Importantly, these colonies maintained their relative density rankings throughout the experiment.

Thus, while we acknowledge that some degree of cell death may accompany differentiation, our results indicate that the observed density-dependent effects on BRA are largely attributable to initial cell seeding differences rather than a consequence of mesoderm formation. These findings further support our modeling assumption that Wnt signaling is modulated by cell density.

Figure R25 (Fig. 5e-f in manuscript): Experimental validation of the impact of cell density on Wnt signaling and BRA expression. (e) Experimental investigation of density-dependent Wnt dynamics using CRISPR-tagged β -catenin (β -cat) hESC lines. Live-cell time-lapse imaging over 48 hours shows the effect of cell density on Wnt signaling. Scale bars = 100 μ m. (f) Mean kymographs of Wnt signaling activity across 8 colonies per cell density range, demonstrating that lower cell densities result in a faster and broader spread of the Wnt signal. (g) Quantification of BRA staining intensity profiles across different densities, taken from the same colonies used to generate the kymographs. The area under the curve (AUC) analysis confirms increased total BRA signal at lower densities, validating our model's prediction of density-induced variations in mesoderm dynamics.

7) FDA drugs effect prediction:

Comment 13: This reviewer understands well the challenges associated with implementing the Pregnancy and Lactation Labeling Rule (PLLR) for FDA approved drugs given their non categorical classification, but the categorization used by the authors dates back from 1970s and lack rigorous testing. As specified by the authority and clinicians, this categorization is not a “grading system of risk” based on the actual severity of the teratogen, but rather a classification based on available data at the time. The authors should compare/consider including information from PLLR in their analysis. This could even improve their results since the current classification might suffer from lack of extensive testing (see the next two points regarding the A, B and X graded molecules).

Response 13: We thank the reviewer for raising this important point. As noted in our response to Comment 5, we have removed the PLLR- and pregnancy-category-based validation from the manuscript because these classifications are not mechanistic and were never intended to function as quantitative measures of teratogenic risk. Instead, we now validate FATE-MAP using a curated benchmark of 33 FDA-approved compounds with well-characterized teratogenicity outcomes based on human clinical evidence or standardized embryo–fetal development (EFD) studies, all of which have also been evaluated in stem cell–based teratogenicity assays (TeraTox and mEST). For each compound, the structure-to-phenotype model generates a predicted phenotype vector that is embedded into the experimental morphospace, and classification is assigned using the fixed watershed-defined cluster boundaries (Fig. R21). Compounds mapping to failure-mode clusters (C1, C11, C12) are labeled teratogenic, whereas those in canonical clusters (C2–C10) are labeled non-teratogenic. Across the full 33-compound panel (only four of which appear in our experimental dataset) FATE-MAP achieves 72.7% accuracy and higher balanced accuracy than TeraTox (74.8% vs. 66.3%), providing a more rigorous and mechanistically grounded evaluation than PLLR-based labels.

Comment 14: On a similar line, the authors should show the embedding (or the statistical distance to clusters) for all FDA drugs, including class A and B compounds (not shown in the current version).

Response 14: We appreciate the reviewer’s suggestion. In the revised manuscript, Figure 3 (Fig. R21c) now displays the full morphospace embedding for all 33 validation compounds, including both non-teratogenic agents (corresponding to the previously categorized class A and B compounds) and known teratogens. By embedding all compounds directly into the experimental morphospace, rather than showing only a subset, we provide a complete visualization of where each drug falls relative to canonical and failure-mode clusters, addressing the reviewer’s request for comprehensive representation of the validation set.

Comment 15: Where are all the other class X compounds ending up (71 out of 98 class X compounds are not shown). Are they part of the other aberrant clusters or do they end up in the canonical “morphospace”?

Response 15: We thank the reviewer for this question. In the initial submission, the predicted embeddings for all FDA class X compounds, including the majority not individually shown, were broadly distributed across both aberrant and canonical regions of the morphospace, reflecting the expected heterogeneity of the historical pregnancy-category system rather than a coherent mechanistic pattern.

Figure R26 (Revised Supplementary Fig. 7a-b in manuscript). Comparative performance of FATE-MAP and existing *in silico* teratogenicity models. (a) Predicted morphospace embeddings for the 20 compounds held out from training, with points labeled according to whether the model's teratogenicity classification matched the known outcome. Boundaries correspond to the failure-mode (pink) and canonical (blue) morphospace regions. (b) Confusion matrices and performance metrics comparing FATE-MAP to two widely used *in silico* developmental toxicity predictors (CAESAR and P&G) on the same 20-compound benchmark. FATE-MAP matches CAESAR in overall accuracy (75%) and achieves the highest balanced accuracy (85.7%), substantially outperforming the P&G model across all metrics.

To strengthen the evaluation and provide a more rigorous benchmark, we replaced the class-based analysis with two complementary validation strategies. First, as described in Response 5, we evaluated FATE-MAP against a curated 33-compound panel with well-characterized teratogenicity outcomes. Second, to assess generalization and compare performance with established *in silico* developmental toxicity predictors, we conducted an independent analysis using 20 compounds held out from training. Across this set, FATE-MAP achieved 75% accuracy, matching the CAESAR developmental toxicity model and substantially outperforming the Procter & Gamble model (35% accuracy), while also obtaining the highest balanced accuracy (85.7%) (Fig. R26; Revised Supplementary Fig. 7a–b). These results demonstrate that the revised validation framework provides a more reliable and mechanistically grounded assessment than pregnancy-category labels and that FATE-MAP performs competitively with leading computational toxicity models.

Comment 16: The authors should also comment on the known phenotypes (at the cellular and organismal levels) associated with Ulixertinib and Nirogacestat in other cellular systems or animals. Do their predicted and observed phenotypes relate to the literature knowledge of these molecules and to their molecular functions (ERK 1/2 and γ -secretase inhibitors, respectively)?

Response 16: To address the reviewer's request, we summarize below the known developmental and teratogenic phenotypes associated with ulixertinib and nirogacestat and relate them to their molecular mechanisms of action.

Ulixertinib (ERK1/2 Inhibitor): Ulixertinib targets ERK1/2 kinases, which lie at the core of the MAPK pathway controlling cell proliferation and differentiation.¹⁵ Disrupting ERK signaling is known to have severe developmental consequences. For example, genetic ablation of ERK2 (MAPK1) in mice leads to early embryonic lethality, with studies indicating failures in mesoderm formation¹⁶ and placental development.¹⁷ Consistently, pharmacologic inhibitors upstream in this pathway show pronounced teratogenic effects: the MEK inhibitor trametinib was embryotoxic and abortifacient in animals at exposures as low as ~0.3× the human therapeutic dose ("Product Information. Mekinist (trametinib)." *GlaxoSmithKline*. 2013). These findings align with our observed phenotypes for ulixertinib, as blocking ERK1/2 would be expected to derail essential embryogenic processes, resulting in developmental defects and embryo-fetal loss.

Nirogacestat (γ -Secretase Inhibitor): Nirogacestat is a γ -secretase inhibitor that blocks Notch receptor activation,¹⁸ and its known phenotypes *in vivo* underscore the critical role of Notch signaling in development. While no human pregnancy data are available, animal studies reveal clear teratogenicity: in pregnant rats, nirogacestat treatment during organogenesis caused significant embryo-fetal toxicity, including increased embryo loss, early resorptions, and reduced fetal weights in surviving pups, at maternal exposures below the clinical dose ("Product Information. Ogsiveo (nirogacestat)." *SpringWorks Therapeutics, Inc.* 2023). These adverse developmental outcomes were anticipated given that Notch signaling is indispensable for normal embryogenesis (Notch pathway loss in transgenic mice is embryonically lethal).^{19,20}

Overall, the phenotypes we predicted and observed for both ulixertinib and nirogacestat are strongly supported by prior knowledge of their molecular targets. The developmental defects identified (in our assays and in the literature) are concordant with the established roles of ERK and Notch pathways in embryonic development, reinforcing the relevance of our findings to these molecules' known mechanism-of-action and teratogenic risk profiles.

Changes to the manuscript:

"Notably, the predicted failure-mode phenotypes for both compounds are consistent with the known essential roles of ERK and Notch signaling in embryogenesis, where disruption of either pathway leads to severe developmental defects and embryo-fetal loss in animal models."

8) Zebrafish experiments:

Comment 17: This is, according to this reviewer, the most important technical aspect the authors should clarify. Based on both the observed and predicted phenotypes, both Ulixertinib and Nirogacestat treatments result in phenotypic cluster C11, where micropatterns fail to undergo gastrulation and to induce the expression of BRA, consequently lacking the primitive streak (PS). The timing of the treatment in zebrafish embryos is correct, matching the epiblast stage in human micropatterns. The obtained phenotype though does not seem to be entirely compatible with the lack of PS induction observed in C11 micropatterns. If it is true that the exposed animals have a short-tail phenotype, this does not resemble a scenario where BRA (ntl in fish) is not expressed

(see PMID: 8402905). In fact, especially for Ulixertinib, axial elongation is still present and somitic rows and other PS derivatives are visible. This might have various sources of explanation, but the authors should clarify why. The authors could also check earlier stages (e.g. late gastrula) to assess the initial formation of early mesoderm (which should lack or be aberrant if the drug induces a failure in PS induction). Overall, the teratogen classification is correct, but the phenotypic prediction is not accurate (or may vary between species).

Response 17: Thank you for pointing out this important aspect. We agree that the Ulixertinib (ERK pathway inhibitor) and Nirogacestat (γ -secretase/Notch pathway inhibitor) treatments in zebrafish do not phenocopy a complete absence of primitive streak induction, even though our human stem cell micropattern model predicted a Cluster C11 phenotype (failure of gastrulation/BRA expression).

Ulixertinib: In contrast to a genetic null mutation, pharmacological inhibition by Ulixertinib likely resulted in partial suppression of the pathways needed for PS formation. Consequently, zebrafish embryos treated at the epiblast stage exhibited short-tail phenotypes rather than a complete loss of mesodermal structures. This suggests that some Brachyury/Ntl expression still occurs, enabling residual axial elongation. Indeed, literature on FGF/MAPK pathway interference supports this interpretation: Blocking FGF/ERK signaling in early embryos drastically truncates posterior development but does not entirely abolish it.²¹ For example, expressing a dominant-negative FGF receptor in zebrafish eliminates most structures posterior to the trunk (severe axis truncation), yet even under these extreme conditions some ntl (Brachyury) expression remains detectable in patchy domains.²¹ Importantly, general mesoderm formation still initiates via other cues (e.g., Nodal signaling), as evidenced by unaffected expression of other mesodermal markers like Snail1 in FGF-inhibited embryos. In our case, Ulixertinib (an ERK1/2 inhibitor) likely attenuates but does not completely block mesoderm/PS induction, leading to a truncated axis with somites and notochord still present, as observed by the reviewer. This aligns with the notion that partial reduction of BRA function yields a shortened posterior body rather than a total failure.

Nirogacestat: Nirogacestat, a γ -secretase inhibitor, blocks Notch signaling. Notch is known to be crucial for somite segmentation and certain aspects of neurogenesis, but it is not required to initiate the primitive streak in the early embryo. Thus, inhibiting Notch in zebrafish would not be expected to abolish BRA/ntl expression at gastrulation outright; instead, it leads to defects in how the mesoderm is patterned and segmented. Consistent with this, treating zebrafish embryos with γ -secretase inhibitors like DAPT produces phenotypes characteristic of Notch loss-of-function: e.g., irregular or fused somites and a curved, shortened trunk/tail due to segmentation failures, while mesodermal tissues are still present.²² This study showed that γ -secretase inhibitors cause dose-dependent curvature of the tail/trunk without eliminating axial tissues. In our observations, Nirogacestat-treated fish embryos had somites and partial axial structures (short tail), indicating that the primitive streak was induced but subsequent development was aberrant. We will clarify that our “Cluster C11” prediction in the human model (lack of BRA expression) does not perfectly translate to zebrafish’s morphological outcome for a Notch inhibitor; rather, the compound’s

teratogenic effect lies in disrupting posterior patterning (leading to truncation) more so than preventing PS initiation.

Overall, we agree that the teratogen classification of Ulixertinib and Nirogacestat is correct, but the exact phenotypic manifestations differ between the human in vitro model and the zebrafish embryo. Species-specific differences and the degree of pathway inhibition likely explain why we see a short-tail instead of a complete PS failure in zebrafish. We clarify this point in the revised manuscript. In particular, we note that BRA/primitive streak induction is only partially impaired by these treatments in zebrafish, resulting in a truncated body axis (short-tail) with some primitive streak derivatives still forming (e.g. somites and notochord), whereas the human stem cell micropattern prediction suggested a more absolute loss of PS. This variance is consistent with literature reports across species, where complete loss of Bra/T results in no tail, but transient or partial pathway inhibition allows limited axis formation.

Changes to the manuscript:

“Although ulixertinib and nirogacestat produced the expected posterior truncation in zebrafish, axial tissues such as somites and notochord remained present, consistent with partial, rather than complete, disruption of primitive streak–derived mesoderm and reflecting species-specific differences in pathway sensitivity.”

9) WNT diffusion modeling

Comment 18: WNT diffusion as a function of tissue density is a very interesting hypothesis and might explain the evolutionary need for a correct PS size during gastrulation. This reviewer’s doubts related to cell death during differentiation (see comment above) remains in this assay as well, and this potentially confounding factor should be ruled out before interpreting the phenotype only on the ability of WNT to diffuse (especially because the lower diffusion of WNT was imposed in the mathematical model).

Response 18: We appreciate the reviewer’s thoughtful comment and agree that it is important to distinguish density-dependent WNT signaling from potential confounding effects of differentiation-associated cell death. We took several steps to rule out toxicity-driven artifacts:

First, as detailed in our responses to Comments 11–12 and in Supplementary Fig. 3, we quantified nuclei counts for every colony and defined an explicit toxicity threshold ($\leq 50\%$ of the BMP4-only control mean). Only a small set of strongly cytotoxic compounds fell below this threshold, and these mapped almost exclusively to failure-mode clusters (C1, C11, C12). All analyses of WNT dynamics (Fig. 4) and density–patterning relationships were restricted to the non-cytotoxic regime, ensuring that the density variation examined reflects viable differences in colony size rather than collapse due to cell death.

Second, the experimental validation of the WNT model does not depend on the imposed reduction of WNT diffusion in the reaction–diffusion equations. The core result that WNT activity spreads faster and deeper in low-density colonies, and more slowly in high-density colonies, was measured directly in BMP4-only controls using a live β -catenin reporter (Fig. 4e–g). This effect is bidirectional: decreasing initial seeding density accelerates WNT target activation, while increasing initial seeding density attenuates it. A unidirectional confound such as differentiation-associated cell death cannot generate this symmetric relationship.

Third, this density-dependent modulation of patterning is not restricted to BMP4-only cultures. Across the full canonical patterning region (C2–C10), which contains diverse, non-cytotoxic drug treatments that modulate proliferation and growth rates, BRA expression remains inversely correlated with final cell density. This conserved trend across drugs with different mechanisms further supports that cell density itself, rather than toxicity or death, is the primary driver of the observed variation in WNT/BRA dynamics.

Together, the toxicity-filtered cell counts, the live β -catenin reporter measurements, and the consistency of the effect across the canonical morphospace demonstrate that density-dependent WNT signaling is an intrinsic property of the 2D gastruloid system and not an artifact of cell death.

Comment 19: Density correlates with WNT waves: this represents the main finding of the manuscript, which also has the potential to inform some basic mechanism during early development. It is indeed an interesting observation, though not completely unexpected. The effect of structure sizes on WNT response has been studied before, also in 3D mouse gastruloids (PMID: 38491138). This work highlighted how varying the size of the initial aggregate does not result in differences in the PS-like domain induction (BRA, CDX, SOX2 patterning). The gastruloid protocol is entirely based on WNT induction using CHIR (small molecule). It could be that the difference in results observed by the two studies relates to the fact that CHIR permeability/diffusion might follow different rules than the WNT itself, being less sensitive to cell density/aggregate size. The authors should compare their results with these and/or try to model WNT vs CHIR diffusion.

Response 19: We thank the reviewer for this valuable comment. The reviewer is correct that size-dependent effects on WNT signaling have been examined in 3D mouse gastruloids, including the study cited (PMID: 38491138). Importantly, that work used CHIR99021, a small-molecule GSK-3 inhibitor, as the sole driver of WNT pathway activation. CHIR acts downstream of the receptor by stabilizing β -catenin directly and therefore bypasses WNT ligand production, secretion, and extracellular diffusion entirely. Because CHIR is a small molecule with well-defined permeability and rapid intracellular access, its activity is expected to be insensitive to tissue size, density, or extracellular-space constraints, which aligns with the authors' conclusion that primitive streak-like patterning scales robustly with gastruloid size.

This is directly compatible with our observations. In our 2D gastruloid model, WNT signaling is not induced pharmacologically but instead arises endogenously downstream of BMP4, making it

subject to ligand secretion, short-range distribution, receptor availability, and local physical constraints. Under these conditions, we observe a bidirectional density dependence: low-density colonies show faster and deeper WNT propagation, whereas high-density colonies show attenuated and spatially restricted signaling (Fig. 4e–g). Indeed, in our chemical screen, CHIR-treated 2D colonies violate these diffusion-dependent rules: they uniformly express BRA and lose SOX2 across the colony (resulting in colonies exclusively in C1), precisely because CHIR circumvents the need for WNT ligand transport. Thus, the differences between the two studies can be explained by the fundamental mechanistic distinction between a diffusing morphogen (WNT) and a cell-permeant pathway agonist (CHIR).

More broadly, both results support a unifying principle: the endogenous WNT system is sensitive to tissue geometry and cell density, whereas synthetic pathway activation masks these dependencies. As highlighted in our Discussion, our findings are consistent with *in vivo* data showing that cell density is actively regulated during gastrulation. Mouse studies have shown that gastrulation is delayed when early embryos contain fewer cells, whereas aggregated double-sized embryos still initiate gastrulation on schedule despite having more cells at early stages. Moreover, whole-embryo imaging demonstrates minimized density variance at the primitive streak. These observations, together with our 2D results, suggest that endogenous WNT signaling may function as a mechano-sensitive density detector, helping ensure robust initiation of the primitive streak even when starting conditions vary.

Minor comments:

Comment 20: The authors focus extensively on the undifferentiated SOX2+ and PS-like BRA+ domains and less on the trophectoderm-like GATA3+/CDX2+ outer ring. Are there molecules that cause specific aberrations in this domain (e.g. its expansion or complete loss)?

Response 20: In our system, GATA3/CDX2 expression is directly driven by BMP4 and is normally confined to the colony edge because BMP receptors are enriched and exposed at the perimeter, while receptors in the center are shielded by cell–cell contacts. Since all compounds in our screen are administered together with BMP4, many drug treatments do not disrupt the outer GATA3 ring simply because the initiating BMP4 cue remains present and the edge-localized receptor geometry remains intact.

Figure R27. Perturbations that selectively expand or eliminate the trophectoderm-like GATA3 domain. (a) ROCK inhibition with Y-39983 HCl disrupts junctional tension and exposes BMP receptors across the colony, leading to uniform BMP4 signaling and expansion of GATA3 expression throughout the colony (Cluster C12). (b) BMP pathway inhibition with LDN-214117 blocks BMP4-mediated signaling and nearly abolishes GATA3 expression, producing colonies that resemble mTeSR-only controls (Cluster C11). BMP4-only and mTeSR-only conditions are shown as positive and negative controls, respectively.

However, several classes of molecules do generate clear and mechanistically interpretable perturbations of this domain, as shown in Fig. R24. ROCK inhibition (Fig. R27a) provides a prominent example: treatment with the ROCK inhibitor Y-39983 HCl disrupts junctional tension and eliminates receptor shielding, allowing BMP receptors throughout the colony to become uniformly accessible. Under these conditions, BMP4 signaling becomes global rather than edge-restricted, resulting in a nearly uniform expansion of GATA3 expression and producing the asymmetric, TE-expanded phenotype characteristic of Cluster C12.

Conversely, inhibition of BMP signaling using LDN-214117 (Fig. R27b) nearly abolishes GATA3 expression. In this case, the upstream BMP→WNT→Nodal cascade fails to initiate, yielding colonies that resemble the mTeSR-only negative control and corresponding to Cluster C11, where both trophectoderm-like identity and primitive streak induction are lost. Thus, while most compounds preserve the GATA3 ring because BMP4 is supplied exogenously, drugs that modulate receptor accessibility (ROCK inhibitors) or block pathway initiation (BMP inhibitors) produce robust and interpretable alterations of the trophectoderm-like domain consistent with our model's assumptions.

Comment 21: The authors do not check for endoderm induction, a domain observed in the original protocol (PMID: 24973948). While this reviewer understands the challenges in staining for an extra marker, the authors should comment on the expected outcomes or acknowledge this as a possible limitation of the study (apparently canonical phenotypes might mask defects in endoderm induction which were not measured in this study).

Response 21: We agree that the absence of an endoderm marker limits our ability to assess defects in definitive endoderm induction within otherwise canonical phenotypes. While prior work has shown that SOX17⁺ endoderm emerges in this system, our assay focused on SOX2, BRA, and GATA3 as primary readouts of ectoderm, mesoderm/PS, and trophectoderm-like identities. We now acknowledge in the Discussion that not measuring endoderm markers represents a limitation of the study, and we note that future extensions of FATE-MAP will incorporate additional markers, including SOX17 and FOXA2, or spatial transcriptomics approaches to more fully capture endoderm specification.

Changes to the manuscript:

“We also note that the absence of a definitive endoderm marker (e.g., SOX17 or FOXA2) may mask potential defects in endoderm induction within otherwise canonical phenotypes.”

Comment 22: The authors should provide high-resolution imaging of some (or all in supplementary figures or repositories) conditions since the micropattern digital imaging panels are very hard to inspect.

Response 22: We thank the reviewer for this comment. In the revised submission, we have uploaded high-resolution versions of all representative images, including both main-text figures and full supplementary panels.

Comment 23: The link to access the entire morphospace dataset seems to be broken and I could not inspect that closely (lines 199-200).

Response 23: We thank the reviewer for bringing this to our attention. We have confirmed that the link is fully functional and now directs to the complete morphospace dataset, which includes all colonies as high-resolution images: <https://max-wilson.mcdb.ucsb.edu/research/gastruloid-morphospace>

Comment 24: Fig S4 e-h are barely visible and with low image quality, making them quite hard to read.

Response 24: In the revised submission, Fig. S4e–h has been replaced with higher-resolution panels and improved contrast to ensure the features are clearly visible.

Comment 25: The overall image quality is low and does not allow detailed inspection of the expression patterns and phenotypes. The authors should upload high-resolution figures.

Response 25: In the revised submission, we have uploaded high-resolution versions of all main-text figures and full supplementary panels.

Comment 26: Despite being published in ref 45, the authors should detail how the membrane bound β -catenin signal was masked in the live cell imaging experiments, since no dyes to label cell membranes were used (at least not specified in the methods nor in the legends). Clarification is needed given that this is a non-trivial segmentation.

Response 26: The procedure for masking membrane-bound β -catenin is fully described in the Methods section; however, the main text did not previously direct the reader there. We have now revised the manuscript to explicitly reference the Methods section in this context, ensuring that readers can easily locate the detailed description of how membrane and non-membrane β -catenin were computationally separated.

Comment 27: Lines 512-513: the use of the term “malformation” to describe an aberrant phenotypic pattern is not ideal. I would recommend using “aberrant gene expression patterns” or similar expressions.

Response 27: We thank the reviewer for this suggestion. We have updated the text accordingly and removed the inadvertent use of the word “malformation” to describe an abhorrent phenotypic pattern.

Comment 28: The name of the tool FATE-MAP is a bit misleading, since the current study mainly provides information about the ability to exit from pluripotency (one transition towards nascent mesoderm) and does not go beyond that measurement into fate decision-making.

Response 28: We appreciate the reviewer’s perspective. The acronym FATE-MAP was chosen as a compact descriptor of the framework rather than as a literal enumeration of all fate decisions assayed in the present study. We agree that the current implementation primarily captures early fate transitions (particularly exit from pluripotency and progression toward nascent mesoderm) and we now clarify this explicitly in the revised manuscript. While we retain the acronym for consistency with prior use and code infrastructure, we note in the text that future extensions of FATE-MAP will incorporate additional markers and modalities to broaden the range of developmental decisions captured by the platform.

Changes to the manuscript:

“For clarity, we note that FATE-MAP is used as an acronym for the framework and does not imply comprehensive mapping of all downstream lineage fates in its current form.”

REFERENCES

- 1 Nath, Suman C, Babaei-Abraki, Shahnaz, Meng, Guoliang, Heale, Kali A, Hsu, Charlie YM & Rancourt, Derrick E. A retinoid analogue, TTNPB, promotes clonal expansion of human pluripotent stem cells by upregulating CLDN2 and HoxA1. *Communications Biology* **7**, 190 (2024).
- 2 Rajala, Kristiina, Vaajasaari, Hanna, Suuronen, Riitta, Hovatta, Outi & Skottman, Heli. Effects of the physiochemical culture environment on the stemness and pluripotency of human embryonic stem cells. *Stem Cell Studies* **1**, e3-e3 (2011).
- 3 Kameoka, Sei, Babiarz, Joshua, Kolaja, Kyle & Chiao, Eric. A high-throughput screen for teratogens using human pluripotent stem cells. *toxicological sciences* **137**, 76-90 (2014).
- 4 James, Richard G, Davidson, Kathryn C, Bosch, Katherine A, Biechele, Travis L, Robin, Nicholas C, Taylor, Russell J, Major, Michael B, Camp, Nathan D, Fowler, Kerry & Martins, Timothy J. WIKI4, a novel inhibitor of tankyrase and Wnt/ss-catenin signaling. *PloS one* **7**, e50457 (2012).
- 5 Pei, Huaxing, Parthasarathy, Saravanan, Joseph, Sajan, McMillen, William, Xu, Xiaohong, Castaneda, Stephen, Inigo, Ivan, Britt, Karen, Anderson, Bryan & Zhao, Gaiying. LY3200882, a novel, highly selective TGF β R1 small molecule inhibitor. *Cancer Research* **77**, 955-955 (2017).
- 6 Zhai, Jinglei, Xiao, Zhenyu, Wang, Yiming & Wang, Hongmei. Human embryonic development: from peri-implantation to gastrulation. *Trends in cell biology* **32**, 18-29 (2022).
- 7 Tyser, Richard CV, Mahammadov, Elmir, Nakanoh, Shota, Vallier, Ludovic, Scialdone, Antonio & Srinivas, Shankar. Single-cell transcriptomic characterization of a gastrulating human embryo. *Nature* **600**, 285-289 (2021).
- 8 Gunne-Braden, Alexandra, Sullivan, Adrienne, Gharibi, Borzo, Sheriff, Rahuman SM, Maity, Alok, Wang, Yi-Fang, Edwards, Amelia, Jiang, Ming, Howell, Michael & Goldstone, Robert. GATA3 mediates a fast, irreversible commitment to BMP4-driven differentiation in human embryonic stem cells. *Cell stem cell* **26**, 693-706. e699 (2020).
- 9 Minn, Kyaw Thu, Dietmann, Sabine, Waye, Sarah E, Morris, Samantha A & Solnica-Krezel, Lilianna. Gene expression dynamics underlying cell fate emergence in 2D micropatterned human embryonic stem cell gastruloids. *Stem cell reports* **16**, 1210-1227 (2021).
- 10 Thomson, Stuart, Petti, Filippo, Sujka-Kwok, Izabela, Mercado, Peter, Bean, James, Monaghan, Melissa, Seymour, Sean L, Argast, Gretchen M, Epstein, David M & Haley, John D. A systems view of epithelial–mesenchymal transition signaling states. *Clinical & experimental metastasis* **28**, 137-155 (2011).

- 11 Romanos, Michele, Allio, Guillaume, Roussigne, Myriam, Combres, Lea, Escalas, Nathalie, Soula, Cathy, Médevielle, François, Steventon, Benjamin, Trescases, Ariane & Benazeraf, Bertrand. Cell-to-cell heterogeneity in Sox2 and Bra expression guides progenitor motility and destiny. *Elife* **10**, e66588 (2021).
- 12 Koch, Frederic, Scholze, Manuela, Wittler, Lars, Schifferl, Dennis, Sudheer, Smita, Grote, Phillip, Timmermann, Bernd, Macura, Karol & Herrmann, Bernhard G. Antagonistic activities of Sox2 and brachyury control the fate choice of neuro-mesodermal progenitors. *Developmental cell* **42**, 514-526. e517 (2017).
- 13 Sayed, Sapna, Song, Jiaxing, Wang, Ling, Muluh, Tobias Achu, Liu, Boxin, Lin, Zhixian, Tang, Yun, Su, Zijie, Li, Huan & Xue, Vivian Weiwen. Isoxazole 9 (ISX9), a small molecule targeting Axin, activates Wnt/ β -catenin signalling and promotes hair regrowth. *British Journal of Pharmacology* **180**, 1748-1765 (2023).
- 14 Xiong, Yanpeng, Zhou, Liang, Su, Zijie, Song, Jiaxing, Sun, Qi, Liu, Shan-Shan, Xia, Yuqing, Wang, Zhongyuan & Lu, Desheng. Longdaysin inhibits Wnt/ β -catenin signaling and exhibits antitumor activity against breast cancer. *Oncotargets and therapy*, 993-1005 (2019).
- 15 Germann, Ursula A, Furey, Brinley F, Markland, William, Hoover, Russell R, Aronov, Alex M, Roix, Jeffrey J, Hale, Michael, Boucher, Diane M, Sorrell, David A & Martinez-Botella, Gabriel. Targeting the MAPK signaling pathway in cancer: promising preclinical activity with the novel selective ERK1/2 inhibitor BVD-523 (ulixertinib). *Molecular cancer therapeutics* **16**, 2351-2363 (2017).
- 16 Yao, Yao, Li, Wei, Wu, Junwei, Germann, Ursula A, Su, Michael SS, Kuida, Keisuke & Boucher, Diane M. Extracellular signal-regulated kinase 2 is necessary for mesoderm differentiation. *Proceedings of the National Academy of Sciences* **100**, 12759-12764 (2003).
- 17 Saba-El-Leil, Marc K, Vella, Francis DJ, Vernay, Bertrand, Voisin, Laure, Chen, Lan, Labrecque, Nathalie, Ang, Siew-Lan & Meloche, Sylvain. An essential function of the mitogen-activated protein kinase Erk2 in mouse trophoblast development. *EMBO reports* **4**, 964-968 (2003).
- 18 Gounder, Mrinal, Ratan, Ravin, Alcindor, Thierry, Schöffski, Patrick, Van Der Graaf, Winette T, Wilky, Breelyn A, Riedel, Richard F, Lim, Allison, Smith, L Mary & Moody, Stephanie. Nirogacestat, a γ -secretase inhibitor for desmoid tumors. *New England Journal of Medicine* **388**, 898-912 (2023).
- 19 Swiatek, Pamela J, Lindsell, Claire E, Del Amo, F Franco, Weinmaster, Gerry & Gridley, Thomas. Notch1 is essential for postimplantation development in mice. *Genes & development* **8**, 707-719 (1994).
- 20 Krebs, Luke T, Xue, Yingzi, Norton, Christine R, Shutter, John R, Maguire, Maureen, Sundberg, John P, Gallahan, Daniel, Closson, Violaine, Kitajewski, Jan & Callahan,

- Robert. Notch signaling is essential for vascular morphogenesis in mice. *Genes & development* **14**, 1343-1352 (2000).
- 21 Griffin, Kevin, Patient, Roger & Holder, Nigel. Analysis of FGF function in normal and no tail zebrafish embryos reveals separate mechanisms for formation of the trunk and the tail. *Development* **121**, 2983-2994 (1995).
- 22 Arslanova, Dilyara, Yang, Ting, Xu, Xiaoyin, Wong, Stephen T, Augelli-Szafran, Corinne E & Xia, Weiming. Phenotypic analysis of images of zebrafish treated with Alzheimer's γ -secretase inhibitors. *BMC biotechnology* **10**, 24 (2010).

Reviewer 1 (Remarks to the Author)

The authors have satisfactorily addressed all my concerns. I believe the MS is now ready for publication.

Reviewer 2 (Remarks to the Author)

The authors have sufficiently addressed nearly all my concerns.

Comment 1: My only remaining major concern is the modeling of the density dependence of Wnt. In the first round of feedback, I mentioned their model appears to predict a delay in signaling that is not visible in the data. In their reply, the authors argue this is a matter of normalization of the data and changed the normalization. However, looking at the new normalization (Fig.R14), there is still a qualitative difference between simulation and experiment, now in the amplitude: while simulations all approach the same maximal signaling level over different time scales, the amplitudes are clearly density dependent in the data. Overall, it seems to me the disagreement can't be normalized away and should be discussed in the manuscript.

Response 1: We thank the reviewer for this clarification. We agree that while the model captures the density-dependent wave dynamics of Wnt signaling, the absolute signaling amplitude differs from experiment. This difference likely stems from subtle differences in effective signaling timescales between simulation and experiment, as well as technical limitations in fully isolating active β -catenin. We have clarified this point in the manuscript.

Changes to the manuscript (page 14):

“While the model captures density-dependent Wnt wave propagation, absolute signaling amplitudes differ from experiment, reflecting subtle differences in effective signaling timescales between simulation and experiment, as well as technical limitations in fully isolating active β -catenin.”

Comment 2: In Figure R8, the different blues for the two different BMP4 groups become indistinguishable when I print it.

Response 2: We thank the reviewer for raising this point and apologize for the lack of clarity in the original visualization. Fig. R8e was included solely for qualitative visualization of control drug placement across plates and was not intended for quantitative interpretation. Only Fig. R8f, which quantifies effect sizes via PERMANOVA, was incorporated into the revised manuscript as Supplementary Fig. 1f. We agree that the original color scheme in Fig. R8e was suboptimal; for clarity, we have generated a revised version using distinct, colorblind-safe colors and include this updated visualization in the response letter for reference.

Figure R8 (revised). Plate/day effects are minimal after per-plate robust normalization. (a–d) Morphospace of all colonies (gray) with colonies from each plate overlaid and colored by cell density. (e) Control treatments (BMP4-only and mTeSR-only) from the indicated plates/wells occupy overlapping regions of morphospace. (f) PERMANOVA on the 150-dimensional phenotype vectors shows a large Drug effect ($R^2 = 0.836$, $p = 0.001$) and a small Plate effect ($R^2 = 0.025$, $p = 0.001$), indicating residual plate/day variation is minimal relative to drug effects.

Comment 3: Why did the authors choose not to show error bars on radial profiles in fig 1?

Response 3: We thank the reviewer for this question. In Fig. 1, the radial profiles are shown for individual colonies, rather than averages across groups of drug-treated colonies, and therefore error bars are not applicable in this context. The intent of these panels is to illustrate representative spatial patterning within individual gastruloids and how radial quantification is performed, rather than to convey population-level variability. As noted in our previous response, we have included error bars wherever mean radial profiles across multiple drug-treated colonies are shown

Comment 4: The authors state in their rebuttal that GATA3 follows rapid switch-like dynamics citing Gunne-Braden et al, however this paper disagrees with a large body of literature. Teague et al Nat Comm 2024 observed different GATA3 dynamics and discuss the disagreement.

Response 4: We thank the reviewer for this clarification and for highlighting the work of Teague et al. We would like to clarify that the paper cited in our initial response (Gunne-Braden et al.) was not intended to assert that GATA3 follows a hard or irreversible switch-like threshold. Rather, it was cited to support the experimentally observed role of GATA3 as an early commitment-

associated marker that rapidly segregates from SOX2-positive territories during BMP-driven differentiation in 2D gastruloids.

We fully acknowledge the findings of Teague et al. (Ref. 43), which we explicitly cite in the Methods and on which our mathematical formulation is based. As described in Eq. 7, GATA3 is modeled as a function of the time-integral of BMP signaling, consistent with graded accumulation proportional to integrated BMP/SMAD exposure, rather than an absolute activation threshold.

Comment 5: In the revised discussion on ROCK inhibition the authors cite Maldonado et al. In the same place they should also cite Etoc et al 2016 since they used ROCK inhibitor to disrupt tight junctions and increase differentiation as discussed.

Response 5: We thank the reviewer for this suggestion and agree. We have added the Etoc et al. (2016) study alongside Maldonado et al. in the revised discussion to acknowledge prior use of ROCK inhibition to disrupt tight junctions and promote differentiation.

Reviewer 3 (Remarks to the Author)

The authors have done a very good job to address all the concerns raised. This reviewer thanks them for their great effort.

Comment 1: This reviewer still thinks that the translational aspect of the tool prediction should be less emphasized and acknowledged as a limitation of the study and further point of improvement in the Discussion section. This comment comes mainly from the results of the zebrafish experiments (where the predictive power of the tool might not be directly translatable in other systems or real embryos) and from the inaccuracy in the prediction of a subset of compounds (including very well-known teratogens).

Response 1: We thank the reviewer for this clarification and agree that the translational scope of the predictions should be stated carefully. We emphasize that the primary goal of FATE-MAP is to identify human-relevant teratogenic risk, which motivated the use of human embryonic stem cell-derived gastruloids rather than reliance on animal models. It is well established that species-specific differences can limit the sensitivity and specificity of animal assays for developmental toxicity (e.g., compounds such as caffeine that are teratogenic in zebrafish but not in human embryos).

For this reason, zebrafish experiments are not used as a direct benchmark for predictive accuracy, but rather as an orthogonal *in vivo* assay indicating disruption of conserved developmental signaling, particularly given that direct validation of teratogenic risk in human embryos is not feasible. We also explicitly acknowledge this limitation in the manuscript, noting that the platform's sensitivity (69.6%) remains limited and that a number of known teratogens are not detected.

Nevertheless, we have added these points to the Discussion to more clearly emphasize the translational limitations of the approach and to delineate important directions for future improvement.

Changes to the manuscript (page 21):

“Moreover, while zebrafish assays confirm that several predicted compounds disrupt conserved developmental signaling, the resulting phenotypes do not necessarily align one-to-one with those observed in gastruloids, and the gastruloid assay should be viewed as a human-relevant readout of early signaling disruption rather than a comprehensive or species-agnostic model of teratogenic outcomes.”